# Broadening Target Distributions for Accelerated Diffusion Models via a Novel Analysis Approach

**Yuchen Liang**[†], **Peizhong Ju**[‡], **Yingbin Liang**[†], **Ness Shroff**[†]
[†]The Ohio State University    [‡]University of Kentucky

## Abstract

Accelerated diffusion models hold the potential to significantly enhance the efficiency of standard diffusion processes. Theoretically, these models have been shown to achieve faster convergence rates than the standard $\mathcal{O}(1/\epsilon^2)$ rate of vanilla diffusion models, where $\epsilon$ denotes the target accuracy. However, current theoretical studies have established the acceleration advantage only for restrictive target distribution classes, such as those with smoothness conditions imposed along the entire sampling path or with bounded support. In this work, we significantly broaden the target distribution classes with a new accelerated stochastic DDPM sampler. In particular, we show that it achieves accelerated performance for three broad distribution classes not considered before. Our first class relies on the smoothness condition posed only to the target density $q_0$, which is far more relaxed than the existing smoothness conditions posed to all $q_t$ along the entire sampling path. Our second class requires only a finite second moment condition, allowing for a much wider class of target distributions than the existing finite-support condition. Our third class is Gaussian mixture, for which our result establishes the first acceleration guarantee. Moreover, among accelerated DDPM type samplers, our results specialized for bounded-support distributions show an improved dependency on the data dimension $d$. Our analysis introduces a novel technique for establishing performance guarantees via constructing a tilting factor representation of the convergence error and utilizing Tweedie's formula to handle Taylor expansion terms. This new analytical framework may be of independent interest.

## 1 Introduction

Generative modeling is a fundamental task in machine learning, aiming to generate samples out of a distribution similar to that of training data. Classical generative models include variational autoencoders (VAE) (Kingma & Welling, 2022), generative adversarial networks (GANs) (Goodfellow et al., 2014), and normalizing flows Rezende & Mohamed (2015), etc. Recently, diffusion models (Sohl-Dickstein et al., 2015; Ho et al., 2020; Song & Ermon, 2019) have arisen as an appealing generative model and have received wide popularity due to their excellent performance over a variety of tasks and applications as summarized in many surveys of diffusion models (Yang et al., 2023; Croitoru et al., 2023; Kazerouni et al., 2023).

The empirical success of diffusion models has also inspired extensive theoretical studies, aiming to characterize the convergence guarantee for diffusion models. The convergence rate (i.e., the total number of steps to attain a target accuracy $\varepsilon$) for standard vanilla Denoising Diffusion Probabilistic Models (DDPMs) has been established to be $\mathcal{O}(\varepsilon^{-2})$ for wide classes of target distributions (Chen et al., 2023a; Benton et al., 2024a; Conforti et al., 2023) (see Appendix A for a more complete summary). More recently, various **accelerated** samplers have been proposed and been shown to achieve an improved convergence rate of $\mathcal{O}(\varepsilon^{-1})$. One such acceleration approach is to redesign the (stochastic) DDPM reverse process. This includes augmenting the original reverse process with an additional estimate (Li et al., 2024c), introducing intermediate sampling points along the generation path (Li et al., 2024a), and employing special Markov-chain Monte-Carlo (MCMC) algorithms (Huang et al., 2024b). Another acceleration method is to sample with the corresponding probability ODE (Li et al., 2024c; Chen et al., 2023c; Huang et al., 2024a; Li et al., 2024d).

| Target distribution $Q_0$ | Method | Num of steps | Results |
|---|---|---|---|
| $\nabla \log q_t, s_t$ $L$-Lips. $\forall t$ | ODE-based | $\mathcal{O}\left(\frac{\sqrt{d}L^2}{\varepsilon}\right)$ | (Chen et al., 2023c, Thm 3) |
| $\nabla \log q_t$ $L$-Lips. $\forall t$ | DDPM accl. | $\mathcal{O}\left(\frac{\sqrt{d}L^2}{\varepsilon}\right)$ | (Huang et al., 2024b, Thm 4.4)[†] |
| $\left|\partial_{\boldsymbol{a}}^k s_t(x)\right| \leq L$ $\forall x, t, \boldsymbol{a}$ and $\forall k \leq p+1$, $Q_0$ Bounded Support | ODE | $\mathcal{O}\left(\frac{d^{\frac{p+1}{p}}}{\varepsilon^{\frac{1}{p}}}\right)^*$ | (Huang et al., 2024a, Thm 3.10)[†] |
| $\nabla^2 \log q_0$ $M$-Lips. | DDPM accl. | $\mathcal{O}\left(\frac{d^{1.5}\log^{1.5}M}{\varepsilon}\right)$ | (This paper, Thm 4) |
| $Q_0$ Gaussian Mixture | DDPM accl. | $\mathcal{O}\left(\frac{d^{1.5}N^{1.5}}{\varepsilon}\right)$ | (This paper, Thm 2) |
| $Q_0$ Bounded Support | DDPM accl. | $\mathcal{O}\left(\frac{d^3}{\varepsilon}\right)^*$ | (Li et al., 2024c, Thm 4) (Li et al., 2024a, Thm 2)[†] |
| | ODE | $\mathcal{O}\left(\frac{d^3}{\sqrt{\varepsilon}}\right)^*$ | (Li et al., 2024c, Thm 2) (Li et al., 2024a, Thm 1)[†] |
| | ODE | $\mathcal{O}\left(\frac{d^2}{\varepsilon}\right)^*$ | (Li et al., 2024c, Thm 1) |
| $Q_0$ Finite Variance | DDPM accl. | $\mathcal{O}\left(\frac{d^{1.5}}{\varepsilon}\right)^*$ | (This paper, Thm 3) |

Table 1: *Summary of accelerated convergence results in terms of the number of steps needed to achieve $\varepsilon$-accuracy in total variation, where $d$ is the dimension. For Gaussian mixture, assume that $N \leq d$. The first 4 rows of this table correspond to the results under those target distributions with some smoothness conditions imposed, while the last 4 rows correspond to the results under (possibly) non-smooth targets with finite variance. ($*$) Those results correspond to an* early-stopped *procedure that compares the sampling distribution to $Q_1(\delta)$, where $\mathrm{W}_2\left(Q_0, Q_1\right)^2 \lesssim \delta d$. Here the dependencies on $\delta$ are omitted. ($\dagger$) Those studies are concurrent to our work based on the time that they were posted on arXiv. Note that this table does not include the studies within two months of the conference submission, but those are discussed in the related works.*

However, existing results on the acceleration guarantee suffer from strong assumptions on the target distribution. (i) For smooth target distributions, the analyses of Chen et al. (2023c); Huang et al. (2024a;b) require that all the scores (or their close estimates or both) satisfy certain Lipschitz-smooth condition *along the entire sampling path*, i.e., the smoothness condition is posed to the density $q_t$ for all iteration time $t$. However, such smoothness at intermediate steps is generally restrictive and hard to verify in practice. (ii) For (possibly) non-smooth targets, the analysis of Li et al. (2024a;c;d) requires the distribution to have finite support for early-stopped sampling procedures. Such an assumption is, however, restrictive if compared to that for early-stopped vanilla samplers, where convergence guarantees have been established only under the assumption of finite variance (Chen et al., 2023a; Benton et al., 2024a). The above discussions raise the following important open question:

*Question 1: Can we obtain an accelerated convergence rate for a much broader set of target distributions? Namely, for smooth target distributions, can the smoothness condition be imposed only on the target distribution; and for (possibly) non-smooth targets, can we broaden the target distribution to only have finite variance?*

Further, the existing accelerated diffusion samplers suffer as high dimensional dependencies as $\mathcal{O}\left(d^3\right)$ or $\mathcal{O}\left(d^2\right)$ (Li et al., 2024a;c) for target distributions with bounded support. This motivates us to explore the following intriguing question:

*Question 2: While addressing Question 1 to relax the assumption from finite support to finite variance for possibly non-smooth distributions, can we achieve a lower dimensional dependency?*

This paper will provide affirmative answers to both of the above questions.

## 1.1 OUR CONTRIBUTIONS

Our main contribution is to provide accelerated convergence results for a significantly wider range of distributions than those addressed in previous works (see Table 1 (particularly column 1) for a comparison). To this end, we design a new accelerated stochastic DDPM sampler and develop a novel analytical technique that characterizes its acceleration guarantees across this broader spectrum of distributions. Our detailed contributions are summarized as follows.

**Broadening Target Distributions**: Inspired by optimization methods, we design a new Hessian-based accelerated sampler for the stochastic diffusion processes. We show that our accelerated sampler achieves an accelerated convergence rate of $\mathcal{O}\left(d^{1.5}\min\{d, N\}^{1.5}/\varepsilon\right)$, $\mathcal{O}\left(d^{1.5}/\varepsilon\right)$, and $\mathcal{O}\left(d^{1.5}\log^{1.5}M/\varepsilon\right)$ respectively for Gaussian mixtures, any target distributions having finite variance (with early-stopping), and any target distributions having $M$-Lipschitz Hessian of log-densities. In particular, (i) for smoothness $Q_0$ that has p.d.f., the smoothness condition is only imposed on the log-density of $Q_0$, which is much less restrictive than that imposed on all $Q_t$'s (Chen et al., 2023c; Huang et al., 2024a;b); (ii) for possibly non-smooth $Q_0$, we only require $Q_0$ to have finite variance for the early-stopped procedure, which is a much broader class of distributions than those having bounded support (Li et al., 2024a;c;d); (iii) we provide the first accelerated convergence result for Gaussian mixture $Q_0$'s.[1]

For possibly non-smooth targets with bounded support, our sampler improves the dependency of the convergence rate on $d$ by $\mathcal{O}\left(d^{1.5}\right)$ compared with previous accelerated diffusion samplers (Li et al., 2024a;c).

**Novel Analysis Technique**: We develop a novel technique for analyzing the accelerated DDPM process. Our approach features two new elements: (i) characterization of the error incurred at each discrete step of the reverse process using *tilting factor*; and (ii) analysis of the mean value of tilting factor via *Tweedie's formula* to handle power terms in the Taylor expansion. Such a technique enables us to (a) analyze more general distributions beyond those with restrictive distribution assumptions; (b) tightly identify the dominant term and reduce the dimensional dependency; and (c) handle the estimation error in accelerated samplers for both score and Hessian estimation. This analytical framework is different from the main previous theoretical techniques for analyzing the convergence of diffusion models: (a) the SDE-type analysis for regular diffusion samplers (Chen et al., 2023a; Benton et al., 2024a; Conforti et al., 2023), (b) any ODE-type analysis (Li et al., 2024d; Huang et al., 2024a; Gao & Zhu, 2024), and (c) the use of typical sets (Li et al., 2024a;c).

## 1.2 RELATED WORKS ON ACCELERATED SAMPLING

Here, we focus on the related studies of accelerated samplers. Note that all of these works we discuss below, only except Chen et al. (2023c;e); Li et al. (2024c), are concurrent to or after ours based on their posting time on arXiv. In Appendix A, we provide a thorough summary of convergence analysis of standard samplers as well as other theoretical perspectives of diffusion models.

**Accelerated Stochastic Samplers:** In Li et al. (2024c), accelerated stochastic variants to the original DDPM sampler are proposed and analyzed, *when there is no estimation error*. In Li et al. (2024a), a new accelerated stochastic sampler are proposed by inserting intermediate sampling points along the diffusion path. Both algorithms are analyzed only when the target distribution has bounded support and suffer from large dimensional dependencies. In Huang et al. (2024b), the authors proposed the RTK-MALA and RTK-ULD algorithms which uses MCMC algorithms, such as the Metropolis-adjusted Langevin Algorithm or the Underdamped Langevin Dynamics, at each diffusion step. The analysis is performed under the assumption that all the scores of $\log q_t$'s are Lipschitz-smooth. In comparison, our work substantially broadens the set of target distributions to include those with unbounded support and with smooth log-density only imposed upon $Q_0$ with a completely different analytical technique. Our result also improves the dimensional dependencies of accelerated stochastic samplers in Li et al. (2024a;c) for distributions with bounded support.

**Deterministic Samplers:** Beyond stochastic samplers, another line of research to achieve an accelerated convergence rate is to sample from the corresponding probability flow ordinary differential equation (PF-ODE). Early work provided polynomial guarantees under rather restrictive Lipschitz conditions Chen et al. (2023e). Later in Chen et al. (2023c), an accelerated convergence rate was first derived with the DPUM sampler by mixing the deterministic predictor steps with stochastic corrector steps. The analysis was performed under the assumption of Lipschitz $\nabla \log q_t$'s and $s_t$'s. Note that this assumption is relatively restrictive and hard to verify in practice. After that, for target distributions having bounded support, Li et al. (2024c) provided the first analysis of a purely deterministic sampler (along with an accelerated deterministic sampler), albeit with a high dimensional dependency. Recently, under strong assumptions on $s_t$'s, Huang et al. (2024a) provided an accelerated rate using the $p$-th order Runge-Kutta time integrator for ODEs for those target distributions

---

[1]Although the technique in Huang et al. (2024a) may be applied to Gaussian mixtures, the authors do not provide explicit dependencies in their paper. Also, Huang et al. (2024a) is posted on arXiv after our first draft.

having bounded support. Specifically, for first-order Runge-Kutta methods, it is assumed that the first two orders of partial derivatives of $s_t$'s are uniformly bounded in space and time, which implies Lipschitz-smoothness of $s_t$ and its derivative along the entire sampling path. Most recently, Li et al. (2024d) obtained a linear convergence rate both in $d$ and $\varepsilon^{-1}$ using PF-ODEs as long as $s_t$'s (and their derivatives) are well estimated. However, it is analyzed only on bounded-support targets. Beyond these works, further acceleration to deterministic samplers is sought in Li et al. (2024a;c) that gives the convergence rate of $\mathcal{O}(\varepsilon^{-1/2})$, which are still performed under bounded-support targets. In comparison, our work substantially broadens the target distributions to include those with unbounded support (yet with finite variance) while achieving an accelerated convergence rate.

## 2 PRELIMINARIES OF DDPM

In this section, we provide the background of the DDPM sampler (Ho et al., 2020).

### 2.1 FORWARD PROCESS

Let $x_0 \in \mathbb{R}^d$ be the initial data, and let $x_t \in \mathbb{R}^d, t \in \{1, \ldots, T\}$ be the latent variables in the diffusion algorithm. Let $Q_0$ be the initial data distribution, and let $Q_t$ be the marginal latent distribution at time $t$ in the forward process, for all $1 \leq t \leq T$. In the forward process, white Gaussian noise is gradually added to the data: $x_t = \sqrt{1 - \beta_t} x_{t-1} + \sqrt{\beta_t} w_t, \forall t \in \{1, \ldots, T\}$, where $w_t \overset{i.i.d.}{\sim} \mathcal{N}(0, I_d)$. Equivalently, this can be expressed as a conditional distribution at each time $t$:

$$Q_{t|t-1}(x_t|x_{t-1}) = \mathcal{N}(x_t; \sqrt{1 - \beta_t} x_{t-1}, \beta_t I_d), \tag{1}$$

which means that under $Q$, $X_0 \to X_1 \to \cdots \to X_T$. Here $\beta_t \in (0, 1)$ captures the "amount" of noise that is injected at time $t$, and $\beta_t$'s are called the *noise schedule*. Define

$$\alpha_t := 1 - \beta_t, \quad \bar{\alpha}_t := \prod_{i=1}^t \alpha_i, \quad 1 \leq t \leq T.$$

An immediate result by accumulating the steps is that

$$Q_{t|0}(x_t|x_0) = \mathcal{N}(x_t; \sqrt{\bar{\alpha}_t} x_0, (1 - \bar{\alpha}_t) I_d), \tag{2}$$

or, written equivalently, $x_t = \sqrt{\bar{\alpha}_t} x_0 + \sqrt{1 - \bar{\alpha}_t} \bar{w}_t, \forall t \in \{1, \ldots, T\}$, where $\bar{w}_t \sim \mathcal{N}(0, I_d)$ denotes the *aggregated* noise at time $t$. Intuitively, for large $T$, since $Q_{T|0} \approx \mathcal{N}(0, I_d)$ (which is independent of $x_0$), it is expected that $Q_T \approx \mathcal{N}(0, I_d)$ when $T$ becomes large, as long as the variance under $Q_0$ is finite. Finally, since the conditional noises are Gaussian, each $Q_t (t \geq 1)$ is absolutely continuous w.r.t the Lebesgue measure. Let the corresponding p.d.f. of each $Q_t$ be $q_t (t \geq 1)$. Similarly define $q_{t,t-1}$, $q_{t|t-1}$, and $q_{t-1|t}$ for $t \geq 1$. In case $Q_0$ is also absolutely continuous w.r.t. the Lebesgue measure, let $q_0$ be the corresponding p.d.f. of $Q_0$.

### 2.2 REGULAR REVERSE PROCESS

The goal of the reverse sampling process is to generate samples approximately from the data distribution $Q_0$. We first draw the latent variable at time $T$ from a Gaussian distribution: $x_T \sim \mathcal{N}(0, I_d) =: P_T$. Then, to achieve effective sampling, each forward step is approximated by a reverse sampling step, in which the *mean* matches the posterior mean of $Q_{t-1|t}$. Define

$$\mu_t(x_t) := \frac{1}{\sqrt{\alpha_t}} \left( x_t + (1 - \alpha_t) \nabla \log q_t(x_t) \right). \tag{3}$$

Here $\nabla \log q_t(x)$ is called the *score* of $q_t$, which can be estimated via a training process called score matching. At each time $t = T, T-1, \ldots, 1$, the *true* regular reverse process is defined as $x_{t-1} = \mu_t(x_t) + \sigma_t z$, where $z \sim \mathcal{N}(0, I_d)$. Two choices of $\sigma_t^2$ are commonly used in practice, where $\sigma_t^2 = 1 - \alpha_t$ or $\sigma_t^2 = \frac{1 - \bar{\alpha}_{t-1}}{1 - \bar{\alpha}_t}(1 - \alpha_t)$, and similar results are reported for these choices (Ho et al., 2020). Let $P_t$ be the marginal distributions of $x_t$ in the true regular reverse process, and let $p_t$ be the corresponding p.d.f. of $P_t$ w.r.t. the Lebesgue measure.

### 2.3 METRICS

In case where $Q$ is absolutely continuous w.r.t. the Lebesgue measure, we are interested in measuring the mismatch between $Q$ and $P$ through the total-variation distance, defined as

$$\mathrm{TV}(Q, P) := \sup_{A \subseteq \mathcal{B}(\mathbb{R}^d)} |Q(A) - P(A)|$$

where $\mathcal{B}(\mathbb{R}^d)$ contains all Borel-measureable sets in $\mathbb{R}^d$. This metric is commonly used in prior theoretical studies (Chen et al., 2023a). From Pinsker's inequality, the total-variation (TV) distance is upper bounded as $\mathrm{TV}(Q, P)^2 \leq \frac{1}{2}\mathrm{KL}(Q\|P)$, where the KL divergence is defined as $\mathrm{KL}(Q\|P) := \int \log \frac{\mathrm{d}Q}{\mathrm{d}P}\mathrm{d}Q \geq 0$. Thus, we control the KL divergence when $Q$ is absolutely continuous w.r.t. $P$.

When $q_0$ does not exist (say, when $Q_0$ has point masses), we use the Wasserstein distance to measure the mismatch at $t = 0$, namely $\mathrm{W}_2(Q_0, Q_1)$, which is a technique commonly adopted (Chen et al., 2023a; Benton et al., 2024a). The Wasserstein-2 distance is defined as $\mathrm{W}_2(Q_0, Q_1) := \sqrt{\min_{\Gamma \in \Pi(Q_0, Q_1)} \int_{\mathbb{R}^d \times \mathbb{R}^d} \|x - y\|^2 \, \mathrm{d}\Gamma(x, y)}$, where $\Pi(Q_0, Q_1)$ is the set of all joint probability measures on $\mathbb{R}^d \times \mathbb{R}^d$ with marginal distributions $Q_0$ and $Q_1$, respectively.

## 3  ACCELERATED DIFFUSION SAMPLER

To generate samples from the data distribution $Q_0$, the idea of DDPM is to design a reverse process in which each reverse sampling step well approximates the corresponding forward step. Below, we propose a new **accelerated** sampler along with a new variance estimator, in which both the conditional *mean and variance* of the reverse process match the corresponding posterior quantities.

### 3.1  ACCELERATED REVERSE PROCESS

At each time $t = T, T - 1, \ldots, 1$, define the true *accelerated* reverse process as $x_{t-1} = \mu_t(x_t) + \Sigma_t^{\frac{1}{2}}(x_t)z$, where $\mu_t$ is defined in (3), $z \sim \mathcal{N}(0, I_d)$, and (cf. Lemma 8)

$$\Sigma_t(x_t) := \tfrac{1-\alpha_t}{\alpha_t}\left(I_d + (1 - \alpha_t)\nabla^2 \log q_t(x_t)\right). \tag{4}$$

Let $P'_t$ be the marginal distributions of $x_t$ in the true accelerated reverse process, and let $p'_t$ be the corresponding p.d.f.. Thus, the transition kernel can be written as $P'_{t-1|t} = \mathcal{N}(x_{t-1}; \mu_t(x_t), \Sigma_t(x_t))$, and we let $P'_T := P_T = \mathcal{N}(0, I_d)$. When $(1 - \alpha_t)$ is vanishing for large $T$, $\Sigma_t(x_t) \succ 0$ for all large $T$'s, and thus the conditional Gaussian process is well-defined.[2] The above accelerated sampler has a close relationship to Ozaki's discretization method to approximate a continuous-time stochastic process (Ozaki, 1992; Shoji, 1998; Stramer & Tweedie, 1999).

In practice, one has no access to either $\nabla \log q_t$ or $\nabla^2 \log q_t$. Thus, their estimates, denoted as $s_t$ and $H_t$, are used. Define the *estimated* accelerated reverse process: $x_{t-1} = \widehat{\mu}_t(x_t) + \widehat{\Sigma}_t^{\frac{1}{2}}(x_t)z$, where

$$\widehat{\mu}_t(x_t) := x_t + (1 - \alpha_t)s_t(x_t), \tag{5}$$

$$\widehat{\Sigma}_t(x_t) := \tfrac{1-\alpha_t}{\alpha_t}\left(I_d + (1 - \alpha_t)H_t(x_t)\right). \tag{6}$$

Here, $s_t$ can be obtained through score-matching (Song & Ermon, 2019). In Section 3.2, we propose an estimator for $\nabla^2 \log q_t$, which we refer to as Hessian matching. Let $\widehat{P}'_t$ be the marginal distributions of $x_t$ in the estimated reverse process with corresponding p.d.f. $\widehat{p}'_t$.

### 3.2  HESSIAN MATCHING ESTIMATOR FOR ACCELERATION

Below we provide a method to obtain $H_t(x)$, which estimates $\nabla^2 \log q_t(x)$. Note that

$$\nabla^2 \log q_t(x) = \tfrac{\nabla^2 q_t(x)}{q_t(x)} - (\nabla \log q_t(x))(\nabla \log q_t(x))^{\mathsf{T}}$$

$$= \left(\tfrac{\nabla^2 q_t(x)}{q_t(x)} + \tfrac{1}{1-\bar{\alpha}_t}I_d\right) - \tfrac{1}{1-\bar{\alpha}_t}I_d - (\nabla \log q_t(x))(\nabla \log q_t(x))^{\mathsf{T}}. \tag{7}$$

Apart from the original score estimate, we require an additional Hessian estimate:

$$v_t(x) := \arg\min_{v_\theta : \mathbb{R}^d \to \mathbb{R}^{d \times d}} \mathbb{E}_{X_t \sim Q_t} \left\| v_\theta(X_t) - \left(\tfrac{\nabla^2 q_t(X_t)}{q_t(X_t)} + \tfrac{1}{1-\bar{\alpha}_t}I_d\right) \right\|_F^2.$$

In order to train for $v_t$, the following lemma provides an analogy to score matching, which we refer to as *Hessian matching*.

---

[2]More rigorously, we can project the matrices $\Sigma_t$ and $\widehat{\Sigma}_t$ onto the space of positive-semi definite (PSD) matrices for those $x_t$'s where either of these two matrices is not PSD. Since the probability of the events containing such bad $x_t$'s decreases to zero asymptotically, all theoretical results in this paper, which are derived in expectation, will not be affected.

**Lemma 1.** *With the forward process in* (1)*, we have*

$$\arg\min_{v_\theta:\mathbb{R}^d \to \mathbb{R}^{d\times d}} \mathbb{E}_{X_t \sim Q_t} \left\| v_\theta(X_t) - \left( \tfrac{\nabla^2 q_t(X_t)}{q_t(X_t)} + \tfrac{1}{1-\bar\alpha_t} I_d \right) \right\|_F^2$$

$$= \arg\min_{v_\theta:\mathbb{R}^d \to \mathbb{R}^{d\times d}} \mathbb{E}_{(X_0,\bar{W}_t)\sim Q_0 \otimes \mathcal{N}(0,I_d)} \left\| v_\theta(\sqrt{\bar\alpha_t}X_0 + \sqrt{1-\bar\alpha_t}\bar{W}_t) - \tfrac{1}{1-\bar\alpha_t}\bar{W}_t\bar{W}_t^\intercal \right\|_F^2.$$

With the Hessian estimate $v_t$ using Lemma 1, from (7), an estimate for $\nabla^2 \log q_t(x)$ is given by

$$H_t(x) = v_t(x) - \tfrac{1}{1-\bar\alpha_t}I_d - s_t(x)s_t^\intercal(x). \tag{8}$$

With the estimator of $H_t$ in (8), the Hessian-based sampler using the $\widehat{\bar\Sigma}_t$ later in (9) is the same as the accelerated stochastic sampler in Li et al. (2024c). Yet, our analysis is applicable when estimation errors exist, whereas in Li et al. (2024c) the estimators are assumed to be perfect for the accelerated sampler. In the literature, several other estimators have been proposed for higher order derivatives of $\log q_t(x)$ (Meng et al., 2021; Lu et al., 2022; Dockhorn et al., 2022). In our paper, we proposed another method, the Hessian matching method, which can guarantee accurate Hessian estimations with extra computation resources. Yet, our analysis can be applied to any estimator for $H_t$ as long as Assumption 3 is satisfied.

# 4 ACCELERATED CONVERGENCE BOUNDS FOR BROADER TARGETS

In this section, we provide convergence guarantees for the accelerated stochastic samplers for general $Q_0$. We will first establish our main result for smooth $Q_0$, and then extend it for more general (possibly non-smooth) $Q_0$. We will also provide a sketch of proof to describe key analysis techniques.

## 4.1 TECHNICAL ASSUMPTIONS FOR ACCELERATED SAMPLER

We first provide the following four technical assumptions for the accelerated sampler.
**Assumption 1** (Finite Second Moment). There exists a constant $M_2 < \infty$ (that does not depend on $d$ and $T$) such that $\mathbb{E}_{X_0 \sim Q_0} \|X_0\|^2 \leq M_2 d$.
**Assumption 2** (Absolute Continuity). $Q_0$ is absolutely continuous w.r.t. the Lebesgue measure, and thus $q_0$ exists. Also, suppose that $q_0$ is analytic [3] and that $q_0(x) > 0$.

The above Assumptions 1 and 2 are commonly adopted in the literature (Chen et al., 2023a;d).
**Assumption 3** (Score and Hessian Estimation Error). The estimates $s_t$'s and $H_t$'s satisfy

$$\tfrac{1}{T}\sum_{t=1}^T \mathbb{E}_{X_t \sim Q_t} \|s_t(X_t) - \nabla \log q_t(X_t)\|^2 \leq \varepsilon^2 = \tilde{O}(T^{-2}),$$

$$\tfrac{1}{T}\sum_{t=1}^T \mathbb{E}_{X_t \sim Q_t} \left\| H_t(X_t) - \nabla^2 \log q_t(X_t) \right\|_F^2 \leq \varepsilon_H^2 = \tilde{O}(T^{-1}).$$

Also, suppose that $H_t$ satisfies $\sup_{\ell \geq 1} \left( \mathbb{E}_{X_t \sim Q_t} \|H_t(X_t)\|^\ell \right)^{1/\ell} = \tilde{O}(1)$.

The above assumption (Assumption 3) describes the estimation error for both the score and Hessian. In particular, compared with regular samplers, the score function needs to be estimated at a higher accuracy in order to achieve acceleration. Such higher accuracy is also required in previous analyses of ODE samplers (e.g., Li et al. (2024a;d)). The regularity condition on $H_t$ can be satisfied, for example, when $\|H_t\|$ is bounded as $\tilde{O}(1)$. As another example, it suffices that $\|H_t(x)\|$ has a polynomial upper bound in $x$ when $Q_t$ is sub-exponential. In Lemma 2 (in Appendix C), we provide sufficient conditions such that the $H_t$ in (8) satisfies Assumption 3.
**Assumption 4** (Regular Partial Derivatives). For all $t \geq 1, \ell \geq 1$, and $\boldsymbol{a} \in [d]^p$ such that $|\boldsymbol{a}| = p \geq 1$,

$$\mathbb{E}_{X_t \sim Q_t} |\partial_{\boldsymbol{a}}^p \log q_t(X_t)|^\ell = O(1), \quad \mathbb{E}_{X_t \sim Q_t} |\partial_{\boldsymbol{a}}^p \log q_{t-1}(\mu_t(X_t))|^\ell = O(1).$$

When $q_0$ does not exist, this is required only for $t \geq 2$.[4]

The above regularity assumption (Assumption 4) on the partial derivatives is needed for our analysis based on Taylor expansion. It is rather soft, and it can be verified on the following two common cases: (1) when $Q_0$ has finite variance, and (2) when $Q_0$ is Gaussian mixture (see Section 5). Case 1 clearly covers a broad set of target distributions of practical interest, such as images, and many theoretical studies of diffusion models have been specially focused on such a distribution (Li et al., 2024a;c). Case 2 has also been well studied for diffusion models (Chen et al., 2024; Gatmiry et al., 2024).

---

[3] Here a function is analytic if its Taylor series converges to the functional value at each point in the domain.
[4] In the Appendix, we have provided the more general Assumption 5 under which Theorem 1 would hold.

## 4.2 ACCELERATED CONVERGENCE BOUNDS

We first define a new noise schedule as follows, which will be useful for acceleration.

**Definition 1** (Noise Schedule for Acceleration). For large $T$'s, the step-size $\alpha_t$ satisfies that

$$1 - \alpha_t \lesssim \tfrac{\log T}{T}, \ \forall t \in \{1, \ldots, T\}, \quad \bar{\alpha}_T = \prod_{t=1}^{T} \alpha_t = o\left(T^{-2}\right).$$

When $q_0$ does not exist, the upper bound on $1 - \alpha_t$ is only required for $t \geq 2$.

In Definition 1, the upper bound on $1 - \alpha_t$ requires that $\alpha_t$ is large enough to control the reverse-step error, while the upper bound on $\bar{\alpha}_T$ requires that $\alpha_t$ is small enough to control the initialization error. An example of $\alpha_t$ that satisfies Definition 1 is the constant step-size: $1 - \alpha_t \equiv \tfrac{c \log T}{T}, \ \forall t \geq 1$ with $c > 2$. Then, $\bar{\alpha}_T = \left(1 - \tfrac{c \log T}{T}\right)^T = \exp\left(T \log\left(1 - \tfrac{c \log T}{T}\right)\right) = O\left(e^{T \frac{-c \log T}{T}}\right) = o\left(T^{-2}\right).$ Thus, such $\alpha_t$ satisfies Definition 1.

The following theorem provides the *first* convergence result for accelerated diffusion samplers for general smooth target distributions that have *finite second moment* (along with some mild regularity conditions). The complete proof is given in Appendix D.

**Theorem 1** (Accelerated Sampler for Smooth $Q_0$). *Under Assumptions 1 to 4, with the $\alpha_t$ satisfying Definition 1, we have*

$$\begin{aligned}
\mathrm{KL}(Q_0 \| \widehat{P}_0') \lesssim &(\log T)\varepsilon^2 + \tfrac{\log^2 T}{T}\varepsilon_H^2 \\
&+ \textstyle\sum_{t=1}^{T}(1 - \alpha_t)^3 \mathbb{E}_{X_t \sim Q_t} \sum_{i,j,k=1}^{d} \partial_{ijk}^3 \log q_{t-1}(\mu_t(X_t)) \partial_{ijk}^3 \log q_t(X_t).
\end{aligned}$$

Theorem 1 characterizes the convergence in terms of KL divergence (and thus TV distance) for smooth (possibly unbounded) $Q_0$. The bound in Theorem 1 will be further instantiated with explicit dependency on system parameters for example distributions $Q_0$ in Section 5. To further explain the upper bound in Theorem 1, the first two terms arise from the score and Hessian estimation error, and the last term captures the errors accumulated during the reverse steps over $t = T, \ldots, 1$, which can be further bounded by $\tilde{O}(T^{-2})$ under Assumption 4 (cf. (52)). Thus, when $\varepsilon_H^2$ satisfies Assumption 3, the upper bound in Theorem 1 can be more explicitly characterized w.r.t. $T$ as $\mathrm{KL}(Q_0 \| \widehat{P}_0) \lesssim \tilde{O}(T^{-2}) + (\log T)\varepsilon^2$ (where the dependency on $d$ will be explicitly characterized for specific distributions in Section 5). Thus, in order to achieve $\mathcal{O}(\varepsilon^2)$ error in KL divergence, the number of steps required is $\mathcal{O}(\varepsilon^{-1})$. This improves the dependency of the convergence rate on $\varepsilon$ of the regular sampler by a factor of $\mathcal{O}(\varepsilon^{-1})$.

We next extend Theorem 1 for smooth $Q_0$ to general $Q_0$ that can be possibly non-smooth and hence the density function $q_0$ does not exist. Such distributions occur often in practice; for example, when $Q_0$ has a discrete support such as for images, or when $Q_0$ is supported on a low-dimensional manifold. For non-smooth $Q_0$, its one-step perturbation $Q_1$ does have a p.d.f. $q_1$, which is further analytic (Lemma 6). This enables us to apply Theorem 1 on $Q_1$ to obtain the following convergence bound. Also, we use the Wasserstein distance to measure the perturbation between $Q_0$ and $Q_1$ (Chen et al., 2023d;a; Lee et al., 2023).

**Corollary 1** (General (possibly non-smooth) $Q_0$). *Under Assumptions 1, 3 and 4, if the noise schedule satisfies Definition 1 at $t \geq 2$, the distribution $\widehat{P}_1'$ satisfies*

$$\begin{aligned}
\mathrm{KL}(Q_1 \| \widehat{P}_1') \lesssim &(\log T)\varepsilon^2 + \tfrac{\log^2 T}{T}\varepsilon_H^2 \\
&+ \textstyle\sum_{t=2}^{T}(1 - \alpha_t)^3 \mathbb{E}_{X_t \sim Q_t} \sum_{i,j,k=1}^{d} \partial_{ijk}^3 \log q_{t-1}(\mu_t(X_t)) \partial_{ijk}^3 \log q_t(X_t),
\end{aligned}$$

*where $Q_1$ is such that $\mathrm{W}_2(Q_0, Q_1)^2 \lesssim (1 - \alpha_1)d$.*

In particular, Corollary 1 applies to any general target distribution when the second moment is finite.

## 4.3 PROOF SKETCH OF THEOREM 1

We next provide a proof sketch of Theorem 1 to describe the idea of our analysis approach. The full proof is provided in Appendix D. Our approach is very different from previous SDE-type approaches, which invoke Fokker-Planck equation to express the evolution of p.d.f. and use Girsanov's Theorem to bound the divergence, both along the *continuous* diffusion path. In comparison, we develop a novel Bayesian approach based on tilting factor representation and Tweedie's formula to handle power terms, which is applicable to a much wider class of target distributions, including those having

infinite support. In particular, compared with Li et al. (2024a;c;d), our approach does not assume that the target distribution has finite support.

To begin, we decompose the total error as

$$\text{KL}(Q_0||\widehat{P}_0') \leq \underbrace{\mathbb{E}_{X_T \sim Q_T}\left[\log \frac{q_T(X_T)}{p_T'(X_T)}\right]}_{\text{initialization error}}$$

$$+ \underbrace{\sum_{t=1}^{T} \mathbb{E}_{X_t, X_{t-1} \sim Q_{t,t-1}}\left[\log \frac{p_{t-1|t}'(X_{t-1}|X_t)}{\widehat{p}_{t-1|t}'(X_{t-1}|X_t)}\right]}_{\text{estimation error}} + \underbrace{\sum_{t=1}^{T} \mathbb{E}_{X_t, X_{t-1} \sim Q_{t,t-1}}\left[\log \frac{q_{t-1|t}(X_{t-1}|X_t)}{p_{t-1|t}'(X_{t-1}|X_t)}\right]}_{\text{reverse-step error}}.$$

The initialization error can be bounded easily (Lemma 3). Below we focus on the remaining two terms in five steps.

**Step 1: Bounding estimation error (Lemma 4).** At each time $t = 1, \ldots, T$, rather than upper-bounding via typical sets as in Li et al. (2024c), we directly evaluate the expected value of $\log(p_{t-1|t}'(x_{t-1}|x_t)/\widehat{p}_{t-1|t}'(x_{t-1}|x_t))$. This is straightforward since $P_{t-1|t}'$ and $\widehat{P}_{t-1|t}'$ are Gaussian. We then use Taylor expansion for the $\log \det(\cdot)$ function and the matrix inverse to identify the dominant-order terms under the mismatched variance.

**Step 2: Tilting factor expression of log-likelihood ratio (Lemmas 5 and 6 and Equation (20)).** With Bayes' rule, we show that $q_{t-1|t}$ is an exponentially tilted form of $p_{t-1|t}'$ with tilting factor:

$$\zeta_{t,t-1}' = (\nabla \log q_{t-1}(\mu_t) - \sqrt{\alpha_t} \nabla \log q_t(x_t))^\mathsf{T} (x_{t-1} - \mu_t)$$

$$+ \frac{1}{2}(x_{t-1} - \mu_t)^\mathsf{T} \left(\nabla^2 \log q_{t-1}(\mu_t) - \frac{\alpha_t}{1-\alpha_t} B_t(x_t)\right)(x_{t-1} - \mu_t) + \sum_{p=3}^{\infty} T_p(\log q_{t-1}, x_{t-1}, \mu_t).$$

where $B_t(x_t)$ describes the correction due to the modified variance for acceleration (see (14)), and $T_p(f, x, \mu)$ is the $p$-th order Taylor power term of function $f$ around $x = \mu$. With this tilting factor, we can upper-bound the reverse-step error as, for each fixed $x_t$,

$$\mathbb{E}_{X_{t-1}, X_t \sim Q_{t-1,t}}\left[\log \frac{q_{t-1|t}(X_{t-1}|x_t)}{p_{t-1|t}'(X_{t-1}|x_t)}\right] \leq \mathbb{E}_{X_t, X_{t-1} \sim Q_{t,t-1}}[\zeta_{t,t-1}'] - \mathbb{E}_{X_t \sim Q_t, X_{t-1} \sim P_{t-1|t}'}[\zeta_{t,t-1}'].$$

For regular DDPMs, there is no control for the variance of the reverse sampling process, and thus $B_t(x_t) \equiv 0$. In this case, the dominating rate is determined by the expected values of $T_2$. With the variance correction in our accelerated sampler, the corresponding $B_t(x_t)$ enables us to cancel out the second-order Taylor term (see Lemma 11). As a result, the rate-determining term becomes the expected values of $T_3$, which decays faster. Thus, the acceleration is achieved.

**Step 3: Explicit expression for** $\mathbb{E}_{X_t \sim Q_t, X_{t-1} \sim P_{t-1|t}'}[\zeta_{t,t-1}']$ **(Lemma 7).** Given the Taylor expansion of $\zeta_{t,t-1}'$, this step can be reduced to calculating the expected values of the power terms, which are the Gaussian centralized moments. They are calculated using the classical Isserlis's Theorem.

**Step 4: Explicit expression for** $\mathbb{E}_{X_t, X_{t-1} \sim Q_{t,t-1}}[\zeta_{t,t-1}']$ **(Lemmas 8 to 10).** While $Q_{t|t-1}$ is Gaussian, $Q_{t-1|t}$ is not Gaussian in general, rendering the calculation of all moments non-trivial. To calculate posterior moments, we extend Tweedie's formula (Efron, 2011) in a non-trivial way. Whereas the original Tweedie's formula provides an explicit expression for the posterior mean for Gaussian perturbed observations, we explicitly calculate the first six centralized posterior moments and provide the asymptotic order of all higher-order moments, drawing techniques from combinatorics. The results also justify the expressions of $\mu_t$ and $\Sigma_t$ in (3) and (4).

**Step 5: Bounding reverse-step error (Lemma 11)** In order to employ the moment results for Taylor expansion, we guarantee that it is valid to change the limit (in the Taylor expansion) and the expectation operator. Finally, substituting the calculated moments into $\mathbb{E}_{X_t, X_{t-1} \sim Q_{t,t-1}}[\zeta_{t,t-1}'] - \mathbb{E}_{X_t \sim Q_t, X_{t-1} \sim P_{t-1|t}'}[\zeta_{t,t-1}']$ and noting that higher-order partial derivatives do not affect the rate (by Assumption 4), we can determine the dominating term and obtain the desirable result.

## 5 EXAMPLE $Q_0$'S: ACCELERATED CONVERGENCE RATE WITH EXPLICIT PARAMETER DEPENDENCY

Now, we specialize Theorem 1 and Corollary 1 to several interesting distribution classes, for which convergence bounds with explicit dependency on system parameters can be derived. The key is to locate the dependency in the dominating terms in the reverse-step error.

## 5.1 GAUSSIAN MIXTURE $Q_0$

We first investigate the case where $Q_0$ is Gaussian mixture. This is a rich class of distributions with strong approximation power (Bacharoglou, 2010; Diakonikolas et al., 2017). The following theorem establishes the first accelerated convergence result with explicit dimensional dependencies for such a distribution class.

**Theorem 2** (Accelerated Sampler for Gaussian Mixture $Q_0$). *Suppose that $Q_0$ is Gaussian mixture, whose p.d.f. is given by $q_0(x_0) = \sum_{n=1}^N \pi_n q_{0,n}(x_0)$, where $q_{0,n}$ is the p.d.f. of $\mathcal{N}(\mu_{0,n}, \Sigma_{0,n})$ and $\pi_n \in [0,1]$ is the mixing coefficient where $\sum_{n=1}^N \pi_n = 1$. Under Assumption 3, if the $\alpha_t$ satisfies Definition 1, we have*

$$\mathrm{KL}(Q_0||\widehat{P}_0') \lesssim \frac{d^3 \min\{d,N\}^3 \log^3 T}{T^2} + (\log T)\varepsilon^2 + \frac{\log^2 T}{T}\varepsilon_H^2.$$

Therefore, for any Gaussian mixture target $Q_0$ with $N \leq d$, it takes the accelerated algorithm $\mathcal{O}\left(d^{1.5} N^{1.5}/\varepsilon\right)$ steps to reach convergence under accurate score and Hessian estimation. This is the first result for accelerated DDPM samplers to achieve an accelerated convergence rate for Gaussian mixture targets under score and Hessian estimation error. Compared with the results for regular samplers, the number of convergence steps improves by a factor of $\mathcal{O}(\varepsilon^{-1})$.

The proof of Theorem 2 is non-trivial because in order to show that Assumption 4 holds for Gaussian mixture distributions with any $\alpha_t$ according to Definition 1, it is generally difficult to evaluate and provide an upper bound for *all orders* of partial derivatives of the logarithm of a mixture density. To this end, we employ the multivariate Faá di Bruno's formula (Constantine & Savits, 1996) to develop an explicit bound (Lemmas 13 and 14).

Below we numerically evaluate the performance of our Hessian-accelerated DDPM when $Q_0$ is Gaussian mixture. The original accelerator requires calculating the square-root matrix of $\widehat{\Sigma}_t$ (see (4)), which might be computational burdensome. Below, we propose an approximated Hessian-based accelerated sampler, where $\widehat{\mu}_t$ is still defined in (5) and $\widehat{\Sigma}_t$ is replaced by $\widehat{\widetilde{\Sigma}}_t(x_t)$ where

$$\tilde{\Sigma}_t(x_t) := \frac{1-\alpha_t}{\alpha_t}\left(I_d + \frac{1-\alpha_t}{2}\nabla \log q_t(x_t)\right)^2, \quad \widehat{\widetilde{\Sigma}}_t(x_t) := \frac{1-\alpha_t}{\alpha_t}\left(I_d + \frac{1-\alpha_t}{2}H_t(x_t)\right)^2. \quad (9)$$

With a similar tilting-factor analysis as in Theorem 1, we can verify that the approximated sampler still achieves an accelerated convergence rate (see Corollaries 2 and 3 and Remark 3).

In Figure 1, we compare the following four accelerated samplers: (1) the regular DDPM sampler (in blue); (2) our Hessian-accelerated sampler (in red); (3) the accelerated stochastic sampler in Li et al. (2024a) (in cyan); and (4) the deterministic sampler using PF-ODE, which is analyzed in Li et al. (2024c;d); Huang et al. (2024a). Here $N = 4$ and $d = 4$. The performance is averaged over 30 different trials. In a single trial, 200000 samples are used to estimate the KL divergence. The $\alpha_t$ in (10) is used with $c = 4$ and $\delta = 0.001$. From the comparison, it is observed that our Hessian-based sampler achieves the best convergence (at similar computation levels) in non-asymptotic regimes.

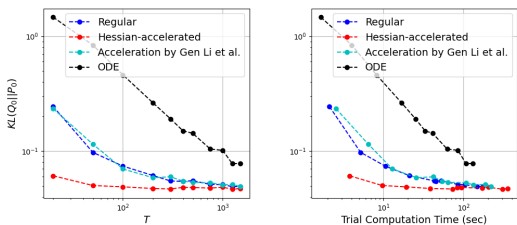

Figure 1: Comparison of different accelerated samplers for Gaussian mixture $Q_0$'s. The $x$-axes are the number of steps (left) and the computation time of a trial (right), respectively.

## 5.2 FINITE VARIANCE $Q_0$ WITH EARLY-STOPPING

Next, we specialize Corollary 1 to a special noise schedule, first proposed in Li et al. (2024c):

$$1 - \alpha_t = \frac{c \log T}{T} \min\left\{\delta\left(1 + \frac{c \log T}{T}\right)^t, 1\right\}, \quad \forall 2 \leq t \leq T, \quad (10)$$

and $1 - \alpha_1 = \delta$. Here $c$ and $\delta$ satisfy that $c > 2$ and $\delta e^c > 1$. Intuitively, $\delta$ characterizes the amount of perturbation between $Q_1$ and $Q_0$ (Lemma 12). Note that any noise schedule satisfying the above condition also satisfies Definition 1 at $t \geq 2$ (see (49)), and hence Corollary 1 still holds here.

**Theorem 3** (Accelerated Sampler for $Q_0$ with Finite Variance). *Under Assumptions 1 and 3, using the $\alpha_t$ defined in* (10) *with $c > 2$ and $c \asymp \log(1/\delta)$, we have*

$$\mathrm{KL}(Q_1 || \widehat{P}'_1) \lesssim \frac{d^3 \log^3(1/\delta) \log^3 T}{T^2} + (\log T)\varepsilon^2 + \frac{\log^2 T}{T}\varepsilon_H^2,$$

*where $Q_1$ is such that $\mathrm{W}_2(Q_0, Q_1)^2 \lesssim \delta d$.*

Theorem 3 indicates that for any $Q_0$ having *finite variance*, it takes the accelerated algorithm $\mathcal{O}\left(d^{1.5} \log^{1.5}(1/\delta)/\varepsilon\right)$ steps to approximate an early-stopped data distribution $Q_1$ within $\mathcal{O}(\varepsilon^2)$ error in KL divergence (or $\mathcal{O}(\varepsilon)$ in TV distance). For early-stopped procedures, this theorem significantly relaxes the previous assumption on the target distribution that requires $Q_0$ to have bounded support (Li et al., 2024a;c; Huang et al., 2024a; Li et al., 2024d). Compared to previous accelerated diffusion samplers for bounded-support targets (Li et al., 2024a;c), our number of convergence steps to achieve $\varepsilon$-TV distance has improved by a factor of $\mathcal{O}(d^{1.5})$.

The proof of Theorem 3 involves the following novel elements. (i) Verifying Assumption 4 requires evaluating and providing an upper bound for *all orders* of partial derivatives of the logarithm of a *continuous* mixture density. Differently from the case of Gaussian (discrete) mixture, here we can only have an upper bound in expectation (i.e., in $\mathcal{L}^p(Q_t)$) (Lemma 15). (ii) The second half of Assumption 4 requires an upper bound for the one-step perturbed score, which can be shown using the change-of-variable formula and the data processing inequality for large $T$ (Lemmas 16 and 17).

### 5.3 $Q_0$ WITH LIPSCHITZ HESSIAN LOG-DENSITY

With the $\alpha_t$ in (10), we derive a convergence result when only the log-density of $Q_0$ is smooth.

**Theorem 4** (Accelerated Sampler for Smooth Hessian Log-Density). *Suppose that $\nabla^2 \log q_0(x)$ is 2-norm $M$-Lipschitz. This means that $\exists M > 0$ such that*

$$\left\|\nabla^2 \log q_0(x) - \nabla^2 \log q_0(y)\right\| \leq M \left\|x - y\right\|, \quad \forall x, y \in \mathbb{R}^d.$$

*Then, under Assumptions 1 and 3, using the $\alpha_t$ in* (10) *with $\delta = 1/(M^{\frac{2}{3}} T^{\frac{3}{2}})$ and $c \geq \log(M^{\frac{2}{3}} T^{\frac{3}{2}})$, we have*

$$\mathrm{KL}(Q_0 || \widehat{P}'_0) \lesssim \frac{d^3(\log^3 M + \log^3 T) \log^3 T}{T^2} + (\log T)\varepsilon^2 + \frac{\log^2 T}{T}\varepsilon_H^2.$$

We also provide an accelerated convergence result with linear $d$ dependency when all the $\nabla^2 \log q_t(x)$ ($t \geq 0$) are 2-norm $M$-Lipschitz (see Theorem 5 in Appendix G.3).

Theorem 4 provides us with the *first* accelerated DDPM result with only a smoothness constraint on $\log q_0$, under the score and Hessian estimation error. In words, in order to reach $\mathcal{O}(\varepsilon)$ TV-distance when $\varepsilon_H^2/T \lesssim \varepsilon^2$, the number of steps needed under Lipschitz-Hessian $Q_0$'s is $\mathcal{O}(d^{1.5} \log^{1.5} M/\varepsilon)$. This is different from Chen et al. (2023c); Huang et al. (2024a;b) in which some smoothness condition is imposed on all $\nabla \log q_t$'s (or $s_t$'s or both). Compared with Theorem 3, this upper bound in Theorem 4 is directly over $\mathrm{KL}(Q_0 || \widehat{P}'_0)$ instead of for some early-stopped distribution. Our results provide new contributions that complement existing studies by exploring different assumptions of distributions, which enriches the existing set of distributions studied in the literature.

Our analysis is significantly different from that in (Chen et al., 2023a, Theorem 5). There, the Poincaré inequality is key to guarantee that the Lipschitz smoothness in $\nabla \log q_0$ is preserved when $\delta$ is small, but this inequality may not hold in our case with smoothness only in $\nabla^2 \log q_0$. Instead, with smooth $\nabla^2 \log q_0$, we expand the tilting factor only to its third-order Taylor polynomial and directly provide an upper bound with techniques used in proving Theorems 3 and 5.

## 6 CONCLUSION

In this paper, we have provided accelerated convergence guarantees for a much larger set of target distributions than in prior literature, including both smooth $Q_0$ and general $Q_0$ with early-stopping. The accelerated rates are achieved with a new accelerated Hessian-based DDPM sampler using a novel analysis technique. One future direction is to further shrink the $d$ dependency for general $Q_0$. It is also interesting to investigate other acceleration schemes to further improve diffusion samplers.

ACKNOWLEDGMENTS

This work has been supported in part by the U.S. National Science Foundation under the grants: DMS-2134145, CCF-1900145, NSF AI Institute (AI-EDGE) 2112471, CNS-2312836, CNS-2225561, ONR grant N000142412729, and was sponsored by the Army Research Laboratory under Cooperative Agreement Number W911NF-23-2-0225. The views and conclusions contained in this document are those of the authors and should not be interpreted as representing the official policies, either expressed or implied, of the Army Research Laboratory or the U.S. Government. The U.S. Government is authorized to reproduce and distribute reprints for Government purposes notwithstanding any copyright notation herein.

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

# Appendix

## A   RELATED WORKS

**Theory on Regular DDPM Samplers:** Many works have explored the performance guarantees of regular DDPM models. Specifically, a number of studies perform analyses under the $L^\infty$ score estimation error (De Bortoli et al., 2021; De Bortoli, 2022). Later, under $L^2$ score estimation error, Lee et al. (2022) developed polynomial[5] bounds for distributions that have Lipschitz scores and satisfy log-Sobolev inequality. Soon after, Chen et al. (2023d); Lee et al. (2023) concurrently developed polynomial bounds for those smooth distributions having Lipschitz scores and those non-smooth distributions having bounded support using early stopping. Later, Chen et al. (2023a) improved the number of steps for those target distributions with finite second moment. Recently, the convergence result was further improved to linear dimensional dependency using stochastic localization (Benton et al., 2024a). In Conforti et al. (2023), by transforming the original process to the relative-score process, it is shown that linear dimensional dependency can also be achieved for those target distributions having finite relative Fisher information against a Gaussian distribution. In all the works above, the analysis technique is to discretize some continuous-time diffusion process to use SDE-type analyses. In Li et al. (2024c), by carefully design a typical set, polynomial-time guarantees are obtained directly for the discrete-time samplers under the $L^2$ estimation error for target distributions having bounded support. Other than the works above, Pedrotti et al. (2023) analyzed a different sampling scheme (e.g., predictor-corrector), and Bruno et al. (2023); Gao et al. (2023); Gao & Zhu (2024) analyzed sampling errors using a different error measure (the Wasserstein-2 distance).

**Theory on Score Estimation:** In order to achieve an end-to-end analysis, several works developed sample complexity bounds to achieve the $L^2$ score estimation error for a variety of distributions. To name a few, this includes results for those having bounded support (Oko et al., 2023), Gaussian mixture (Shah et al., 2023; Gatmiry et al., 2024; Chen et al., 2024), certain families of sub-Gaussian distributions (Cole & Lu, 2024; Zhang et al., 2024), high-dimensional graphical models (Mei & Wu, 2023), and those supported on a low-dimensional linear subspace (Chen et al., 2023b). More recently, Li et al. (2024e) considered the generalizability of the continuous-time diffusion models, and Wibisono et al. (2024) proposed a regularized score estimator that attains the minimax rate of estimating the scores.

**Other Theoretical Works:** Other than the works listed above and in Section 1.2, Gao & Zhu (2024) studied the ODE convergence for strongly-concave target distributions under Wasserstein-2 error. Cao et al. (2023) compared the performance of SDE and PF-ODE and investigated conditions where one might outperform the other. Besides PF-ODE, Cheng et al. (2023); Benton et al. (2024b); Jiao et al. (2024); Gao et al. (2024) provided guarantees for the closely-related flow-matching model, which learns a deterministic coupling between any two distributions. Chang et al. (2024) proposed a novel ODE for sampling from a conditional distribution. Lyu et al. (2024); Li et al. (2024b) provided convergence guarantees for the more recent consistency models (Song et al., 2023).

**Relationship to GENIE (Dockhorn et al., 2022):** To obtain higher-order scores, another method is to use automatic differentiation, as in GENIE (Dockhorn et al., 2022). There, higher-order score functions are used to accelerate the diffusion sampling process empirically. In particular, Dockhorn et al. (2022) shows that GENIE achieves better empirical performance than deterministic samplers such as DDIM (Song et al., 2021). Our paper theoretically justifies the accelerated empirical performance of Dockhorn et al. (2022) in the regime when the Hessian of $\log q_t$ is well-estimated.

---

[5]By "polynomial" we mean that the number of steps has polynomial dependency on the score estimation error, along with other parameters.

## B  FULL LIST OF NOTATIONS

For any two functions $f(d, \delta, T)$ and $g(d, \delta, T)$, we write $f(d, \delta, T) \lesssim g(d, \delta, T)$ (resp. $f(d, \delta, T) \gtrsim g(d, \delta, T)$) for some universal constant (not depending on $\delta$, $d$ or $T$) $L < \infty$ (resp. $L > 0$) if $\limsup_{T \to \infty} |f(d, \delta, T)/ g(d, \delta, T)| \leq L$ (resp. $\liminf_{T \to \infty} |f(d, \delta, T)/g(d, \delta, T)| \geq L$). We write $f(d, \delta, T) \asymp g(d, \delta, T)$ when both $f(d, \delta, T) \lesssim g(d, \delta, T)$ and $f(d, \delta, T) \gtrsim g(d, \delta, T)$ hold. Note that the dependency on $\delta$ and $d$ is retained with $\lesssim, \gtrsim, \asymp$. We write $f(d, \delta, T) = O(g(T))$ (resp. $f(d, \delta, T) = \Omega(g(T))$) if $f(d, \delta, T) \lesssim L(d, \delta)g(T)$ (resp. $f(d, \delta, T) \gtrsim L(d, \delta)g(T)$) holds for some $L(d, \delta)$ (possibly depending on $\delta$ and $d$). We write $f(d, \delta, T) = o(g(T))$ if $\limsup_{T \to \infty} |f(d, \delta, T) /g(T)| = 0$. We write $f(d, \delta, T) = \tilde{O}(g(T))$ if $f(d, \delta, T) = O(g(T)(\log g(T))^k)$ for some constant $k$. Note that the big-$O$ notation omits the dependency on $\delta$ and $d$. In the asymptotic when $\varepsilon^{-1} \to \infty$, we write $f(d, \varepsilon^{-1}) = \mathcal{O}(g(d, \varepsilon^{-1}))$ if $f(d, \delta, \varepsilon^{-1}) \lesssim g(d, \delta, \varepsilon^{-1})(\log g(\varepsilon^{-1}))^k$ for some constant $k$. Unless otherwise specified, we write $x^i (1 \leq i \leq d)$ as the $i$-th element of a vector $x \in \mathbb{R}^d$ and $[A]^{ij}$ as the $(i, j)$-th element of a matrix $A$. For a function $f(x) : \mathbb{R}^d \to \mathbb{R}$, we write $\partial_i f(z)$ as a shorthand for $\frac{\partial}{\partial x^i} f(x)\big|_{x=z}$, and similarly for higher moments. For matrices $A$, $B$, $\text{Tr}(A)$ is the trace of $A$, and $A \preceq B$ means that $B - A$ is positive semi-definite. For a positive integer $n$, $[n] := \{1, \ldots, n\}$.

## C  PROOFS OF LEMMAS 1 AND 2

In this section, we provide lemmas and proofs related to Hessian estimation.

### C.1  PROOF OF LEMMA 1

The idea is similar to score matching. Define $v'_\theta(x) := v_\theta(x) - \frac{1}{1-\bar{\alpha}_t} I_d$. For each $i, j \in [d]$,

$$\mathbb{E}_{X_t \sim Q_t} \left( v_\theta^{ij}(X_t) - \left( \frac{\partial_{ij}^2 q_t(X_t)}{q_t(X_t)} + \frac{\mathbb{1}\{i = j\}}{1 - \bar{\alpha}_t} \right) \right)^2$$

$$= \mathbb{E}_{X_t \sim Q_t} \left( [v'_\theta(X_t)]^{ij} - \frac{\partial_{ij}^2 q_t(X_t)}{q_t(X_t)} \right)^2$$

$$= \mathbb{E}_{X_t \sim Q_t} \left( [v'_\theta(X_t)]^{ij} \right)^2 - 2\mathbb{E}_{X_t \sim Q_t} \left[ [v'_\theta(X_t)]^{ij} \frac{\partial_{ij}^2 q_t(X_t)}{q_t(X_t)} \right] + \text{const}$$

$$= \mathbb{E}_{X_t \sim Q_t} \left( [v'_\theta(X_t)]^{ij} \right)^2 - 2 \int [v'_\theta(x_t)]^{ij} \partial_{ij}^2 q_t(x_t) \mathrm{d}x_t + \text{const}$$

where const denotes terms that are independent of $\theta$, and

$$\int [v'_\theta(x_t)]^{ij} \partial_{ij}^2 q_t(x_t) \mathrm{d}x_t$$

$$= \int [v'_\theta(x_t)]^{ij} \int \partial_{ij}^2 q_{t|0}(x_t|x_0) \mathrm{d}Q_0(x_0) \mathrm{d}x_t$$

$$= \int \int q_{t|0}(x_t|x_0) [v'_\theta(x_t)]^{ij} \frac{\partial_{ij}^2 q_{t|0}(x_t|x_0)}{q_{t|0}(x_t|x_0)} \mathrm{d}Q_0(x_0) \mathrm{d}x_t$$

$$\overset{(i)}{=} \int \int q_{t|0}(x_t|x_0) [v'_\theta(x_t)]^{ij} \left( \partial_{ij}^2 \log q_{t|0}(x_t|x_0) + \partial_i \log q_{t|0}(x_t|x_0) \partial_j \log q_{t|0}(x_t|x_0) \right) \mathrm{d}Q_0(x_0) \mathrm{d}x_t$$

$$= \int \int q_{t|0}(x_t|x_0) [v'_\theta(x_t)]^{ij} \left( -\frac{\mathbb{1}\{i = j\}}{1 - \bar{\alpha}_t} + \frac{x_t^i - \sqrt{\bar{\alpha}_t} x_0^i}{1 - \bar{\alpha}_t} \cdot \frac{x_t^j - \sqrt{\bar{\alpha}_t} x_0^j}{1 - \bar{\alpha}_t} \right) \mathrm{d}Q_0(x_0) \mathrm{d}x_t$$

$$\overset{(ii)}{=} \mathbb{E}_{\substack{(X_0, \bar{W}_t) \sim Q_0 \otimes \mathcal{N}(0, I_d) \\ X_t = \sqrt{\bar{\alpha}_t} X_0 + \sqrt{1-\bar{\alpha}_t} \bar{W}_t}} \left[ [v'_\theta(X_t)]^{ij} \left( -\frac{\mathbb{1}\{i = j\}}{1 - \bar{\alpha}_t} + \frac{1}{1 - \bar{\alpha}_t} \bar{W}_t^i \bar{W}_t^j \right) \right]$$

where $(i)$ follows because for any function $f(x)$ we have $\partial_{ij}^2 \log f(x) = \frac{\partial_{ij}^2 f(x)}{f(x)} - (\partial_i \log f(x))(\partial_j \log f(x))$, and $(ii)$ follows because $x_t = \sqrt{\bar{\alpha}_t} x_0 + \sqrt{1 - \bar{\alpha}_t} \bar{w}_t$. Therefore,

$$\mathbb{E}_{X_t \sim Q_t} \left( v_\theta^{ij}(X_t) - \left( \frac{\partial_{ij}^2 q_t(X_t)}{q_t(X_t)} - \frac{\mathbb{1}\{i = j\}}{1 - \bar{\alpha}_t} \right) \right)^2$$

$$= \mathop{\mathbb{E}}_{\substack{(X_0, \bar{W}_t) \sim Q_0 \otimes \mathcal{N}(0, I_d) \\ X_t = \sqrt{\bar{\alpha}_t} X_0 + \sqrt{1 - \bar{\alpha}_t} \bar{W}_t}} \left( [v_\theta'(X_t)]^{ij} - \left( -\frac{\mathbb{1}\{i = j\}}{1 - \bar{\alpha}_t} + \frac{1}{1 - \bar{\alpha}_t} \bar{W}_t^i \bar{W}_t^j \right) \right)^2 + \text{const}$$

$$= \mathop{\mathbb{E}}_{\substack{(X_0, \bar{W}_t) \sim Q_0 \otimes \mathcal{N}(0, I_d) \\ X_t = \sqrt{\bar{\alpha}_t} X_0 + \sqrt{1 - \bar{\alpha}_t} \bar{W}_t}} \left( [v_\theta(X_t)]^{ij} - \frac{1}{1 - \bar{\alpha}_t} \bar{W}_t^i \bar{W}_t^j \right)^2 + \text{const}$$

and the result follows immediately after we sum up over $i, j \in [d]$.

### C.2 LEMMA 2 AND ITS PROOF

The following lemma provides sufficient conditions such that the $H_t$ in (8) satisfies Assumption 3.

**Lemma 2.** *Under Assumption 5, with the $\alpha_t$ defined in Definition 1, suppose that $v_t$ and $s_t$ satisfy, as $T \to \infty$,*

$$\frac{1}{T} \sum_{t=1}^{T} \mathbb{E}_{X_t \sim Q_t} \left\| v_t(X_t) - \left( \frac{\nabla^2 q_t(X_t)}{q_t(X_t)} + \frac{1}{1 - \bar{\alpha}_t} I_d \right) \right\|_F^2 = \tilde{O}(T^{-1}), \tag{11}$$

$$\max_{1 \le t \le T} (1 - \alpha_t)^{-2} \sqrt{\mathbb{E}_{X_t \sim Q_t} \| s_t(X_t) - \nabla \log q_t(X_t) \|^4} = \tilde{O}(1). \tag{12}$$

*Also suppose that the $H_t$ defined in (8) satisfies $\sup_{\ell \ge 1} \left( \mathbb{E}_{X_t \sim Q_t} \| H_t(X_t) \|^\ell \right)^{1/\ell} = \tilde{O}(1)$. Then, the $H_t$ and the $s_t$ from score matching (Song & Ermon, 2019) satisfy Assumption 3.*

*Proof of Lemma 2.* The condition on the score estimation error in Assumption 3 is immediately satisfied using Jensen's inequality. We next focus on the condition on the Hessian estimation. Recall that

$$H_t(x) = v_t(x) - \frac{1}{1 - \bar{\alpha}_t} I_d - s_t(x) s_t^\intercal(x).$$

The goal is to show that $H_t$ is close to $\nabla^2 \log q_t$ (i.e., the second relationship in Assumption 3). Given that $\nabla^2 \log q_t(x) = \frac{\nabla^2 q_t(x)}{q_t(x)} - (\nabla \log q_t(x))(\nabla \log q_t(x))^\intercal$, the key is to control the error incurred by $s_t(x) s_t(x)^\intercal$, which is

$$\mathbb{E}_{X_t \sim Q_t} \sum_{i,j=1}^{d} \left( s_t^i(X_t) s_t^j(X_t) - [\nabla \log q_t(X_t)]^i [\nabla \log q_t(X_t)]^j \right)^2$$

$$= \mathbb{E}_{X_t \sim Q_t} \sum_{i,j=1}^{d} \left( (s_t^i(X_t) - [\nabla \log q_t(X_t)]^i) s_t^j(X_t) + [\nabla \log q_t(X_t)]^i (s_t^j(X_t) - [\nabla \log q_t(X_t)]^j) \right)^2$$

$$\overset{(i)}{\le} 2\mathbb{E}_{X_t \sim Q_t} \sum_{i,j=1}^{d} (s_t^i(X_t) - [\nabla \log q_t(X_t)]^i)^2 (s_t^j(X_t))^2 + ([\nabla \log q_t(X_t)]^i)^2 (s_t^j(X_t) - [\nabla \log q_t(X_t)]^j)^2$$

$$= 2\mathbb{E}_{X_t \sim Q_t} \left[ \| s_t(X_t) - \nabla \log q_t(X_t) \|^2 \left( \| \nabla \log q_t(X_t) \|^2 + \| s_t(X_t) \|^2 \right) \right]$$

where $(i)$ follows because $(a + b)^2 = a^2 + b^2 + 2ab \le 2a^2 + 2b^2$. To continue, we use the Cauchy-Schwartz inequality and obtain

$$\mathbb{E}_{X_t \sim Q_t} \| s_t(X_t) s_t^\intercal(X_t) - (\nabla \log q_t(X_t))(\nabla \log q_t(X_t))^\intercal \|_F^2$$

$$\le 2\sqrt{\mathbb{E}_{X_t \sim Q_t} \| s_t(X_t) - \nabla \log q_t(X_t) \|^4} \sqrt{2\mathbb{E}_{X_t \sim Q_t} \left[ \| \nabla \log q_t(X_t) \|^4 + \| s_t(X_t) \|^4 \right]}.$$

Here the second term has that

$$\mathbb{E}[\| s_t(X_t) \|^4] \le 8\mathbb{E}[\| s_t(X_t) - \nabla \log q_t(X_t) \|^4] + 8\mathbb{E}[\| \nabla \log q_t(X_t) \|^4]$$

$$\lesssim \mathbb{E}[\| \nabla \log q_t(X_t) \|^4].$$

Therefore,

$$\frac{1}{T} \sum_{t=1}^{T} \mathbb{E}_{X_t \sim Q_t} \| H_t(X_t) - \nabla^2 \log q_t(X_t) \|_F^2$$

$$\le \frac{1}{T} \sum_{t=1}^{T} \mathbb{E}_{X_t \sim Q_t} \left\| v_\theta(X_t) - \left( \frac{\nabla^2 q_t(X_t)}{q_t(X_t)} + \frac{1}{1 - \bar{\alpha}_t} I_d \right) \right\|_F^2$$

$$+ \frac{1}{T} \sum_{t=1}^{T} \mathbb{E}_{X_t \sim Q_t} \left\| s_t(X_t) s_t^\intercal(X_t) - (\nabla \log q_t(X_t))(\nabla \log q_t(X_t))^\intercal \right\|_F^2$$

$$\lesssim \frac{1}{T} \sum_{t=1}^{T} \mathbb{E}_{X_t \sim Q_t} \left\| v_\theta(X_t) - \left( \frac{\nabla^2 q_t(X_t)}{q_t(X_t)} + \frac{1}{1 - \bar{\alpha}_t} I_d \right) \right\|_F^2$$

$$+ \frac{1}{T} \sum_{t=1}^{T} \sqrt{\mathbb{E}_{X_t \sim Q_t} \left\| s_t(X_t) - \nabla \log q_t(X_t) \right\|^4} \sqrt{\mathbb{E}_{X_t \sim Q_t} \left\| \nabla \log q_t(X_t) \right\|^4}$$

$$\overset{(ii)}{=} \frac{1}{T} \sum_{t=1}^{T} \mathbb{E}_{X_t \sim Q_t} \left\| v_\theta(X_t) - \left( \frac{\nabla^2 q_t(X_t)}{q_t(X_t)} + \frac{1}{1 - \bar{\alpha}_t} I_d \right) \right\|_F^2$$

$$+ \tilde{O} \left( \sqrt{\frac{1}{T} \sum_{t=1}^{T} (1 - \alpha_t)^2 \mathbb{E}_{X_t \sim Q_t} \left\| \nabla \log q_t(X_t) \right\|^4} \right)$$

$$\overset{(iii)}{=} \frac{1}{T} \sum_{t=1}^{T} \mathbb{E}_{X_t \sim Q_t} \left\| v_\theta(X_t) - \left( \frac{\nabla^2 q_t(X_t)}{q_t(X_t)} + \frac{1}{1 - \bar{\alpha}_t} I_d \right) \right\|_F^2 + \tilde{O}(T^{-1})$$

where $(ii)$ follows from (12) using the fact that $\frac{1}{T} \sum_{t=1}^{T} \sqrt{a_t} \le \sqrt{\frac{1}{T} \sum_{t=1}^{T} a_t}$ by Jensen's inequality, and $(iii)$ follows under Assumption 5. Combining this with (11), we finally get

$$\frac{1}{T} \sum_{t=1}^{T} \mathbb{E}_{X_t \sim Q_t} \left\| v_t(X_t) - \left( \frac{\nabla^2 q_t(X_t)}{q_t(X_t)} + \frac{1}{1 - \bar{\alpha}_t} I_d \right) \right\|_F^2 = \tilde{O}(T^{-1})$$

and thus the second relationship in Assumption 3 is satisfied. The proof is now complete. $\qquad\square$

## D  PROOF OF THEOREM 1

Instead of Assumption 4, we will prove Theorem 1 under the following more general assumption, which obviously implies Assumption 4 for any $\alpha_t$.

**Assumption 5** (Regular Partial Derivatives+). For all $t \ge 1$, $\ell \ge 1$, and $\boldsymbol{a} \in [d]^p$ such that $|\boldsymbol{a}| = p \ge 1$,

$$(1 - \alpha_t)^{p\ell/2} \mathbb{E}_{X_t \sim Q_t} |\partial_{\boldsymbol{a}}^p \log q_t(X_t)|^\ell = \tilde{O}\left((1 - \alpha_t)^{p\ell/2}\right),$$

$$(1 - \alpha_t)^{p\ell/2} \mathbb{E}_{X_t \sim Q_t} |\partial_{\boldsymbol{a}}^p \log q_{t-1}(\mu_t(X_t))|^\ell = \tilde{O}\left((1 - \alpha_t)^{p\ell/2}\right).$$

When $q_0$ does not exist, this is required only for $t \ge 2$.

To begin the proof of Theorem 1, note that

$$\mathrm{KL}(Q \| \widehat{P}') = \mathbb{E}_{X_0, \ldots, X_T \sim Q} \left[ \log \frac{q(X_0, \ldots, X_T)}{\widehat{p}'(X_0, \ldots, X_T)} \right]$$

$$\overset{(i)}{=} \mathbb{E}_{X_0, \ldots, X_T \sim Q} \left[ \log \frac{q_0(X_0) \prod_{t=1}^{T} q_{t|t-1}(X_t | X_{t-1})}{\widehat{p}'(X_0, \ldots, X_T)} \right]$$

$$\overset{(ii)}{=} \mathbb{E}_{X_0, \ldots, X_T \sim Q} \left[ \log \frac{q_0(X_0) \prod_{t=1}^{T} q_{t|t-1}(X_t | X_{t-1})}{\widehat{p}_0'(X_0) \prod_{t=1}^{T} \widehat{p}_{t|t-1}'(X_t | X_{t-1})} \right]$$

$$= \mathbb{E}_{X_0 \sim Q_0} \left[ \log \frac{q_0(X_0)}{\widehat{p}_0'(X_0)} \right] + \sum_{t=1}^{T} \mathbb{E}_{X_{t-1}, X_t \sim Q_{t-1,t}} \left[ \log \frac{q_{t|t-1}(X_t | X_{t-1})}{\widehat{p}_{t|t-1}'(X_t | X_{t-1})} \right]$$

$$= \mathbb{E}_{X_0 \sim Q_0} \left[ \log \frac{q_0(X_0)}{\widehat{p}_0'(X_0)} \right] + \sum_{t=1}^{T} \mathbb{E}_{X_{t-1} \sim Q_{t-1}} \left[ \mathbb{E}_{X_t \sim Q_{t|t-1}} \left[ \log \frac{q_{t|t-1}(X_t | X_{t-1})}{\widehat{p}_{t|t-1}'(X_t | X_{t-1})} \right] \right]$$

$$= \mathrm{KL}(Q_0||\widehat{P}'_0) + \sum_{t=1}^{T} \mathbb{E}_{X_{t-1} \sim Q_{t-1}} \left[ \mathrm{KL}(Q_{t|t-1}(\cdot|X_{t-1})||\widehat{P}'_{t|t-1}(\cdot|X_{t-1})) \right].$$

Here $(i)$ holds because of the Markov property of the forward process. We explain $(ii)$ below. By the backward Markov property of the reverse process, for any $t \geq 1$, given $X_{t-1} = x_{t-1}$, each of $X_{t-2}, \ldots, X_0$ is independent with $X_t$. This implies that

$$\widehat{p}'_{t|t-1,\ldots,0}(x_t|x_{t-1}, \ldots, x_0) = \widehat{p}'_{t|t-1}(x_t|x_{t-1}), \quad \forall t \geq 1.$$

Thus, $\widehat{p}'(x_0, \ldots, x_T) = \widehat{p}'_0(x_0) \prod_{t=1}^{T} \widehat{p}'_{t|t-1}(x_t|x_{t-1})$. In other words, $X_0, \ldots, X_t$ is also forward Markov under $\widehat{P}'$.

Following from similar arguments,

$$\mathrm{KL}(Q||\widehat{P}') = \mathrm{KL}(Q_T||\widehat{P}'_T) + \sum_{t=1}^{T} \mathbb{E}_{X_t \sim Q_t} \left[ \mathrm{KL}(Q_{t-1|t}(\cdot|X_t)||\widehat{P}'_{t-1|t}(\cdot|X_t)) \right].$$

Since KL-divergence is non-negative, an upper bound on $\mathrm{KL}(Q_0||\widehat{P}'_0)$ is given by

$$\mathrm{KL}(Q_0||\widehat{P}'_0)$$

$$= \mathrm{KL}(Q_T||\widehat{P}'_T) + \sum_{t=1}^{T} \mathbb{E}_{X_t \sim Q_t} \left[ \mathrm{KL}(Q_{t-1|t}(\cdot|X_t)||\widehat{P}'_{t-1|t}(\cdot|X_t)) \right]$$

$$- \sum_{t=1}^{T} \mathbb{E}_{X_{t-1} \sim Q_{t-1}} \left[ \mathrm{KL}(Q_{t|t-1}(\cdot|X_{t-1})||\widehat{P}'_{t|t-1}(\cdot|X_{t-1})) \right]$$

$$\leq \mathrm{KL}(Q_T||\widehat{P}'_T) + \sum_{t=1}^{T} \mathbb{E}_{X_t \sim Q_t} \left[ \mathrm{KL}(Q_{t-1|t}(\cdot|X_t)||\widehat{P}'_{t-1|t}(\cdot|X_t)) \right]$$

$$= \underbrace{\mathbb{E}_{X_T \sim Q_T} \left[ \log \frac{q_T(X_T)}{p'_T(X_T)} \right]}_{\text{Term 1: initialization error}} + \underbrace{\sum_{t=1}^{T} \mathbb{E}_{X_t, X_{t-1} \sim Q_{t,t-1}} \left[ \log \frac{p'_{t-1|t}(X_{t-1}|X_t)}{\widehat{p}'_{t-1|t}(X_{t-1}|X_t)} \right]}_{\text{Term 2: estimation error}}$$

$$+ \underbrace{\sum_{t=1}^{T} \mathbb{E}_{X_t, X_{t-1} \sim Q_{t,t-1}} \left[ \log \frac{q_{t-1|t}(X_{t-1}|X_t)}{p'_{t-1|t}(X_{t-1}|X_t)} \right]}_{\text{Term 3: reverse-step error}}. \tag{13}$$

The last equality holds because $\widehat{p}'_T = p'_T$.

Next, we bound the above three terms separately in a few steps.

### D.1   STEP 0: BOUNDING TERM 1 – INITIALIZATION ERROR

**Lemma 3.** *Suppose $\bar{\alpha}_T \searrow 0$ as $T \to \infty$. Then, under Assumption 1,*

$$\mathbb{E}_{X_T \sim Q_T} \left[ \log \frac{q_T(X_T)}{p'_T(X_T)} \right] \leq \frac{1}{2} M_2 \bar{\alpha}_T d + O\left(\bar{\alpha}_T^2\right), \ \ as \ T \to \infty.$$

*Remark* 1. Under Assumption 1, if the noise schedule satisfies Definition 1, we have

$$\mathbb{E}_{X_T \sim Q_T} \left[ \log \frac{q_T(X_T)}{p'_T(X_T)} \right] = o(T^{-2}).$$

*Proof.* See Appendix F.1. $\qquad\qquad\qquad\qquad\qquad\qquad\qquad\qquad\qquad\qquad\qquad\qquad\square$

We now introduce the following notation for analyzing the estimation error and the reverse-step error for the accelerated sampler.

**Definition 2** (Big-O in $\mathcal{L}^r$ space). For a random variable $Z_T$, we say that $Z_T(x) = O_{\mathcal{L}^r(Q)}(1)$ if $(\mathbb{E}_{X \sim Q} |Z_T(X)|^r)^{1/r} = O(1)$ for all $r \geq 1$ as $T \to \infty$.

One property is that if $Z_T(x) = O_{\mathcal{L}^r(Q)}(1)$ then $\mathbb{E}_{X \sim Q} |Z_T(X)| = O(1)$. Another property is that if $Z_1 = O_{\mathcal{L}^r(Q)}(a_T)$ and $Z_2 = O_{\mathcal{L}^r(Q)}(b_T)$ for all $r \geq 1$, applying Cauchy-Schwartz inequality we get, for all $r \geq 1$,

$$\left(\mathbb{E} |Z_1 Z_2|^r\right)^{1/r} \leq \left(\mathbb{E} Z_1^{2r} \mathbb{E} Z_2^{2r}\right)^{1/(2r)} = O(a_T b_T),$$

which implies that $O_{\mathcal{L}^r(Q)}(a_T) O_{\mathcal{L}^r(Q)}(b_T) = O_{\mathcal{L}^r(Q)}(a_T b_T)$. Now, with this notation, the regularity condition on $H_t$ can be written as

$$(1 - \alpha_t) \|H_t(X_t)\| = \tilde{O}_{\mathcal{L}^r(Q_t)}(1 - \alpha_t), \ \forall r \geq 1.$$

Also, Assumption 5 can be equivalently written as, $\forall r \geq 1$,

$$(1 - \alpha_t)^{p/2} |\partial_a^p \log q_t(X_t)| = \tilde{O}_{\mathcal{L}^r(Q_t)}\left((1 - \alpha_t)^{p/2}\right),$$

$$(1 - \alpha_t)^{p/2} |\partial_a^p \log q_{t-1}(\mu_t(X_t))| = \tilde{O}_{\mathcal{L}^r(Q_t)}\left((1 - \alpha_t)^{p/2}\right).$$

## D.2  STEP 1: BOUNDING TERM 2 – SCORE AND HESSIAN ESTIMATION ERROR

We first bound the estimation error, which includes the errors that come from the score and the Hessian estimation. In particular, Assumption 5 guarantees that all higher Taylor terms are well controlled in expectation over $X_t \sim Q_t$.

**Lemma 4.** *Under Assumptions 3 and 5, with the $\alpha_t$ satisfying Definition 1, we have*

$$\sum_{t=1}^{T} \mathbb{E}_{X_t, X_{t-1} \sim Q_{t,t-1}} \left[ \log \frac{p'_{t-1|t}(X_{t-1}|X_t)}{\widehat{p}'_{t-1|t}(X_{t-1}|X_t)} \right] \lesssim (\log T)\varepsilon^2 + \frac{\log^2 T}{T} \varepsilon_H^2.$$

*Remark* 2. Under Assumption 3, Lemma 4 guarantees that

$$\sum_{t=1}^{T} \mathbb{E}_{X_t, X_{t-1} \sim Q_{t,t-1}} \left[ \log \frac{p'_{t-1|t}(X_{t-1}|X_t)}{\widehat{p}'_{t-1|t}(X_{t-1}|X_t)} \right] = \tilde{O}\left(\frac{1}{T^2}\right).$$

*Proof.* See Appendix F.2. □

Before we proceed to the reverse-step error, we provide the following lemma to provide an upper bound when we use the $\tilde{\Sigma}_t$ and its estimate according to (9).

**Corollary 2.** *Under the same conditions of Lemma 4, the upper bound in Lemma 4 on the estimation error still holds with the slightly perturbed $\tilde{\Sigma}_t$ provided in* (9).

*Proof.* See Appendix F.3. □

## D.3  STEP 2: EXPRESSING LOG-LIKELIHOOD RATIO VIA TILTING FACTOR

Next we focus on the reverse-step error for the accelerated process. Recall that $Q_0$ is smooth under Assumption 2. We introduce the following notations for analysis. Let

$$A_t(x_t) := (1 - \alpha_t) \nabla^2 \log q_t(x_t), \quad B_t(x_t) := I_d - (I_d + A_t(x_t))^{-1}, \tag{14}$$

which imply that

$$\Sigma_t(x_t) = \frac{1 - \alpha_t}{\alpha_t}(I_d + A_t(x_t)), \quad \Sigma_t^{-1}(x_t) = \frac{\alpha_t}{1 - \alpha_t}(I_d - B_t(x_t)).$$

Now, with the notation in Definition 2, for each $i, j \in [d]$, $A_t^{ij}(x_t) = \tilde{O}_{\mathcal{L}^r(Q_t)}(1 - \alpha_t)$ for all $r \geq 1$ under Assumption 5. Also, when $(1 - \alpha_t)$ is small, we can perform Taylor expansion on $B_t(\cdot)$ around $A_t(\cdot)$ and obtain, under Assumption 5,

$$B_t(X_t) = A_t(X_t) + \tilde{O}_{\mathcal{L}^r(Q_t)}\left((1 - \alpha_t)^2\right). \tag{15}$$

*Remark* 3. In general, suppose that we choose $P'_{t-1|t}$ whose conditional covariance satisfies

$$\tilde{\Sigma}_t(X_t) = \frac{1-\alpha_t}{\alpha_t}\left(I_d + A_t(X_t) + \tilde{O}_{\mathcal{L}^r(Q_t)}\left((1-\alpha_t)^2\right)\right) = \Sigma_t(X_t) + \tilde{O}_{\mathcal{L}^r(Q_t)}\left((1-\alpha_t)^3\right),$$

where a small perturbation is added to the covariance matrix. An immediate consequence is that

$$\tilde{\Sigma}_t^{-1}(X_t) = \frac{\alpha_t}{1-\alpha_t}\left(I_d - B_t(X_t) + \tilde{O}_{\mathcal{L}^r(Q_t)}\left((1-\alpha_t)^2\right)\right) = \Sigma_t^{-1}(X_t) + \tilde{O}_{\mathcal{L}^r(Q_t)}\left(1-\alpha_t\right).$$

Then, with such $P'_{t-1|t}$ having a slightly perturbed covariance, the following Lemmas 5 and 7 still hold with $\tilde{A}_t(x_t)$ and $\tilde{B}_t(x_t)$ such that

$$\tilde{A}_t(x_t) := \frac{\alpha_t}{1-\alpha_t}\tilde{\Sigma}_t(x_t) - I_d, \quad \tilde{B}_t(x_t) := I_d - (I_d + \tilde{A}_t(x_t))^{-1}.$$

Note that $\tilde{A}_t(X_t) = A_t(X_t) + \tilde{O}_{\mathcal{L}^r(Q_t)}\left((1-\alpha_t)^2\right)$ and $\tilde{B}_t(X_t) = B_t(X_t) + \tilde{O}_{\mathcal{L}^r(Q_t)}\left((1-\alpha_t)^2\right)$.

In the following we write $\mu_t = \mu_t(x_t)$, $A_t = A_t(x_t)$, and $B_t = B_t(x_t)$ for brevity.

**Lemma 5.** *For any fixed $x_t \in \mathbb{R}^d$, as long as $q_{t-1}$ is defined, we have*

$$q_{t-1|t}(x_{t-1}|x_t) = \frac{p'_{t-1|t}(x_{t-1}|x_t)e^{\zeta'_{t,t-1}(x_t,x_{t-1})}}{\mathbb{E}_{X_{t-1}\sim P'_{t-1|t}}[e^{\zeta'_{t,t-1}(x_t,X_{t-1})}]},$$

*where*

$$\zeta_{t,t-1}(x_t,x_{t-1}) := \log q_{t-1}(x_{t-1}) - \log q_{t-1}(\mu_t) - (x_{t-1} - \mu_t)^\mathsf{T}(\sqrt{\alpha_t}\nabla \log q_t(x_t)), \quad (16)$$

*and*

$$\zeta'_{t,t-1}(x_t,x_{t-1}) := \zeta_{t,t-1}(x_t,x_{t-1}) - \frac{\alpha_t}{2(1-\alpha_t)}(x_{t-1}-\mu_t)^\mathsf{T}B_t(x_{t-1}-\mu_t)$$

$$= \log q_{t-1}(x_{t-1}) - \log q_{t-1}(\mu_t) - (x_{t-1}-\mu_t)^\mathsf{T}(\sqrt{\alpha_t}\nabla \log q_t(x_t))$$

$$- \frac{\alpha_t}{2(1-\alpha_t)}(x_{t-1}-\mu_t)^\mathsf{T}B_t(x_{t-1}-\mu_t). \quad (17)$$

*Proof.* See Appendix F.4. $\square$

In the following we write $\zeta_{t,t-1} = \zeta_{t,t-1}(x_t,x_{t-1})$ and $\zeta'_{t,t-1} = \zeta'_{t,t-1}(x_t,x_{t-1})$ and omit dependencies on $x_t$ and $x_{t-1}$ for brevity. As we will see, (16) is the tilting factor for the regular diffusion process. Given the definition of $\zeta'_{t,t-1}$ in (17), below we analyze $\log q_{t-1}(x)$ around $x = \mu_t$ using Taylor expansion. We first provide the following notations for the Taylor expansion.

**Definition 3** (Taylor Expansion). Recall that $x^i$ $(1 \le i \le d)$ denotes the $i$-th element of a vector $x$. Given an *analytic* function $f(x)$, its Taylor expansion around $x = \mu$ is given by

$$f(x) = f(\mu) + \sum_{p=1}^{\infty} T_p(f,x,\mu)$$

$$= f(\mu) + \nabla f(\mu)^\mathsf{T}(x-\mu) + \frac{1}{2}\sum_{i=1}^{d}\partial_{ii}^2 f(\mu)(x^i-\mu^i)^2 + \frac{1}{2}\sum_{\substack{i,j=1\\i\neq j}}^{d}\partial_{ij}^2 f(\mu)(x^i-\mu^i)(x^j-\mu^j)$$

$$+ \sum_{p=3}^{\infty} T_p(f,x,\mu)$$

where, for $p \ge 1$, we define

$$T_p(f,x,\mu) := \frac{1}{p!}\sum_{\gamma\in\mathbb{N}^d:\sum_i \gamma^i=p}\partial_{\boldsymbol{a}}^p f(\mu)\prod_{i=1}^{d}(x^i-\mu^i)^{\gamma^i} \quad (18)$$

where in $\boldsymbol{a} \in [d]^p$ the multiplicity of $i$ $(\in [d])$ is $\gamma^i$.

If we specialize it to the case where $f = \log q_{t-1}$, $x = x_{t-1}$, and $\mu = \mu_t$, we need the following lemma to guarantee the validity of Taylor expansion for $t \geq 1$.

**Lemma 6.** *Fix $t \geq 1$. For any $Q_0$ (not necessarily having a p.d.f. w.r.t. the Lebesgue measure), given any $k \geq 1$ and any vector of indices $\boldsymbol{a} \in [d]^k$, $q_t$ exists and $|\partial_{\boldsymbol{a}}^k \log q_t(x_t)| < \infty$, $\forall x_t \in \mathbb{R}^d$ (which possibly depends on $T$). Further, $q_t$ and $\log q_t$ are both analytic.*

*Proof.* See Appendix F.5. $\qquad\square$

Thus, by Assumption 2 and Lemma 6, since $\log q_{t-1}$ is analytic, its Taylor expansion around $x_{t-1} = \mu_t$ is equal to (cf. (16))

$$\zeta_{t,t-1} = (\nabla \log q_{t-1}(\mu_t) - \sqrt{\alpha_t} \nabla \log q_t(x_t))^\intercal (x_{t-1} - \mu_t) + \sum_{p=2}^{\infty} T_p(\log q_{t-1}, x_{t-1}, \mu_t), \quad (19)$$

and the Taylor expansion of $\zeta'_{t,t-1}(x_t, x_{t-1})$ around $x_{t-1} = \mu_t$ is (cf. (17))

$$\begin{aligned}
\zeta'_{t,t-1} = {} & (\nabla \log q_{t-1}(\mu_t) - \sqrt{\alpha_t} \nabla \log q_t(x_t))^\intercal (x_{t-1} - \mu_t) \\
& + \frac{1}{2}(x_{t-1} - \mu_t)^\intercal \left( \nabla^2 \log q_{t-1}(\mu_t) - \frac{\alpha_t}{1 - \alpha_t} B_t \right) (x_{t-1} - \mu_t) \\
& + \sum_{p=3}^{\infty} T_p(\log q_{t-1}, x_{t-1}, \mu_t).
\end{aligned} \quad (20)$$

In order to differentiate the second-order terms in (19) and (20), we reserve $T_2$ for (19) and employ for (20):

$$T'_2(\log q_{t-1}, x_{t-1}, \mu_t) := \frac{1}{2}(x_{t-1} - \mu_t)^\intercal \left( \nabla^2 \log q_{t-1}(\mu_t) - \frac{\alpha_t}{1 - \alpha_t} B_t \right) (x_{t-1} - \mu_t).$$

Compared with the tilting factor for the regular process in $\zeta_{t,t-1}$, an additional term that is related to $\Sigma_t$ (and thus $B_t$) is introduced in $\zeta'_{t,t-1}$. From the perspective of Taylor expansion, we can further control the *second*-order term in the Taylor expansion of $\log q_{t-1}$ around $\mu_t$ through this extra term, which improves the accuracy of posterior approximation at each step.

To use Taylor expansion to upper-bound the reverse-step error in (13), we first note that, for any fixed $x_t$,

$$\begin{aligned}
& \mathbb{E}_{X_{t-1} \sim Q_{t-1|t}} \left[ \log \frac{q_{t-1|t}(X_{t-1}|x_t)}{p'_{t-1|t}(X_{t-1}|x_t)} \right] \\
& = \mathbb{E}_{X_{t-1} \sim Q_{t-1|t}} \left[ \zeta'_{t,t-1} - \log \mathbb{E}_{X_{t-1} \sim P'_{t-1|t}} [e^{\zeta'_{t,t-1}}] \right] \\
& = \mathbb{E}_{X_{t-1} \sim Q_{t-1|t}} \left[ \zeta'_{t,t-1} \right] - \log \mathbb{E}_{X_{t-1} \sim P'_{t-1|t}} [e^{\zeta'_{t,t-1}}] \\
& \overset{(i)}{\leq} \mathbb{E}_{X_{t-1} \sim Q_{t-1|t}} \left[ \zeta'_{t,t-1} \right] + \mathbb{E}_{X_{t-1} \sim P'_{t-1|t}} \left[ -\log e^{\zeta'_{t,t-1}} \right] \\
& = \mathbb{E}_{X_{t-1} \sim Q_{t-1|t}} [\zeta'_{t,t-1}] - \mathbb{E}_{X_{t-1} \sim P'_{t-1|t}} [\zeta'_{t,t-1}]
\end{aligned} \quad (21)$$

where in $(i)$ we use Jensen's inequality and note that $-\log(\cdot)$ is convex. In the remaining steps, we analyze the expected values of the tilting factor separately.

### D.4 STEP 3: CONDITIONAL EXPECTATION OF $\zeta'_{t,t-1}$ UNDER $P'_{t-1|t}$

With Taylor expansion around the posterior mean, the calculation of the expected values is reduced to that of all the (centralized) moments. To start, it is useful to examine the rate of $\frac{1-\alpha_t}{\alpha_t}$. A direct implication of Definition 1 is that, with some constant $C_1$, since $\alpha_t \searrow 0$ as $T \to \infty$,

$$\frac{(1 - \alpha_t)^p}{\alpha_t^q} \leq \frac{C_1^p \log^p T / T^p}{(1 - C_1 \log T / T)^q} \lesssim (1 - \alpha_t)^p, \quad \forall p, q \geq 1, t \geq 1. \quad (22)$$

Below, we first calculate the centralized moments under $P'_{t-1|t}$. We employ Isserlis's Theorem for our help, which constitutes the main idea in the lemma below. Note that the results in this subsection hold as long as $Q_0$ has a p.d.f..

**Lemma 7.** *Fix $t \geq 1$. For brevity write $Z_i = X_{t-1}^i - \mu_t^i$, $\forall i \in [d]$, $A = A_t(x_t)$, and $\mathbb{E}[\cdot]$ as a shorthand for $\mathbb{E}_{X_{t-1} \sim P'_{t-1|t}}[\cdot]$. Note that we have $A_t^{ij}(x_t) = \tilde{O}_{\mathcal{L}^p(Q_t)}(1 - \alpha_t)$ for all $i, j \in [d]$ under Assumption 5. Thus, the following results hold: $\forall p \geq 1$,*

$$\mathbb{E}\left[\prod_{i \in \boldsymbol{a}} Z_i\right] = 0, \quad \forall \boldsymbol{a} : |\boldsymbol{a}| \text{ odd},$$

$$\mathbb{E}\left[\prod_{i \in \boldsymbol{a}} Z_i\right] = \tilde{O}_{\mathcal{L}^p(Q_t)}\left((1 - \alpha_t)^{\frac{|\boldsymbol{a}|}{2}}\right), \quad \forall \boldsymbol{a} : |\boldsymbol{a}| \text{ even}.$$

*Specifically, for $i, j, k, l \in [d]$ all differ, the fourth moment is*

$$\mathbb{E}[Z_i^4] = 3\left(\frac{1 - \alpha_t}{\alpha_t}\right)^2 (1 + A^{ii})^2$$

$$\mathbb{E}[Z_i^3 Z_j] = 3\left(\frac{1 - \alpha_t}{\alpha_t}\right)^2 A^{ij}(1 + A^{ii})$$

$$\mathbb{E}[Z_i^2 Z_j^2] = \left(\frac{1 - \alpha_t}{\alpha_t}\right)^2 (1 + A^{ii})(1 + A^{jj}) + \tilde{O}_{\mathcal{L}^p(Q_t)}((1 - \alpha_t)^4)$$

$$\mathbb{E}[Z_i^2 Z_j Z_k] = \left(\frac{1 - \alpha_t}{\alpha_t}\right)^2 (1 + A^{ii}) A^{jk} + \tilde{O}_{\mathcal{L}^p(Q_t)}((1 - \alpha_t)^4)$$

$$\mathbb{E}[Z_i Z_j Z_k Z_l] = \tilde{O}_{\mathcal{L}^p(Q_t)}((1 - \alpha_t)^4).$$

*For $i, j, k \in [d]$ all differ, the sixth moment is*

$$\mathbb{E}[Z_i^6] = 15\left(\frac{1 - \alpha_t}{\alpha_t}\right)^3 (1 + A^{ii})^3$$

$$\mathbb{E}[Z_i^4 Z_j^2] = 3\left(\frac{1 - \alpha_t}{\alpha_t}\right)^3 (1 + A^{ii})^2 (1 + A^{jj}) + \tilde{O}_{\mathcal{L}^p(Q_t)}((1 - \alpha_t)^4)$$

$$\mathbb{E}[Z_i^2 Z_j^2 Z_k^2] = \left(\frac{1 - \alpha_t}{\alpha_t}\right)^3 (1 + A^{ii})(1 + A^{jj})(1 + A^{kk}) + \tilde{O}_{\mathcal{L}^p(Q_t)}((1 - \alpha_t)^4),$$

*and $\mathbb{E}\left[\prod_{i \in \boldsymbol{a}: |\boldsymbol{a}| = 6} Z_i\right] = \tilde{O}_{\mathcal{L}^p(Q_t)}((1 - \alpha_t)^4)$ otherwise. All the rates are under Assumption 5.*

*Proof.* See Appendix F.6. $\square$

### D.5 STEP 4: CONDITIONAL EXPECTATION OF $\zeta'_{t,t-1}$ UNDER $Q_{t-1|t}$

Although each $Q_{t|t-1}$ is conditionally Gaussian, the posterior $Q_{t-1|t}$ is not Gaussian in general. In the following, we analyze the posterior centralized moments under $Q_{t-1|t}$ using the idea of Tweedie's formula Efron (2011). Then, we apply them to analyze $\mathbb{E}_{X_{t-1} \sim Q_{t-1|t}}[\zeta_{t,t-1}]$, again using the Taylor expansion in (19). Again, the result is more generally applicable to non-smooth $Q_0$ at $t \geq 2$ due to Lemma 6.

**Lemma 8.** *Fix $t \geq 1$ such that $q_{t-1}$ exists. Define $\tilde{x}_t := \frac{\sqrt{\alpha_t}}{1 - \alpha_t} x_t$, and*

$$\kappa(\tilde{x}_t) := \log q_t\left(\frac{1 - \alpha_t}{\sqrt{\alpha_t}} \tilde{x}_t\right) + \frac{1 - \alpha_t}{2\alpha_t} \|\tilde{x}_t\|^2 + \frac{d}{2} \log\left(2\pi(1 - \alpha_t)\right). \tag{23}$$

*Let $1 \leq i, j, k, l \leq d$, which are possibly equal to each other. The first 3 centralized moments under $Q_{t-1|t}$ satisfy*

$$\mathbb{E}_{X_{t-1} \sim Q_{t-1|t}}[X_{t-1}] = \nabla\kappa = \mu_t$$

$$\mathbb{E}_{X_{t-1} \sim Q_{t-1|t}}[(X_{t-1} - \mu_t)(X_{t-1} - \mu_t)^{\mathsf{T}}] = \nabla^2\kappa = \frac{1 - \alpha_t}{\alpha_t} I_d + \frac{(1 - \alpha_t)^2}{\alpha_t} \nabla^2 \log q_t(x_t)$$

$$\mathbb{E}_{X_{t-1}, X_t \sim Q_{t-1,t}}\left[(X_{t-1}^i - \mu_t^i)(X_{t-1}^j - \mu_t^j)(X_{t-1}^k - \mu_t^k)\right]$$

$$= \mathbb{E}_{X_t \sim Q_t}[\partial^3_{ijk}\kappa] = \frac{(1-\alpha_t)^3}{\alpha_t^{3/2}}\mathbb{E}_{X_t \sim Q_t}[\partial^3_{ijk}\log q_t(X_t)] = \tilde{O}((1-\alpha_t)^3).$$

*The fourth centralized moment satisfies*

$$\mathbb{E}_{X_{t-1}, X_t \sim Q_{t-1,t}}\left[(X^i_{t-1} - \mu^i_t)(X^j_{t-1} - \mu^j_t)(X^k_{t-1} - \mu^k_t)(X^l_{t-1} - \mu^l_t)\right]$$

$$= \mathbb{E}_{X_t \sim Q_t}[(\partial^2_{ij}\kappa)(\partial^2_{kl}\kappa) + (\partial^2_{ik}\kappa)(\partial^2_{jl}\kappa) + (\partial^2_{il}\kappa)(\partial^2_{jk}\kappa) + \partial^4_{ijkl}\kappa]$$

$$= \begin{cases} 3\left(\frac{1-\alpha_t}{\alpha_t}\right)^2 + \tilde{O}((1-\alpha_t)^3), & \text{if } i = j = k = l, \\ \left(\frac{1-\alpha_t}{\alpha_t}\right)^2 + \tilde{O}((1-\alpha_t)^3), & \text{if } i = k \neq j = l, \\ \tilde{O}((1-\alpha_t)^3), & \text{otherwise.} \end{cases}$$

*Note that all derivatives above are w.r.t. $\tilde{x}_t$. All the rates are under Assumption 5.*

*Proof.* See Appendix F.7. □

Lemma 8 also justifies the expression of $\mu_t$ and $\Sigma_t$ in the diffusion process (i.e., (3) and (4)), which match the posterior mean and variance, respectively.

Next we turn to calculate the fifth and sixth centralized moment under $Q_{t-1|t}$, again drawing the idea of Tweedie's formula (Efron, 2011). This is a direct extension to Lemma 8.

**Lemma 9.** *Fix $t \geq 1$ such that $q_{t-1}$ exists. Fix $x_t \in \mathbb{R}^d$. Under Assumption 5, with the same definitions of $\tilde{x}_t$ and $\kappa(\tilde{x}_t)$ as in Lemma 8, the fifth centralized moment is*

$$\mathbb{E}_{X_{t-1} \sim Q_{t-1|t}}\left[(X^i_{t-1} - \mu^i_t)(X^j_{t-1} - \mu^j_t)(X^k_{t-1} - \mu^k_t)(X^l_{t-1} - \mu^l_t)(X^m_{t-1} - \mu^m_t)\right]$$

$$= \sum_{\xi \in \binom{\{i,j,k,l,m\}}{2}} (\partial^2_\xi \kappa)(\partial^3_{\{i,j,k,l,m\}\setminus\xi}\kappa) + \partial^5_{ijklm}\kappa = \tilde{O}_{\mathcal{L}^p(Q_t)}((1-\alpha_t)^4)$$

*where, given a set $A$, we define*

$$\binom{A}{2} := \left\{\{a_1, a_2\} : a_1, a_2 \in A,\ a_1 \neq a_2\right\}.$$

*Let $P^k_n$ be the set that contains all distinct size-$k$ partitions of $[n]$. Define*

$$\text{part}_2(A) := \{((a_i, a_j) : \{i, j\} \in p) : p \in P^2_{|A|}\}.$$

*The sixth centralized moment is*

$$\mathbb{E}_{X_{t-1} \sim Q_{t-1|t}}\left[(X^i_{t-1} - \mu^i_t)(X^j_{t-1} - \mu^j_t)(X^k_{t-1} - \mu^k_t)(X^l_{t-1} - \mu^l_t)(X^m_{t-1} - \mu^m_t)(X^n_{t-1} - \mu^n_t)\right]$$

$$= \sum_{(\xi_1, \xi_2, \xi_3) \in \text{part}_2(\{i,j,k,l,m,n\})} (\partial^2_{\xi_1}\kappa)(\partial^2_{\xi_2}\kappa)(\partial^2_{\xi_3}\kappa) + \tilde{O}_{\mathcal{L}^p(Q_t)}((1-\alpha_t)^4)$$

$$= \begin{cases} 15\left(\frac{1-\alpha_t}{\alpha_t}\right)^3 + \tilde{O}_{\mathcal{L}^p(Q_t)}((1-\alpha_t)^4), & \text{if } i = j = k = l = m = n \\ 3\left(\frac{1-\alpha_t}{\alpha_t}\right)^3 + \tilde{O}_{\mathcal{L}^p(Q_t)}((1-\alpha_t)^4), & \text{if } i = k = m = n \neq j = l \\ \left(\frac{1-\alpha_t}{\alpha_t}\right)^3 + \tilde{O}_{\mathcal{L}^p(Q_t)}((1-\alpha_t)^4), & \text{if } i = l, j = m, k = n \text{ while } i, j, k \text{ all differ} \\ \tilde{O}_{\mathcal{L}^p(Q_t)}((1-\alpha_t)^4), & \text{otherwise} \end{cases}$$

*Again note that all derivatives above are w.r.t. $\tilde{x}_t$.*

*Proof.* See Appendix F.8. □

The following lemma provides the correct order (in terms of $(1 - \alpha_t)$) for all higher-order posterior centralized moments. In other words, this shows that $Q_{t-1|t}$ has nice Gaussian-like concentration.

**Lemma 10.** *Fix $t \geq 1$ and $p \geq 2$. Let $\boldsymbol{a} = (a_1, \ldots, a_p) \in [d]^p$ be a vector of indices of length $p$. Under the same conditions as in Lemma 8, if $p$ is odd,*

$$\mathbb{E}_{X_{t-1}, X_t \sim Q_{t-1,t}} \left[ \prod_{i=1}^{p} (X_{t-1}^{a_i} - \mu_t^{a_i}) \right] = \tilde{O}\left( (1-\alpha_t)^{\frac{p+3}{2}} \right), \ \forall \boldsymbol{a} \in [d]^p. \tag{24}$$

*If $p$ is even,*

$$\mathbb{E}_{X_{t-1}, X_t \sim Q_{t-1,t}} \left[ \prod_{i=1}^{p} (X_{t-1}^{a_i} - \mu_t^{a_i}) \right] = \tilde{O}((1-\alpha_t)^{\frac{p}{2}}), \ \forall \boldsymbol{a} \in [d]^p. \tag{25}$$

*Proof.* See Appendix F.9. $\qquad\square$

### D.6 STEP 5: BOUNDING TERM 3 – REVERSE-STEP ERROR

We are now ready to assemble the respective moments into the final convergence rate. In the following lemma, we use the results in the previous lemmas to control the difference $\mathbb{E}_{X_{t-1} \sim Q_{t-1|t}}[\zeta'_{t,t-1}] - \mathbb{E}_{X_{t-1} \sim P'_{t-1|t}}[\zeta'_{t,t-1}]$ in (21).

**Lemma 11.** *Suppose that Assumption 5 holds and that $q_{t-1}$ exists. Then,*

$$\mathbb{E}_{X_t \sim Q_t} \left( \mathbb{E}_{X_{t-1} \sim Q_{t-1|t}} - \mathbb{E}_{X_{t-1} \sim P'_{t-1|t}} \right) [\zeta'_{t,t-1}]$$

$$= \frac{(1-\alpha_t)^3}{3! \alpha_t^{3/2}} \sum_{i,j,k=1}^{d} \mathbb{E}_{X_t \sim Q_t}[\partial_{ijk}^3 \log q_{t-1}(\mu_t(X_t)) \partial_{ijk}^3 \log q_t(X_t)] + \tilde{O}((1-\alpha_t)^4).$$

*Proof.* See Appendix F.10. $\qquad\square$

Therefore, under Assumptions 2 and 5 we combine Lemma 11 and (21) and get

$$\sum_{t=1}^{T} \mathbb{E}_{X_{t-1}, X_t \sim Q_{t-1|t}} \left[ \log \frac{q_{t-1|t}(X_{t-1}|X_t)}{p'_{t-1|t}(X_{t-1}|X_t)} \right]$$

$$\lesssim (1-\alpha_t)^3 \sum_{i,j,k=1}^{d} \mathbb{E}_{X_t \sim Q_t}[\partial_{ijk}^3 \log q_{t-1}(\mu_t(X_t)) \partial_{ijk}^3 \log q_t(X_t)]. \tag{26}$$

This completes the proof of Theorem 1.

Before we end this section, we provide an upper bound of the reverse-step error when the conditional covariance of $P'_{t-1|t}$ is slightly perturbed (see Remark 3).

**Corollary 3.** *Suppose that Assumption 5 holds and that $q_{t-1}$ exists. Suppose that the conditional covariance of $P'_{t-1|t}$ is slightly perturbed, which satisfies*

$$\tilde{\Sigma}_t(x_t) = \frac{1-\alpha_t}{\alpha_t} \left( I_d + A_t(x_t) + \Xi_t(x_t) \right),$$

*where $\Xi_t(X_t) = \tilde{O}_{\mathcal{L}^r(Q_t)} \left( (1-\alpha_t)^2 \right)$ for all $r \geq 1$. Then,*

$$\sum_{t=1}^{T} \mathbb{E}_{X_{t-1}, X_t \sim Q_{t-1|t}} \left[ \log \frac{q_{t-1|t}(X_{t-1}|X_t)}{p'_{t-1|t}(X_{t-1}|X_t)} \right]$$

$$\lesssim -(1-\alpha_t) \mathbb{E}_{X_t \sim Q_t} \operatorname{Tr} \left( \left( \nabla^2 \log q_{t-1}(\mu_t(X_t)) - \alpha_t \nabla^2 \log q_t(X_t) \right) \Xi_t(X_t) \right)$$

$$+ (1-\alpha_t)^3 \sum_{i,j,k=1}^{d} \mathbb{E}_{X_t \sim Q_t}[\partial_{ijk}^3 \log q_{t-1}(\mu_t(X_t)) \partial_{ijk}^3 \log q_t(X_t)]$$

$$= \tilde{O}\left( \frac{1}{T^2} \right).$$

*Proof.* See Appendix F.11. $\qquad\square$

## E  PROOF OF COROLLARY 1

Note that $q_1$ always exists and is analytic by Lemma 6. Therefore, it remains to upper-bound the mismatch between $Q_0$ and $Q_1$. In the following lemma we provide such a common bound in Wasserstein distance, which is provided only for completeness.

**Lemma 12.** *For any $Q_0$,*
$$W_2(Q_0, Q_1)^2 \le (1 - \alpha_1)(M_2 + 1)d.$$

*Remark 4.* If $1 - \alpha_1 = \delta$, this implies that
$$W_2(Q_0, Q_1)^2 \lesssim \delta d.$$

*Proof.* See Appendix F.12. □

The proof of this corollary is thus complete. A consequence of Lemma 12 is that, in order to obtain convergence guarantees for general distributions, one can view $1 - \alpha_1$ as controlling the mismatch between $Q_0$ and $Q_1$ (in terms of the Wasserstein distance), and $1 - \alpha_t$, $\forall t \ge 2$ as controlling the mismatch between $Q_1$ and $\widehat{P}_1'$ (in terms of the KL-divergence).

## F  AUXILIARY PROOFS FOR THEOREM 1 AND COROLLARY 1

In this section, we provide the proofs for those auxiliary lemmas in the proof of Theorem 1 and Corollary 1.

### F.1  PROOF OF LEMMA 3

First, note that
$$q_T(x_T) = \mathbb{E}_{X_0 \sim Q_0}[q_{T|0}(x_T | X_0)].$$

Also note that the function $f(x) = x \log(x)$ is convex. Thus, by Jensen's inequality,

$$
\begin{aligned}
\mathbb{E}_{X_T \sim Q_T}[\log q_T(X_T)] &= \int \mathbb{E}_{X_0 \sim Q_0}[q_{T|0}(x_T | X_0)] \log \mathbb{E}_{X_0 \sim Q_0}[q_{T|0}(x_T | X_0)] \mathrm{d}x_T \\
&\le \int \mathbb{E}_{X_0 \sim Q_0} \left[ q_{T|0}(x_T | X_0) \log q_{T|0}(x_T | X_0) \right] \mathrm{d}x_T \\
&= \mathbb{E}_{X_0 \sim Q_0} \left[ \int q_{T|0}(x_T | X_0) \log q_{T|0}(x_T | X_0) \mathrm{d}x_T \right].
\end{aligned}
$$

Since $Q_{T|0}$ is conditional Gaussian $\mathcal{N}(\sqrt{\bar{\alpha}_T} x_0, (1 - \bar{\alpha}_T) I_d)$, its negative conditional entropy equals

$$\int q_{T|0}(x_T | x_0) \log q_{T|0}(x_T | x_0) \mathrm{d}x_T = -\frac{d}{2} - \frac{d}{2} \log(2\pi(1 - \bar{\alpha}_T))$$

for any $x_0 \in \mathbb{R}^d$. On the other hand, since $P_T' = \mathcal{N}(0, I_d)$,

$$\mathbb{E}_{X_T \sim Q_T}[\log p_T'(X_T)] = -\frac{d}{2} \log(2\pi) - \frac{1}{2} \mathbb{E}_{X_T \sim Q_T} \|X_T\|^2$$

where

$$
\begin{aligned}
\mathbb{E}_{X_T \sim Q_T} \|X_T\|^2 &= \bar{\alpha}_T \mathbb{E}_{X_0 \sim Q_0} \|X_0\|^2 + (1 - \bar{\alpha}_T) \mathbb{E}_{\bar{W}_T \sim \mathcal{N}(0, I_d)} \|\bar{W}_T\|^2 \\
&= \bar{\alpha}_T \mathbb{E}_{X_0 \sim Q_0} \|X_0\|^2 + (1 - \bar{\alpha}_T) d.
\end{aligned}
$$

Putting the two together,

$$
\begin{aligned}
\mathbb{E}_{X_T \sim Q_T} \left[ \log \frac{q_T(X_T)}{p_T'(X_T)} \right] &= \mathbb{E}_{X_T \sim Q_T}[\log q_T(X_T)] - \mathbb{E}_{X_T \sim Q_T}[\log p_T'(X_T)] \\
&\le -\frac{d}{2} - \frac{d}{2} \log(2\pi(1 - \bar{\alpha}_T)) + \frac{d}{2} \log(2\pi) + \frac{1}{2} \left( \bar{\alpha}_T \mathbb{E}_{X_0 \sim Q_0} \|X_0\|^2 + (1 - \bar{\alpha}_T) d \right) \\
&= \frac{1}{2} \bar{\alpha}_T \mathbb{E}_{X_0 \sim Q_0} \|X_0\|^2 - \frac{d \bar{\alpha}_T}{2} - \frac{d}{2} \log(1 - \bar{\alpha}_T).
\end{aligned}
$$

When $T$ is large (and thus when $\bar{\alpha}_T$ is small), the Taylor expansion w.r.t. $\bar{\alpha}_T$ around 0 yields

$$\log(1 - \bar{\alpha}_T) = -\bar{\alpha}_T + O\left(\bar{\alpha}_T^2\right).$$

Therefore,

$$\mathbb{E}_{X_T \sim Q_T} \left[ \log \frac{q_T(X_T)}{p'_T(X_T)} \right] \leq \frac{1}{2} \bar{\alpha}_T \mathbb{E}_{X_0 \sim Q_0} \|X_0\|^2 - \frac{d\bar{\alpha}_T}{2} - \frac{d}{2}(-\bar{\alpha}_T) + O\left(\bar{\alpha}_T^2\right)$$

$$\leq \frac{1}{2} \bar{\alpha}_T M_2 d + O\left(\bar{\alpha}_T^2\right).$$

### F.2 PROOF OF LEMMA 4

To start, note that both $P'_{t-1|t}$ and $\widehat{P}'_{t-1|t}$ are Gaussian (yet having different mean *and variance*). Thus, for each $t = 1, \ldots, T$,

$$\log \frac{p'_{t-1|t}(x_{t-1}|x_t)}{\widehat{p}'_{t-1|t}(x_{t-1}|x_t)}$$

$$= \log\left(\det(\Sigma_t)^{-\frac{1}{2}}\right) - \log\left(\det(\widehat{\Sigma}_t)^{-\frac{1}{2}}\right)$$

$$- \frac{1}{2}(x_{t-1} - \mu_t)^\intercal \Sigma_t^{-1}(x_{t-1} - \mu_t) + \frac{1}{2}(x_{t-1} - \widehat{\mu}_t)^\intercal \widehat{\Sigma}_t^{-1}(x_{t-1} - \widehat{\mu}_t)$$

$$= \frac{1}{2}\left(\log(\det(\widehat{\Sigma}_t)) - \log(\det(\Sigma_t))\right) + \frac{1}{2}(x_{t-1} - \mu_t)^\intercal (\widehat{\Sigma}_t^{-1} - \Sigma_t^{-1})(x_{t-1} - \mu_t)$$

$$+ \frac{1}{2}(x_{t-1} - \widehat{\mu}_t)^\intercal \widehat{\Sigma}_t^{-1}(x_{t-1} - \widehat{\mu}_t) - \frac{1}{2}(x_{t-1} - \mu_t)^\intercal \widehat{\Sigma}_t^{-1}(x_{t-1} - \mu_t)$$

$$= \frac{1}{2}\left(\log(\det(\widehat{\Sigma}_t)) - \log(\det(\Sigma_t))\right) + \frac{1}{2}(x_{t-1} - \mu_t)^\intercal (\widehat{\Sigma}_t^{-1} - \Sigma_t^{-1})(x_{t-1} - \mu_t)$$

$$+ \frac{1}{2}(\mu_t - \widehat{\mu}_t)^\intercal \widehat{\Sigma}_t^{-1}(x_{t-1} - \mu_t) + \frac{1}{2}(x_{t-1} - \mu_t)^\intercal \widehat{\Sigma}_t^{-1}(\mu_t - \widehat{\mu}_t) + \frac{1}{2}(\mu_t - \widehat{\mu}_t)^\intercal \widehat{\Sigma}_t^{-1}(\mu_t - \widehat{\mu}_t).$$

$$(27)$$

There are five terms in (27). We first consider the third and the fourth term, for which we have

$$\mathbb{E}_{X_{t-1} \sim Q_{t-1|t}} \left[ (\mu_t - \widehat{\mu}_t)^\intercal \widehat{\Sigma}_t^{-1}(X_{t-1} - \mu_t) \right] = (\mu_t - \widehat{\mu}_t)^\intercal \widehat{\Sigma}_t^{-1} \mathbb{E}_{X_{t-1} \sim Q_{t-1|t}} \left[ X_{t-1} - \mu_t \right] = 0,$$

$$\mathbb{E}_{X_{t-1} \sim Q_{t-1|t}} \left[ (X_{t-1} - \mu_t)^\intercal \widehat{\Sigma}_t^{-1}(\mu_t - \widehat{\mu}_t) \right] = \mathbb{E}_{X_{t-1} \sim Q_{t-1|t}} \left[ X_{t-1} - \mu_t \right]^\intercal \widehat{\Sigma}_t^{-1}(\mu_t - \widehat{\mu}_t) = 0.$$

Now consider the expectation of the last term in (27). From the definition of $\widehat{\Sigma}_t$ in (6), for small $1 - \alpha_t$ we have $\widehat{\Sigma}_t \succ 0$, and we can define $\widehat{B}_t := I_d - (I_d + (1 - \alpha_t)H_t)^{-1}$, and thus $\widehat{\Sigma}_t^{-1} = \frac{\alpha_t}{1 - \alpha_t}(I_d - \widehat{B}_t)$. From Taylor expansion, we have $\widehat{B}_t = (1 - \alpha_t)H_t + \tilde{O}_{\mathcal{L}^p(Q_t)}((1 - \alpha_t)^2)$. Thus, for each $t \geq 1$,

$$\mathbb{E}_{X_t \sim Q_t} \left[ (\mu_t(X_t) - \widehat{\mu}_t(X_t))^\intercal \widehat{\Sigma}_t^{-1}(X_t)(\mu_t(X_t) - \widehat{\mu}_t(X_t)) \right]$$

$$= (1 - \alpha_t)\mathbb{E}_{X_t \sim Q_t} \left[ (s_t(X_t) - \nabla \log q_t(X_t))^\intercal (I_d - \widehat{B}_t(X_t))(s_t(X_t) - \nabla \log q_t(X_t)) \right]$$

$$= (1 - \alpha_t)\mathbb{E}_{X_t \sim Q_t} \left[ (s_t(X_t) - \nabla \log q_t(X_t))^\intercal (I_d + (1 - \alpha_t)H_t(X_t))^{-1}(s_t(X_t) - \nabla \log q_t(X_t)) \right]$$

$$\lesssim (1 - \alpha_t)\mathbb{E}_{X_t \sim Q_t} \|s_t(X_t) - \nabla \log q_t(X_t)\|^2$$

where the last line follows from the regularity condition on $H_t$ in Assumption 3. Therefore, the expectation of the last term in (27) can be bounded as

$$\sum_{t=1}^T \mathbb{E}_{X_t \sim Q_t} \left[ (\mu_t(X_t) - \widehat{\mu}_t(X_t))^\intercal \widehat{\Sigma}_t^{-1}(X_t)(\mu_t(X_t) - \widehat{\mu}_t(X_t)) \right]$$

$$\lesssim \sum_{t=1}^T (1 - \alpha_t)\mathbb{E}_{X_t \sim Q_t} \|s_t(X_t) - \nabla \log q_t(X_t)\|^2$$

$$\lesssim (\log T)\varepsilon^2,$$

$$(28)$$

where the last line follows by the score estimation error in Assumption 3.

Next we turn to the first two terms in (27). First, note that for all $i, j \in [d]$, we have $(1-\alpha_t)H_t^{ij}(X_t) = \tilde{O}_{\mathcal{L}^p(Q_t)}(1 - \alpha_t)$ under Assumption 3. Now, the first term of (27) is given by

$$\log(\det(\widehat{\Sigma}_t)) - \log(\det(\Sigma_t)) = \log(\det(I_d + (1 - \alpha_t)H_t)) - \log(\det(I_d + (1 - \alpha_t)\nabla^2 \log q_t(x_t))).$$

When $(1 - \alpha_t)$ is small, we can use Taylor expansion for the functions $\det(\cdot)$ and $\log(\cdot)$ to get

$$\log(\det(I_d + (1 - \alpha_t)H_t))$$
$$= \log\left(1 + (1 - \alpha_t)\text{Tr}(H_t) + \frac{(1 - \alpha_t)^2}{2}(\text{Tr}(H_t)^2 - \text{Tr}(H_t^2)) + \tilde{O}_{\mathcal{L}^p(Q_t)}((1 - \alpha_t)^3)\right)$$
$$= (1 - \alpha_t)\text{Tr}(H_t) + \frac{(1 - \alpha_t)^2}{2}(\text{Tr}(H_t)^2 - \text{Tr}(H_t^2)) - \frac{(1 - \alpha_t)^2}{2}\text{Tr}(H_t)^2 + \tilde{O}_{\mathcal{L}^p(Q_t)}((1 - \alpha_t)^3)$$
$$= (1 - \alpha_t)\text{Tr}(H_t) - \frac{(1 - \alpha_t)^2}{2}\text{Tr}(H_t^2) + \tilde{O}_{\mathcal{L}^p(Q_t)}((1 - \alpha_t)^3).$$

Similar expression can be obtained for $\log(\det(I_d + (1 - \alpha_t)\nabla^2 \log q_t(x_t)))$. Thus, the first term in (27) is equal to

$$\log(\det(\widehat{\Sigma}_t)) - \log(\det(\Sigma_t))$$
$$= (1 - \alpha_t)\left(\text{Tr}(H_t) - \text{Tr}(\nabla^2 \log q_t(x_t))\right) - \frac{(1 - \alpha_t)^2}{2}\left[\text{Tr}(H_t^2) - \text{Tr}((\nabla^2 \log q_t(x_t))^2)\right]$$
$$+ \tilde{O}_{\mathcal{L}^p(Q_t)}((1 - \alpha_t)^3).$$

For the second term in (27), we first take expectation over $x_{t-1}$ and get

$$\mathbb{E}_{X_{t-1} \sim Q_{t-1|t}}\left[(X_{t-1} - \mu_t)^\intercal(\widehat{\Sigma}_t^{-1} - \Sigma_t^{-1})(X_{t-1} - \mu_t)\right] = \text{Tr}\left((\widehat{\Sigma}_t^{-1} - \Sigma_t^{-1})\Sigma_t\right).$$

To proceed, note that

$$(I_d + (1 - \alpha_t)H_t)^{-1} \overset{(iii)}{=} I_d - (1 - \alpha_t)H_t + (1 - \alpha_t)^2 H_t^2 + \tilde{O}_{\mathcal{L}^p(Q_t)}((1 - \alpha_t)^3). \tag{29}$$

To see $(iii)$, we write $S_t$ as the true inverse of $I_d + (1 - \alpha_t)H_t$. Its existence is guaranteed if $(1 - \alpha_t)$ is small. Since

$$(I_d + (1 - \alpha_t)H_t)(I_d - (1 - \alpha_t)H_t + (1 - \alpha_t)^2 H_t^2) = I_d + \tilde{O}_{\mathcal{L}^p(Q_t)}((1 - \alpha_t)^3),$$

we have

$$(I_d + (1 - \alpha_t)H_t)(I_d - (1 - \alpha_t)H_t + (1 - \alpha_t)^2 H_t^2 - S_t) = \tilde{O}_{\mathcal{L}^p(Q_t)}((1 - \alpha_t)^3)$$

which implies that $S_t = I_d - (1 - \alpha_t)H_t + (1 - \alpha_t)^2 H_t^2 + \tilde{O}_{\mathcal{L}^p(Q_t)}((1 - \alpha_t)^3)$. This shows the validity of $(iii)$. Therefore,

$$\text{Tr}\left((\widehat{\Sigma}_t^{-1} - \Sigma_t^{-1})\Sigma_t\right) = \text{Tr}(\widehat{\Sigma}_t^{-1}\Sigma_t - I_d)$$
$$= \text{Tr}\Bigg(\left[I_d - (1 - \alpha_t)H_t + (1 - \alpha_t)^2 H_t^2 + \tilde{O}_{\mathcal{L}^p(Q_t)}((1 - \alpha_t)^3)\right]$$
$$\left[I_d + (1 - \alpha_t)\nabla^2 \log q_t(x_t)\right] - I_d\Bigg)$$
$$= (1 - \alpha_t)\left[\text{Tr}(\nabla^2 \log q_t(x_t)) - \text{Tr}(H_t)\right]$$
$$+ (1 - \alpha_t)^2\left[\text{Tr}(H_t^2) - \text{Tr}(H_t \nabla^2 \log q_t(x_t))\right] + \tilde{O}_{\mathcal{L}^p(Q_t)}((1 - \alpha_t)^3).$$

Adding this to the first term of (27) and taking expectation over $X_t \sim Q_t$ (noting Assumption 5 here), we get

$$\mathbb{E}_{X_{t-1}, X_t \sim Q_{t-1,t}}\Bigg[\left(\log(\det(\widehat{\Sigma}_t(X_t))) - \log(\det(\Sigma_t(X_t)))\right)$$
$$+ (X_{t-1} - \mu_t(X_t))^\intercal(\widehat{\Sigma}_t^{-1}(X_t) - \Sigma_t^{-1}(X_t))(X_{t-1} - \mu_t(X_t))\Bigg]$$
$$= \frac{(1 - \alpha_t)^2}{2}\mathbb{E}_{X_t \sim Q_t}\left[\text{Tr}(H_t(X_t)^2) - 2\text{Tr}(H_t(X_t)\nabla^2 \log q_t(X_t)) + \text{Tr}((\nabla^2 \log q_t(X_t))^2)\right]$$

$$+ \tilde{O}((1 - \alpha_t)^3)$$

$$\overset{(iv)}{=} \frac{(1 - \alpha_t)^2}{2} \mathbb{E}_{X_t \sim Q_t} \left\| H_t(X_t) - \nabla^2 \log q_t(X_t) \right\|_F^2 + \tilde{O}((1 - \alpha_t)^3),$$

where $(iv)$ follows because for two symmetric matrices $A$ and $B$,

$$\mathrm{Tr}(A^2) - 2\mathrm{Tr}(AB) + \mathrm{Tr}(B^2) = \mathrm{Tr}(A^2) - \mathrm{Tr}(AB) - \mathrm{Tr}(BA) + \mathrm{Tr}(B^2)$$
$$= \mathrm{Tr}((A - B)(A - B)) = \mathrm{Tr}((A - B)^\mathsf{T}(A - B)) = \|A - B\|_F^2 \, .$$

Thus, following from Assumption 3,

$$\sum_{t=1}^{T} \mathbb{E}_{X_{t-1}, X_t \sim Q_{t-1,t}} \Big[ \Big( \log(\det(\widehat{\Sigma}_t(X_t))) - \log(\det(\Sigma_t(X_t))) \Big)$$

$$+ (X_{t-1} - \mu_t(X_t))^\mathsf{T}(\widehat{\Sigma}_t^{-1}(X_t) - \Sigma_t^{-1}(X_t))(X_{t-1} - \mu_t(X_t)) \Big] \lesssim \frac{\log^2 T}{T} \varepsilon_H^2. \quad (30)$$

Here $\varepsilon_H$ is the Hessian estimation error. Combining (28) and (30) yields the desired result for the accelerated estimation error, which is in the order $\tilde{O}(1/T^2)$.

### F.3 PROOF OF COROLLARY 2

Given the perturbed $\tilde{\Sigma}_t$ in (9), following the definition in (14), we define, $\forall p \geq 1$,

$$\tilde{A}_t := (1 - \alpha_t)\nabla^2 \log q_t(x_t) + \frac{(1 - \alpha_t)^2}{4}(\nabla^2 \log q_t(x_t))^2$$

$$= (1 - \alpha_t)\left( \nabla^2 \log q_t(x_t) + \frac{1 - \alpha_t}{4}\nabla^2 \log q_t(x_t) \right),$$

$$\tilde{B}_t := I_d - \frac{1 - \alpha_t}{\alpha_t}\tilde{\Sigma}_t^{-1} = I_d - \tilde{A}_t + \tilde{A}_t^2 + \tilde{O}_{\mathcal{L}^p(Q_t)}((1 - \alpha_t)^3)$$

$$\tilde{H}_t := H_t + \frac{1 - \alpha_t}{4}H_t.$$

Note that under Assumption 3,

$$(1 - \alpha_t)\left\| \tilde{H}_t \right\| \lesssim (1 - \alpha_t)\|H_t\| + (1 - \alpha_t)^2\|H_t\|^2 = \tilde{O}_{\mathcal{L}^r(Q_t)}(1 - \alpha_t), \ \forall r \geq 1.$$

Then, the rest of the proof Lemma 4 still holds with $\nabla^2 \log q_t(x_t)$ and $H_t$ replaced by $\nabla^2 \log q_t(x_t) + \frac{1 - \alpha_t}{4}\nabla^2 \log q_t(x_t)$ and $\tilde{H}_t$. The proof is complete by noting that

$$\mathbb{E}_{X_t \sim Q_t} \left\| \tilde{H}_t(X_t) - \left( \nabla^2 \log q_t(X_t) + \frac{1 - \alpha_t}{4}\nabla^2 \log q_t(X_t) \right) \right\|_F^2$$

$$\lesssim (1 + (1 - \alpha_t))\mathbb{E}_{X_t \sim Q_t} \left\| H_t(X_t) - \nabla^2 \log q_t(X_t) \right\|_F^2$$

$$\lesssim \varepsilon_H^2.$$

### F.4 PROOF OF LEMMA 5

By Bayes' rule, for any $x_{t-1}$ given fixed $x_t$, we have

$q_{t-1|t}(x_{t-1}|x_t)$

$$\propto q_{t-1}(x_{t-1})\exp\left( -\frac{\left\| x_t - \sqrt{\alpha_t}x_{t-1} \right\|^2}{2(1 - \alpha_t)} \right)$$

$$\propto q_{t-1}(x_{t-1})p'_{t-1|t}(x_{t-1}|x_t)\exp\left( \frac{1}{2}(x_{t-1} - \mu_t)^\mathsf{T}\Sigma_t^{-1}(x_{t-1} - \mu_t) - \frac{\left\| x_{t-1} - x_t/\sqrt{\alpha_t} \right\|^2}{2(1 - \alpha_t)/\alpha_t} \right)$$

$$= q_{t-1}(x_{t-1})p'_{t-1|t}(x_{t-1}|x_t)\exp\left( \frac{\alpha_t}{2(1 - \alpha_t)}(x_{t-1} - \mu_t)^\mathsf{T}(I_d - B_t)(x_{t-1} - \mu_t) - \frac{\left\| x_{t-1} - x_t/\sqrt{\alpha_t} \right\|^2}{2(1 - \alpha_t)/\alpha_t} \right)$$

(by Equation (14))

$$\propto p'_{t-1|t}(x_{t-1}|x_t)\exp\left(\zeta_{t,t-1}(x_t,x_{t-1})-\frac{\alpha_t}{2(1-\alpha_t)}(x_{t-1}-\mu_t)^\intercal B_t(x_{t-1}-\mu_t)\right),$$

where the last line follows from the definition of $\zeta_{t,t-1}(x_t,x_{t-1})$ in (16). Now, with the definition of $\zeta'_{t,t-1}(x_t,x_{t-1})$ in (17), we have

$$q_{t-1|t}(x_{t-1}|x_t)=\frac{p'_{t-1|t}(x_{t-1}|x_t)e^{\zeta'_{t,t-1}(x_t,x_{t-1})}}{\mathbb{E}_{X_{t-1}\sim P'_{t-1|t}}[e^{\zeta'_{t,t-1}(x_t,X_{t-1})}]}.$$

## F.5 PROOF OF LEMMA 6

Recall Equation (2). Let $\tilde{Q}_0$ denote the distribution of $\sqrt{\bar{\alpha}_t}x_0$, and let $g(z)$ denote the p.d.f. (w.r.t. the Lebesgue measure) of the distribution of $\sqrt{1-\bar{\alpha}_t}\bar{w}_t$. Note that $g$ is a scaled version of the unit Gaussian p.d.f., and $\int_{z\in\mathbb{R}^d}g(z)\mathrm{d}z=1<\infty$. Now, for any event $A\subseteq\mathcal{B}(\lambda)$,

$$Q_t(A)=\int_{x\in A}\int_{\tilde{x}_0\in\mathbb{R}^d}g(x-\tilde{x}_0)\mathrm{d}\tilde{Q}_0(\tilde{x}_0)\mathrm{d}x=\int_{\tilde{x}_0\in\mathbb{R}^d}\left(\int_{x\in A}g(x-\tilde{x}_0)\mathrm{d}x\right)\mathrm{d}\tilde{Q}_0(\tilde{x}_0)$$

by Fubini's theorem. If $A$ has Lebesgue measure 0, by continuity of $g(x)$ we get $\int_{x\in A}g(x-\tilde{x}_0)\mathrm{d}x=0$, and thus $Q_t(A)=0$. This shows that $Q_t$ is absolutely continuous w.r.t. the Lebesgue measure, and its p.d.f. exists, denoted as $q_t$.

Now, since any order of derivative of the Gaussian p.d.f. is bounded away from infinity and $\tilde{Q}_0$ is a probability measure, we can invoke the dominated convergence theorem here to change the order of derivative and integral as

$$\partial_{\boldsymbol{a}}^k q_t(x)=\partial_{\boldsymbol{a}}^k\int_{\tilde{x}_0\in\mathbb{R}^d}g(x-\tilde{x}_0)\mathrm{d}\tilde{Q}_0(\tilde{x}_0)=\int_{\tilde{x}_0\in\mathbb{R}^d}\partial_{\boldsymbol{a}}^k g(x-\tilde{x}_0)\mathrm{d}\tilde{Q}_0(\tilde{x}_0).\tag{31}$$

Thus, for any $k\geq 1$ and any vector of indices $\boldsymbol{a}\in[d]^k$, we have

$$\left|\partial_{\boldsymbol{a}}^k q_t(x)\right|\leq\sup_{x\in\mathbb{R}^d}\left|\partial_{\boldsymbol{a}}^k g(x)\right|\int_{\tilde{x}_0\in\mathbb{R}^d}\mathrm{d}\tilde{Q}_0(\tilde{x}_0)=\sup_{x\in\mathbb{R}^d}\left|\partial_{\boldsymbol{a}}^k g(x)\right|<\infty.$$

This also implies that the Taylor term $|T_k(q_t,x,\mu)|<\infty$ for any $x$ and $\mu$, and

$$q_t(x)=\int_{\tilde{x}_0\in\mathbb{R}^d}g(x-\tilde{x}_0)\mathrm{d}\tilde{Q}_0(\tilde{x}_0)\overset{(i)}{=}\int_{\tilde{x}_0\in\mathbb{R}^d}\lim_{p\to\infty}\sum_{k=0}^p T_k(g(x-\tilde{x}_0),x,\mu)\mathrm{d}\tilde{Q}_0(\tilde{x}_0)$$

$$\overset{(ii)}{=}\lim_{p\to\infty}\int_{\tilde{x}_0\in\mathbb{R}^d}\sum_{k=0}^p T_k(g(x-\tilde{x}_0),x,\mu)\mathrm{d}\tilde{Q}_0(\tilde{x}_0)$$

$$\overset{(iii)}{=}\lim_{p\to\infty}\sum_{k=0}^p T_k(q_t,x,\mu)$$

where $(i)$ follows because (scaled) Gaussian density is analytic, $(ii)$ follows from dominated convergence theorem and the fact that $g$ is a Gaussian density and has an upper bound independent of $\tilde{x}_0$, and $(iii)$ follows from (31). This shows that $q_t$ is analytic.

Finally, since $\partial_{\boldsymbol{a}}^k\log q_t$ is a smooth function of $q_t,\partial^1 q_t,\ldots,\partial^k q_t$, we have $\partial_{\boldsymbol{a}}^k\log q_t(x_t)<\infty$ (possibly depending on $T$) for all $k\geq 1$ and fixed (finite) $x_t\in\mathbb{R}^d$. Also, $\log q_t$ is analytic because $\log(\cdot)$ is analytic and $q_t(x_t)>0,\forall x_t\in\mathbb{R}^d$.

## F.6 PROOF OF LEMMA 7

The result follows directly from Isserlis's Theorem, which says that

$$\mathbb{E}\left[\prod_{i=1}^n Z_i\right]=\sum_{p\in P_n^2}\prod_{\{i,j\}\in p}\mathbb{E}[Z_iZ_j]=\sum_{p\in P_n^2}\prod_{\{i,j\}\in p}\mathrm{Cov}(Z_i,Z_j)$$

since each $Z_i$ is centered. Here $P_n^2$ is the set that contains all distinct size-2 partitions of $[n]$. For example, $P_4^2 = \{(\{1,2\},\{3,4\}),(\{1,3\},\{2,4\}),(\{1,4\},\{2,3\})\}$. Thus, since $A_t = \tilde{O}_{\mathcal{L}^p(Q)}(1 - \alpha_t)$ under Assumption 5,

$$\mathbb{E}\left[\prod_{i=1}^n Z_i\right] = 0, \text{ if } n \text{ is odd}$$

$$\mathbb{E}\left[\prod_{i=1}^n Z_i\right] = \tilde{O}_{\mathcal{L}^p(Q_t)}\left(\left(\frac{1-\alpha_t}{\alpha_t}\right)^{\frac{n}{2}}\right) = \tilde{O}_{\mathcal{L}^p(Q_t)}\left((1-\alpha_t)^{\frac{n}{2}}\right), \text{ if } n \text{ is even.}$$

More specifically, following from Isserlis's Theorem, the fourth moment is

$$\mathbb{E}[Z_i Z_j Z_k Z_l] = \text{Cov}(Z_i, Z_j)\text{Cov}(Z_k, Z_l) +$$
$$\text{Cov}(Z_i, Z_k)\text{Cov}(Z_j, Z_l) + \text{Cov}(Z_i, Z_l)\text{Cov}(Z_j, Z_k), \; \forall i, j, k, l \in [d].$$

Here $\text{Cov}(Z_i, Z_j) = \frac{1-\alpha_t}{\alpha_t}(\mathbb{1}\{i = j\} + (1-\alpha_t)A^{ij})$. The fourth moment result follows immediately by plugging into the formula. Turning to the sixth moment, we note that we are interested only in the coefficients for the terms that grow at a rate $\tilde{O}_{\mathcal{L}^p(Q_t)}((1-\alpha_t)^3)$. Since the sixth moment consists of sum of product terms in which three covariance matrices are multiplied (giving us a rate at least $\tilde{O}_{\mathcal{L}^p(Q_t)}((1-\alpha_t)^3)$), at least one product term in the sum must take covariance values only on the diagonal of the matrix. Therefore, only $\mathbb{E}[Z_i^6]$, $\mathbb{E}[Z_i^4 Z_j^2]$, and $\mathbb{E}[Z_i^2 Z_j^2 Z_k^2]$ with $i, j, k$ all differ satisfy this requirement, and we immediately get the desired result from Isserlis's Theorem.

### F.7 Proof of Lemma 8

We first fix $x_t$ and will take expectation at the end. Note that $q_{t|t-1}(x_t|x_{t-1}) = \frac{1}{(2\pi(1-\alpha_t))^{d/2}}\exp\left(-\frac{\|x_t - \sqrt{\alpha_t}x_{t-1}\|^2}{2(1-\alpha_t)}\right)$. Following from the idea of Tweedie Efron (2011), we have

$$q_{t-1|t}(x_{t-1}|x_t)$$
$$= \frac{q_{t-1}(x_{t-1})}{q_t(x_t)}q_{t|t-1}(x_t|x_{t-1})$$
$$= \frac{q_{t-1}(x_{t-1})}{q_t(x_t)}q_{t|t-1}(x_t|0)\exp\left(\frac{\sqrt{\alpha_t}}{1-\alpha_t}x_t^\mathsf{T}x_{t-1} - \frac{\alpha_t}{2(1-\alpha_t)}\|x_{t-1}\|^2\right)$$
$$= \left(q_{t-1}(x_{t-1})e^{-\frac{\alpha_t}{2(1-\alpha_t)}\|x_{t-1}\|^2}\right)\exp\left(\frac{\sqrt{\alpha_t}}{1-\alpha_t}x_t^\mathsf{T}x_{t-1} - \log q_t(x_t) + \log q_{t|t-1}(x_t|0)\right)$$
$$=: f(x_{t-1})\exp\left(x_{t-1}^\mathsf{T}\tilde{x}_t - \kappa(\tilde{x}_t)\right) \tag{32}$$

where we have used the definitions of $\tilde{x}_t$ and $\kappa(\tilde{x}_t)$ in (23). This shows that $x_{t-1}$ is a conditional exponential family given $\tilde{x}_t$. Thus, the first moment can be found as (cf. Prop. 11.1 in Moulin & Veeravalli (2018))

$$0 = \nabla_{\tilde{x}_t}\int q_{t-1|t}(x_{t-1}|x_t)\mathrm{d}x_{t-1} = \nabla_{\tilde{x}_t}\int f(x_{t-1})\exp\left(x_{t-1}^\mathsf{T}\tilde{x}_t - \kappa(\tilde{x}_t)\right)\mathrm{d}x_{t-1}$$
$$= \int f(x_{t-1})\nabla_{\tilde{x}_t}\exp\left(x_{t-1}^\mathsf{T}\tilde{x}_t - \kappa(\tilde{x}_t)\right)\mathrm{d}x_{t-1}$$
$$= \int f(x_{t-1})\exp\left(x_{t-1}^\mathsf{T}\tilde{x}_t - \kappa(\tilde{x}_t)\right)(x_{t-1} - \nabla_{\tilde{x}_t}\kappa(\tilde{x}_t))\mathrm{d}x_{t-1}$$
$$= \int f(x_{t-1})\exp\left(x_{t-1}^\mathsf{T}\tilde{x}_t - \kappa(\tilde{x}_t)\right)x_{t-1}\mathrm{d}x_{t-1} - \nabla_{\tilde{x}_t}\kappa(\tilde{x}_t)$$

which implies that

$$\mathbb{E}_{X_{t-1}\sim Q_{t-1|t}}[X_{t-1}] = \nabla\kappa. \tag{33}$$

For the second moment,

$$0 = \partial_{ij}^2\int q_{t-1|t}(x_{t-1}|x_t)\mathrm{d}x_{t-1}$$

$$= \int f(x_{t-1}) \frac{\partial}{\partial \tilde{x}_t^j} \Big( \exp \big( x_{t-1}^\mathsf{T} \tilde{x}_t - \kappa(\tilde{x}_t) \big) \big( x_{t-1}^i - \partial_i \kappa(\tilde{x}_t) \big) \Big) \mathrm{d}x_{t-1}$$

$$= \int f(x_{t-1}) \exp \big( x_{t-1}^\mathsf{T} \tilde{x}_t - \kappa(\tilde{x}_t) \big) \Big( (x_{t-1}^i - \partial_i \kappa(\tilde{x}_t))(x_{t-1}^j - \partial_j \kappa(\tilde{x}_t)) - \partial_{ij}^2 \kappa(\tilde{x}_t) \Big) \mathrm{d}x_{t-1}$$

which yields

$$\mathbb{E}_{X_{t-1} \sim Q_{t-1|t}} \big[ (X_{t-1} - \mu_t)(X_{t-1} - \mu_t)^\mathsf{T} \big] = \nabla^2 \kappa = \frac{1 - \alpha_t}{\alpha_t} I_d + \frac{(1 - \alpha_t)^2}{\alpha_t} \nabla^2 \log q_t(x_t). \quad (34)$$

Below, we write $x = x_{t-1}$ and $\kappa = \kappa(\tilde{x}_t)$ for brevity. We remind readers that all derivatives are w.r.t. $\tilde{x}_t$ instead of $x = x_{t-1}$. For the third moment,

$$0 = \partial_{ijk}^3 \int q_{t-1|t} \mathrm{d}x =: \int f(x) \exp \big( x^\mathsf{T} \tilde{x}_t - \kappa \big) D_3(x, \tilde{x}_t) \mathrm{d}x$$

where

$$\begin{aligned}
D_3(x, \tilde{x}_t) &= \exp \big( -x^\mathsf{T} \tilde{x}_t + \kappa \big) \partial_k \Big( \exp \big( x^\mathsf{T} \tilde{x}_t - \kappa \big) \big( (x^i - \partial_i \kappa)(x^j - \partial_j \kappa) - \partial_{ij}^2 \kappa \big) \Big) \\
&= (x^k - \partial_k \kappa) \big( (x^i - \partial_i \kappa)(x^j - \partial_j \kappa) - \partial_{ij}^2 \kappa \big) \\
&\quad + (-\partial_{ik}^2 \kappa)(x^j - \partial_j \kappa) + (-\partial_{jk}^2 \kappa)(x^i - \partial_i \kappa) - \partial_{ijk}^3 \kappa. \quad (35)
\end{aligned}$$

Now, for any function $\mathrm{fn}(\tilde{x}_t)$ and $1 \le i \le d$,

$$\int f(x) \exp \big( x^\mathsf{T} \tilde{x}_t - \kappa \big) \mathrm{fn}(\tilde{x}_t)(x^i - \partial_i \kappa) \mathrm{d}x = 0$$

by the first moment result (33). Thus, we get

$$\mathbb{E}_{X_{t-1} \sim Q_{t-1|t}} \Big[ (X_{t-1}^i - \mu_t^i)(X_{t-1}^j - \mu_t^j)(X_{t-1}^k - \mu_t^k) \Big] = \partial_{ijk}^3 \kappa,$$

and by Assumption 5, $\mathbb{E}_{X_t \sim Q_t}[\partial_{ijk}^3 \kappa] = \tilde{O}((1 - \alpha_t)^3)$.

For the fourth moment, we have

$$0 = \partial_{ijkl}^4 \int q_{t-1|t} \mathrm{d}x =: \int f(x) \exp \big( x^\mathsf{T} \tilde{x}_t - \kappa \big) D_4(x, \tilde{x}_t) \mathrm{d}x$$

where

$$\begin{aligned}
D_4(x, \tilde{x}_t) &= \exp \big( -x^\mathsf{T} \tilde{x}_t + \kappa \big) \partial_l \Big( \exp \big( x^\mathsf{T} \tilde{x}_t - \kappa \big) \big( (x^i - \partial_i \kappa)(x^j - \partial_j \kappa)(x^k - \partial_k \kappa) \\
&\quad - \partial_{ij}^2 \kappa (x^k - \partial_k \kappa) - \partial_{ik}^2 \kappa (x^j - \partial_j \kappa) - \partial_{jk}^2 \kappa (x^i - \partial_i \kappa) - \partial_{ijk}^3 \kappa \big) \Big) \\
&= (x^i - \partial_i \kappa)(x^j - \partial_j \kappa)(x^k - \partial_k \kappa)(x^l - \partial_l \kappa) + \partial_l \Big( (x^i - \partial_i \kappa)(x^j - \partial_j \kappa)(x^k - \partial_k \kappa) \Big) \\
&\quad - \partial_{ij}^2 \kappa (x^k - \partial_k \kappa)(x^l - \partial_l \kappa) - \partial_{ijl}^3 \kappa (x^k - \partial_k \kappa) + \partial_{ij}^2 \kappa \partial_{kl}^2 \kappa \\
&\quad - \partial_{ik}^2 \kappa (x^j - \partial_j \kappa)(x^l - \partial_l \kappa) - \partial_{ikl}^3 \kappa (x^j - \partial_j \kappa) + \partial_{ik}^2 \kappa \partial_{jl}^2 \kappa \\
&\quad - \partial_{jk}^2 \kappa (x^i - \partial_i \kappa)(x^l - \partial_l \kappa) - \partial_{jkl}^3 \kappa (x^i - \partial_i \kappa) + \partial_{jk}^2 \kappa \partial_{il}^2 \kappa \\
&\quad - \partial_{ijk}^3 \kappa (x^l - \partial_l \kappa) - \partial_{ijkl}^4 \kappa \quad (36)
\end{aligned}$$

and

$$\begin{aligned}
&\partial_l \Big( (x^i - \partial_i \kappa)(x^j - \partial_j \kappa)(x^k - \partial_k \kappa) \Big) \\
&= -\partial_{il}^2 \kappa (x^j - \partial_j \kappa)(x^k - \partial_k \kappa) - \partial_{jl}^2 \kappa (x^i - \partial_i \kappa)(x^k - \partial_k \kappa) - \partial_{kl}^2 \kappa (x^i - \partial_i \kappa)(x^j - \partial_j \kappa).
\end{aligned}$$

Using the first and second moment results in (33) and (34), we get

$$\begin{aligned}
\mathbb{E}_{X_{t-1} \sim Q_{t-1|t}} \Big[ (X_{t-1}^i - \mu_t^i)(X_{t-1}^j - \mu_t^j)(X_{t-1}^k - \mu_t^k)(X_{t-1}^l - \mu_t^l) \Big] = \\
(\partial_{ij}^2 \kappa)(\partial_{kl}^2 \kappa) + (\partial_{ik}^2 \kappa)(\partial_{jl}^2 \kappa) + (\partial_{il}^2 \kappa)(\partial_{jk}^2 \kappa) + \partial_{ijkl}^4 \kappa.
\end{aligned}$$

And the fourth moment result follows directly by applying (34) to each of the terms and taking the expectation over $X_t \sim Q_t$. The rate follows from Assumption 5 (cf. Definition 2).

### F.8 PROOF OF LEMMA 9

The proof continues the idea of Lemma 8. The idea is to use the inductive relationship (provided in the proof of Lemmas 8 and 10):

$$D_5(x, \tilde{x}_t) = \exp\left(-x^\intercal \tilde{x}_t + \kappa\right) \partial_m\left(\exp\left(x^\intercal \tilde{x}_t - \kappa\right) D_4(x, \tilde{x}_t)\right)$$
$$= (x^m - \partial_m \kappa) D_4(x, \tilde{x}_t) + \partial_m D_4(x, \tilde{x}_t)$$
$$D_6(x, \tilde{x}_t) = \exp\left(-x^\intercal \tilde{x}_t + \kappa\right) \partial_n\left(\exp\left(x^\intercal \tilde{x}_t - \kappa\right) D_5(x, \tilde{x}_t)\right)$$
$$= (x^n - \partial_n \kappa) D_5(x, \tilde{x}_t) + \partial_n D_5(x, \tilde{x}_t).$$

Let $P_\ell^k$ be the set that contains all distinct size-$k$ partitions of $[\ell]$. We use the definitions:

$$\binom{A}{k} := \left\{\{a_1, \ldots, a_k\} : a_1, \ldots, a_k \in A, \ a_1, \ldots, a_k \text{ all differ}\right\}, \ k \leq |A|$$
$$\text{part}_k(A) := \{((a_i, a_j) : \{i, j\} \in p) : p \in P_{|A|}^k\}.$$

Recall the formula for $D_4$ in (36), which can be abbreviated as (here $|\boldsymbol{a}| = 4$):

$$D_4(x, \tilde{x}_t) = \prod_{i \in \boldsymbol{a}}(x^i - \partial_i \kappa) - \sum_{\boldsymbol{b} \in \binom{\boldsymbol{a}}{2}} \partial_{\boldsymbol{b}}^2 \kappa \prod_{i \in \boldsymbol{a} \setminus \boldsymbol{b}}(x^i - \partial_i \kappa) + \sum_{(\boldsymbol{b}, \boldsymbol{c}) \in \text{part}_2(\boldsymbol{a})} \partial_{\boldsymbol{b}}^2 \kappa \partial_{\boldsymbol{c}}^2 \kappa$$
$$- \sum_{i \in \boldsymbol{a}} \partial_{\boldsymbol{a} \setminus \{i\}}^3 \kappa (x^i - \partial_i \kappa) - \partial_{\boldsymbol{a}}^4 \kappa.$$

Also recall the definition of $f(x)$ in Lemma 8 and that $\int f(x) e^{x^\intercal \tilde{x}_t - \kappa} D_p(x, \tilde{x}_t) \mathrm{d}x = 0$, through which we can find the expected $p$-th moments of $\mathbb{E}_{X_{t-1} \sim Q_{t-1|t}}\left[\prod_{i \in \boldsymbol{a}}(X_{t-1}^i - \mu_t^i)\right]$. For reference, the first four moments are

$$\int f(x) \exp\left(x^\intercal \tilde{x}_t - \kappa\right)(x^i - \partial_i \kappa) \mathrm{d}x = 0$$

$$\int f(x) \exp\left(x^\intercal \tilde{x}_t - \kappa\right)(x^i - \partial_i \kappa)(x^j - \partial_j \kappa) \mathrm{d}x = \partial_{ij}^2 \kappa = \tilde{O}_{\mathcal{L}^p(Q_t)}(1 - \alpha_t)$$

$$\int f(x) \exp\left(x^\intercal \tilde{x}_t - \kappa\right)(x^i - \partial_i \kappa)(x^j - \partial_j \kappa)(x^k - \partial_k \kappa) \mathrm{d}x = \partial_{ijk}^3 \kappa = \tilde{O}_{\mathcal{L}^p(Q_t)}((1 - \alpha_t)^3)$$

$$\int f(x) \exp\left(x^\intercal \tilde{x}_t - \kappa\right)(x^i - \partial_i \kappa)(x^j - \partial_j \kappa)(x^k - \partial_k \kappa)(x^l - \partial_l \kappa) \mathrm{d}x$$
$$= (\partial_{ij}^2 \kappa)(\partial_{kl}^2 \kappa) + (\partial_{ik}^2 \kappa)(\partial_{jl}^2 \kappa) + (\partial_{il}^2 \kappa)(\partial_{jk}^2 \kappa) + \partial_{ijkl}^4 \kappa = \tilde{O}_{\mathcal{L}^p(Q_t)}((1 - \alpha_t)^2)$$

where we note that $\partial_{\boldsymbol{a}}^k \kappa = \tilde{O}_{\mathcal{L}^p(Q_t)}((1 - \alpha_t)^k)$ for all $k \geq 3$.

We can calculate $D_5$ as (with $|\boldsymbol{a}| = 5$):

$$D_5(x, \tilde{x}_t) = (x^{a_5} - \partial_{a_5} \kappa) D_4(x, \tilde{x}_t) + \partial_{a_5} D_4(x, \tilde{x}_t)$$
$$= \prod_{i \in \boldsymbol{a}}(x^i - \partial_i \kappa) - \sum_{\boldsymbol{b} \in \binom{\boldsymbol{a}}{2}} \partial_{\boldsymbol{b}}^2 \kappa \prod_{i \in \boldsymbol{a} \setminus \boldsymbol{b}}(x^i - \partial_i \kappa) - \sum_{\boldsymbol{b} \in \binom{\boldsymbol{a}}{2}} \partial_{\boldsymbol{a} \setminus \boldsymbol{b}}^3 \kappa \prod_{i \in \boldsymbol{b}}(x^i - \partial_i \kappa)$$
$$+ \sum_{\substack{i \in \boldsymbol{a} \\ (\boldsymbol{b}, \boldsymbol{c}) \in \text{part}_2(\boldsymbol{a} \setminus \{i\})}} \partial_{\boldsymbol{b}}^2 \kappa \partial_{\boldsymbol{c}}^2 \kappa (x^i - \partial_i \kappa)$$
$$- \sum_{i \in \boldsymbol{a}} \partial_{\boldsymbol{a} \setminus \{i\}}^4 \kappa (x^i - \partial_i \kappa) + \sum_{\boldsymbol{b} \in \binom{\boldsymbol{a}}{2}} \partial_{\boldsymbol{b}}^2 \kappa \partial_{\boldsymbol{a} \setminus \boldsymbol{b}}^3 \kappa - \partial_{\boldsymbol{a}}^5 \kappa.$$

Therefore,

$$\mathbb{E}_{X_{t-1} \sim Q_{t-1|t}}\left[\prod_{i \in \boldsymbol{a} : |\boldsymbol{a}| = 5}(X_{t-1}^i - \mu_t^i)\right]$$

$$= \sum_{\boldsymbol{b} \in \binom{\boldsymbol{a}}{2}} \partial_{\boldsymbol{b}}^2 \kappa \partial_{\boldsymbol{a} \setminus \boldsymbol{b}}^3 \kappa + \sum_{\boldsymbol{b} \in \binom{\boldsymbol{a}}{2}} \partial_{\boldsymbol{a} \setminus \boldsymbol{b}}^3 \kappa \partial_{\boldsymbol{b}}^2 \kappa - \sum_{\boldsymbol{b} \in \binom{\boldsymbol{a}}{2}} \partial_{\boldsymbol{b}}^2 \kappa \partial_{\boldsymbol{a} \setminus \boldsymbol{b}}^3 \kappa + \partial_{\boldsymbol{a}}^5 \kappa$$

$$= \sum_{\boldsymbol{b} \in \binom{\boldsymbol{a}}{2}} \partial_{\boldsymbol{b}}^2 \kappa \partial_{\boldsymbol{a} \setminus \boldsymbol{b}}^3 \kappa + \partial_{\boldsymbol{a}}^5 \kappa = \tilde{O}_{\mathcal{L}^p(Q_t)}((1 - \alpha_t)^4).$$

Now we turn to calculate $D_6$ (and let $|\boldsymbol{a}| = 6$):

$$D_6(x, \tilde{x}_t) = (x^{a_6} - \partial_{a_6} \kappa) D_5(x, \tilde{x}_t) + \partial_{a_6} D_5(x, \tilde{x}_t)$$

$$= \prod_{i \in \boldsymbol{a}} (x^i - \partial_i \kappa) - \sum_{\boldsymbol{b} \in \binom{\boldsymbol{a}}{2}} \partial_{\boldsymbol{b}}^2 \kappa \prod_{i \in \boldsymbol{a} \setminus \boldsymbol{b}} (x^i - \partial_i \kappa) - \sum_{\boldsymbol{b} \in \binom{\boldsymbol{a}}{3}} \partial_{\boldsymbol{a} \setminus \boldsymbol{b}}^3 \kappa \prod_{i \in \boldsymbol{b}} (x^i - \partial_i \kappa)$$

$$- \sum_{\boldsymbol{b} \in \binom{\boldsymbol{a}}{2}} \partial_{\boldsymbol{a} \setminus \boldsymbol{b}}^4 \kappa \prod_{i \in \boldsymbol{b}} (x^i - \partial_i \kappa) + \sum_{\substack{\boldsymbol{b} \in \binom{\boldsymbol{a}}{2} \\ (\boldsymbol{c}, \boldsymbol{e}) \in \mathrm{part}_2(\boldsymbol{a} \setminus \boldsymbol{b})}} \partial_{\boldsymbol{c}}^2 \kappa \partial_{\boldsymbol{e}}^2 \kappa \prod_{i \in \boldsymbol{b}} (x^i - \partial_i \kappa) + \sum_{i \in \boldsymbol{a}} \mathrm{fn}(\kappa)(x^i - \partial_i \kappa)$$

$$- \sum_{(\boldsymbol{b}, \boldsymbol{c}, \boldsymbol{e}) \in \mathrm{part}_2(\boldsymbol{a})} \partial_{\boldsymbol{b}}^2 \kappa \partial_{\boldsymbol{c}}^2 \kappa \partial_{\boldsymbol{e}}^2 \kappa + \sum_{\boldsymbol{b} \in \binom{\boldsymbol{a}}{2}} \partial_{\boldsymbol{b}}^2 \kappa \partial_{\boldsymbol{a} \setminus \boldsymbol{b}}^4 \kappa + \sum_{(\boldsymbol{b}, \boldsymbol{c}) \in \mathrm{part}_3(\boldsymbol{a})} \partial_{\boldsymbol{b}}^3 \kappa \partial_{\boldsymbol{c}}^3 \kappa - \partial_{\boldsymbol{a}}^6 \kappa.$$

Here $\mathrm{fn}(\kappa)$ is a function of $\kappa$ which does not depend on $x$. Note that fn does not affect the expected value because $\mathbb{E}_{X_{t-1} \sim Q_{t-1|t}}[X_{t-1} - \mu_t] = 0$. Therefore, we have

$$\mathbb{E}_{X_{t-1} \sim Q_{t-1|t}} \left[ \prod_{i \in \boldsymbol{a}: |\boldsymbol{a}| = 6} (X_{t-1}^i - \mu_t^i) \right]$$

$$= \sum_{\boldsymbol{b} \in \binom{\boldsymbol{a}}{2}} \partial_{\boldsymbol{b}}^2 \kappa \left( \sum_{(\boldsymbol{c}, \boldsymbol{e}) \in \mathrm{part}_2(\boldsymbol{a} \setminus \boldsymbol{b})} \partial_{\boldsymbol{c}}^2 \kappa \partial_{\boldsymbol{e}}^2 \kappa + \partial_{\boldsymbol{a} \setminus \boldsymbol{b}}^4 \kappa \right) + \sum_{\boldsymbol{b} \in \binom{\boldsymbol{a}}{3}} \partial_{\boldsymbol{a} \setminus \boldsymbol{b}}^3 \kappa \partial_{\boldsymbol{b}}^3 \kappa$$

$$+ \sum_{\boldsymbol{b} \in \binom{\boldsymbol{a}}{2}} \partial_{\boldsymbol{a} \setminus \boldsymbol{b}}^4 \kappa \partial_{\boldsymbol{b}}^2 \kappa - \sum_{\substack{\boldsymbol{b} \in \binom{\boldsymbol{a}}{2} \\ (\boldsymbol{c}, \boldsymbol{e}) \in \mathrm{part}_2(\boldsymbol{a} \setminus \boldsymbol{b})}} \partial_{\boldsymbol{b}}^2 \kappa \partial_{\boldsymbol{c}}^2 \kappa \partial_{\boldsymbol{e}}^2 \kappa$$

$$+ \sum_{(\boldsymbol{b}, \boldsymbol{c}, \boldsymbol{e}) \in \mathrm{part}_2(\boldsymbol{a})} \partial_{\boldsymbol{b}}^2 \kappa \partial_{\boldsymbol{c}}^2 \kappa \partial_{\boldsymbol{e}}^2 \kappa - \sum_{\boldsymbol{b} \in \binom{\boldsymbol{a}}{2}} \partial_{\boldsymbol{b}}^2 \kappa \partial_{\boldsymbol{a} \setminus \boldsymbol{b}}^4 \kappa - \sum_{(\boldsymbol{b}, \boldsymbol{c}) \in \mathrm{part}_3(\boldsymbol{a})} \partial_{\boldsymbol{b}}^3 \kappa \partial_{\boldsymbol{c}}^3 \kappa + \partial_{\boldsymbol{a}}^6 \kappa$$

$$= \sum_{\boldsymbol{b} \in \binom{\boldsymbol{a}}{2}} \partial_{\boldsymbol{b}}^2 \kappa \partial_{\boldsymbol{a} \setminus \boldsymbol{b}}^4 \kappa + \sum_{(\boldsymbol{b}, \boldsymbol{c}) \in \mathrm{part}_3(\boldsymbol{a})} \partial_{\boldsymbol{b}}^3 \kappa \partial_{\boldsymbol{c}}^3 \kappa + \sum_{(\boldsymbol{b}, \boldsymbol{c}, \boldsymbol{e}) \in \mathrm{part}_2(\boldsymbol{a})} \partial_{\boldsymbol{b}}^2 \kappa \partial_{\boldsymbol{c}}^2 \kappa \partial_{\boldsymbol{e}}^2 \kappa + \partial_{\boldsymbol{a}}^6 \kappa$$

$$= \sum_{(\boldsymbol{b}, \boldsymbol{c}, \boldsymbol{e}) \in \mathrm{part}_2(\boldsymbol{a})} \partial_{\boldsymbol{b}}^2 \kappa \partial_{\boldsymbol{c}}^2 \kappa \partial_{\boldsymbol{e}}^2 \kappa + \tilde{O}_{\mathcal{L}^p(Q_t)}((1 - \alpha_t)^5).$$

The proof is now complete.

### F.9 PROOF OF LEMMA 10

We fix $x_t$ first and will take the expectation at the end. We first introduce some notations used in the proof. We write $x = x_{t-1}$ and $\kappa = \kappa(\tilde{x}_t)$. Given a set of indices $A$, define its bipartition as

$$\mathrm{bipart}(A) := \{(B, C) : A = B \sqcup C\}$$

where $B$ and $C$ are both *sets* of indices (and therefore the order of indices within each of $B$ and $C$ does not matter). Here $\sqcup$ refers to the *disjoint* union of the two sets (which is only defined when the two sets are disjoint). Next, given a set $B$, define $\mathrm{allpart}_{\geq 2}(B)$ as a set containing all partitions of $B$ such that there are *at least* 2 elements in each part of the partition. As an example, $\mathrm{allpart}_{\geq 2}(\{1, 2, 3, 4\}) = \{\{\{1, 2\}, \{3, 4\}\}, \{\{1, 3\}, \{2, 4\}\}, \{\{1, 4\}, \{2, 3\}\}$, and $\{\{1\}, \{2, 3, 4\}\} \notin \mathrm{allpart}_{\geq 2}(\{1, 2, 3, 4\})$ despite the fact that it is a valid partition. For each partition $b \in \mathrm{allpart}_{\geq 2}(B)$, define

$$\partial_b \kappa := \prod_{\xi \in b} \partial_{a_\xi}^{|\xi|} \kappa.$$

Here note that $\xi$ is also a set, and $\partial_b \kappa$ is well defined since the order of indices to take partial derivative with does not matter. Define

$$D_0(x, \tilde{x}_t) := 1$$

$$D_p(x, \tilde{x}_t) := \exp\left(-x^\mathsf{T}\tilde{x}_t + \kappa\right) \partial_{a_p}\left( \exp\left(x^\mathsf{T}\tilde{x}_t - \kappa\right) D_{p-1}(x, \tilde{x}_t)\right)$$

for all $p \geq 1$. We again remind readers that all derivatives are w.r.t. $\tilde{x}_t$ instead of $x = x_{t-1}$.

By working out the derivative, a direct implication of the definition of $D_p$ is a recursive relationship:

$$D_p(x, \tilde{x}_t) = (x^{a_p} - \partial_{a_p}\kappa)D_{p-1}(x, \tilde{x}_t) + \partial_{a_p}D_{p-1}(x, \tilde{x}_t).$$

Also, if we unroll the recursion of $D_p$, we get

$$D_p(x, \tilde{x}_t) = \exp\left(-x^\mathsf{T}\tilde{x}_t + \kappa\right) \partial_{a_p}\left( \exp\left(x^\mathsf{T}\tilde{x}_t - \kappa\right) D_{p-1}(x, \tilde{x}_t)\right)$$

$$= \exp\left(-x^\mathsf{T}\tilde{x}_t + \kappa\right) \partial_{a_p}\left( \exp\left(x^\mathsf{T}\tilde{x}_t - \kappa\right) \exp\left(-x^\mathsf{T}\tilde{x}_t + \kappa\right)\right.$$

$$\left. \partial_{a_{p-1}}\left( \exp\left(x^\mathsf{T}\tilde{x}_t - \kappa\right) D_{p-2}(x, \tilde{x}_t)\right)\right)$$

$$= \exp\left(-x^\mathsf{T}\tilde{x}_t + \kappa\right) \partial^2_{a_p, a_{p-1}}\left( \exp\left(x^\mathsf{T}\tilde{x}_t - \kappa\right) D_{p-2}(x, \tilde{x}_t)\right)$$

$$= \exp\left(-x^\mathsf{T}\tilde{x}_t + \kappa\right) \partial^p_{a_p, \dots, a_1}\left( \exp\left(x^\mathsf{T}\tilde{x}_t - \kappa\right)\right)$$

and thus

$$0 = \partial^p_{a_1, \dots, a_p} \int q_{t-1|t}\mathrm{d}x = \int f(x)\partial^p_{a_1, \dots, a_p}\left( \exp\left(x^\mathsf{T}\tilde{x}_t - \kappa\right)\right)\mathrm{d}x$$

$$= \int f(x) \exp\left(x^\mathsf{T}\tilde{x}_t - \kappa\right) D_p(x, \tilde{x}_t)\mathrm{d}x \qquad (37)$$

where we recall the definition of $f(x)$ back in (32).

In the following, we present the entire proof into two parts. In part 1, we inductively show that each $D_p(x, \tilde{x}_t)$ satisfies a particular polynomial form. In part 2, we inductively show that this polynomial form results in the desired rates.

**Part 1 of the proof of Lemma 10:** The first step toward proving the desired results is to obtain the form of $D_p$ for all $p \geq 2$. Now, we aim to show inductively that

$$D_p(x, \tilde{x}_t) = \prod_{i=1}^p (x^{a_i} - \partial_{a_i}\kappa) - \sum_{(B,C) \in \mathrm{bipart}([p])} \sum_{b \in \mathrm{allpart}_{\geq 2}(B)} d_p(b, C)(\partial_b\kappa) \prod_{c \in C}(x^{a_c} - \partial_{a_c}\kappa) \quad (38)$$

where $d_p(b, C)$ is a constant from combinatorics, which is possibly 0 and which only depends on $p$. From Lemma 8, the bases cases have been established that (cf. (35) and (36))

$$D_2(x, \tilde{x}_t) = (x^i - \partial_i\kappa)(x^j - \partial_j\kappa) - \partial^2_{ij}\kappa$$

$$D_3(x, \tilde{x}_t) = (x^i - \partial_i\kappa)(x^j - \partial_j\kappa)(x^k - \partial_k\kappa)$$
$$\quad - \partial^2_{ij}\kappa(x^k - \partial_k\kappa) - \partial^2_{ik}\kappa(x^j - \partial_j\kappa) - \partial^2_{jk}\kappa(x^i - \partial_i\kappa) - \partial^3_{ijk}\kappa$$

$$D_4(x, \tilde{x}_t) = (x^i - \partial_i\kappa)(x^j - \partial_j\kappa)(x^k - \partial_k\kappa)(x^l - \partial_l\kappa)$$
$$\quad - \partial^2_{ij}\kappa(x^k - \partial_k\kappa)(x^l - \partial_l\kappa) - \partial^2_{ik}\kappa(x^j - \partial_j\kappa)(x^l - \partial_l\kappa) - \partial^2_{jk}\kappa(x^i - \partial_i\kappa)(x^l - \partial_l\kappa)$$
$$\quad + \partial_l((x^i - \partial_i\kappa)(x^j - \partial_j\kappa)(x^k - \partial_k\kappa)) - \partial^3_{ijk}\kappa(x^l - \partial_l\kappa) - \partial^3_{ijl}(x^k - \partial_k\kappa)$$
$$\quad - \partial^3_{ikl}(x^j - \partial_j\kappa) - \partial^3_{jkl}(x^i - \partial_i\kappa) + \partial^2_{ij}\kappa\partial^2_{kl}\kappa + \partial^2_{ik}\kappa\partial^2_{jl}\kappa + \partial^2_{jk}\kappa\partial^2_{il}\kappa - \partial^4_{ijkl}\kappa.$$

In particular, each term of $D_p$ ($p = 2, 3, 4$) is in the form of either $\prod_{i=1}^p(x^{a_i} - \partial_{a_i}\kappa)$ or $(\partial_b\kappa)\prod_{c \in C}(x^{a_c} - \partial_{a_c}\kappa)$, where $|\xi| \geq 2$, $\forall \xi \in b$, and $(\sqcup_{\xi \in b}\xi) \sqcup C = [p]$. Therefore, $D_2, D_3, D_4$ all satisfy the hypothesis (38).

Turning to the inductive step, we suppose that $D_k$ satisfies (38), i.e.,

$$D_k(x, \tilde{x}_t) = \prod_{i=1}^{k}(x^{a_i} - \partial_{a_i}\kappa) - \sum_{(B,C) \in \text{bipart}([k])} \sum_{b \in \text{allpart}_{\geq 2}(B)} d_k(b, C)(\partial_b \kappa) \prod_{c \in C}(x^{a_c} - \partial_{a_c}\kappa).$$

Then, using the recursive relationship, we have

$$D_{k+1}(x, \tilde{x}_t)$$
$$= (x^{a_{k+1}} - \partial_{a_{k+1}}\kappa)D_k(x, \tilde{x}_t) + \partial_{a_{k+1}}D_k(x, \tilde{x}_t)$$
$$= \underbrace{\prod_{i=1}^{k+1}(x^{a_i} - \partial_{a_i}\kappa)}_{T_1} - \underbrace{\sum_{(B,C) \in \text{bipart}([k])} \sum_{b \in \text{allpart}_{\geq 2}(B)} d_k(b, C)(\partial_b \kappa) \prod_{c \in C}(x^{a_c} - \partial_{a_c}\kappa)(x^{a_{k+1}} - \partial_{a_{k+1}}\kappa)}_{T_2}$$
$$- \underbrace{\partial_{a_{k+1}}\left(-\prod_{i=1}^{k}(x^{a_i} - \partial_{a_i}\kappa)\right)}_{T_3} - \underbrace{\sum_{(B,C) \in \text{bipart}([k])} \sum_{b \in \text{allpart}_{\geq 2}(B)} d_k(b, C)(\partial_b \kappa)\left(\partial_{a_{k+1}}\prod_{c \in C}(x^{a_c} - \partial_{a_c}\kappa)\right)}_{T_4}$$
$$- \underbrace{\sum_{(B,C) \in \text{bipart}([k])} \sum_{b \in \text{allpart}_{\geq 2}(B)} d_k(b, C)\left(\partial_{a_{k+1}}(\partial_b \kappa)\right) \prod_{c \in C}(x^{a_c} - \partial_{a_c}\kappa)}_{T_5}$$
$$= T_1 - T_2 - T_3 - T_4 - T_5$$

where we define each term as $T_1, \ldots, T_5$. Now we discuss these terms separately:

1. $T_1$ (and only $T_1$) is in the form $\prod_{i=1}^{k+1}(x^{a_i} - \partial_{a_i}\kappa)$.

2. $T_2$ is a summation of individual terms: $(\partial_b \kappa) \prod_{c \in C}(x^{a_c} - \partial_{a_c}\kappa)(x^{a_{k+1}} - \partial_{a_{k+1}}\kappa)$. Here $b \in \text{allpart}_{\geq 2}(B)$ and $(B, C) \in \text{bipart}([k])$. Thus, by definition of bipart and allpart$_{\geq 2}$, for each $\xi \in b$, $|\xi| \geq 2$ and $(\sqcup_{\xi \in b}\xi) \sqcup C = [k]$. Therefore, $k + 1 \notin B \sqcup C$ and

$$(\sqcup_{\xi \in b}\xi) \sqcup C \sqcup \{k + 1\} = [k] \sqcup \{k + 1\} = [k + 1].$$

This implies that each individual term of $T_2$ is in the form of $(\partial_b \kappa) \prod_{c \in C_2}(x^c - \partial_c \kappa)$ where $b \in \text{allpart}_{\geq 2}(B_2)$, such that $B_2 := B$ and $C_2 := C \sqcup \{k + 1\}$. Here $C_2$ is well defined because $k + 1 \notin C$. Since $(B_2, C_2) \in \text{bipart}([k + 1])$,

$$T_2 = \sum_{(B,C) \in \text{bipart}([k+1])} \sum_{b \in \text{allpart}_{\geq 2}(B)} d_2(b, C)(\partial_b \kappa) \prod_{c \in C}(x^{a_c} - \partial_{a_c}\kappa)$$

for some constant $d_2(b, C)$.

3. $T_3$ is the derivative of product, which is a summation of individual terms: $(\partial_{a_j, a_{k+1}}^2\kappa) \prod_{\substack{i=1 \\ i \neq j}}^{d}(x^{a_i} - \partial_{a_i}\kappa)$, $j = 1, \ldots, k$. Therefore, for each $j = 1, \ldots, k$, each term is of the form $(\partial_b \kappa) \prod_{c \in C_3}(x^{a_c} - \partial_{a_c}\kappa)$ where $b \in \text{allpart}_{\geq 2}(B_3)$, such that $B_3 := \{j, k+1\}$ and $C_3 := [k] \setminus \{j\}$. Since $(B_3, C_3) \in \text{bipart}([k + 1])$,

$$T_3 = \sum_{(B,C) \in \text{bipart}([k+1])} \sum_{b \in \text{allpart}_{\geq 2}(B)} d_3(b, C)(\partial_b \kappa) \prod_{c \in C}(x^{a_c} - \partial_{a_c}\kappa)$$

for some constant $d_3(b, C)$.

4. $T_4$ is a summation of individual terms: $(\partial_b \kappa)\left(\partial_{a_{k+1}} \prod_{c \in C}(x^{a_c} - \partial_{a_c}\kappa)\right)$ where $b \in \text{allpart}_{\geq 2}(B)$ and $(B, C) \in \text{bipart}([k])$. Now,

$$(\partial_b \kappa)\left(\partial_{a_{k+1}} \prod_{c \in C}(x^{a_c} - \partial_{a_c}\kappa)\right) = -(\partial_b \kappa)(\partial_{a_j, a_{k+1}}^2\kappa) \prod_{\substack{i \in C \\ i \neq c}}(x^{a_i} - \partial_{a_i}\kappa)$$

$$= -(\partial_{b_4}\kappa) \prod_{i \in C_4} (x^{a_i} - \partial_{a_i}\kappa)$$

where $b_4 := b \sqcup \{k+1, c\}$ and $C_4 := C \setminus \{c\}$. Here $b_4$ is well defined because $k+1, c \notin b$. Define $B_4 := [k+1] \setminus C_4$, and we have $b_4 \in \text{allpart}_{\geq 2}(B_4)$. Since $(B_4, C_4)$ is a valid partition of $[k+1]$, we have

$$T_4 = \sum_{(B,C) \in \text{bipart}([k+1])} \sum_{b \in \text{allpart}_{\geq 2}(B)} d_4(b,C)(\partial_b\kappa) \prod_{c \in C} (x^{a_c} - \partial_{a_c}\kappa)$$

for some constant $d_4(b, C)$.

5. $T_5$ is a summation of individual terms: $\left(\partial_{a_{k+1}}(\partial_b\kappa)\right) \prod_{c \in C}(x^{a_c} - \partial_{a_c}\kappa)$, where $b \in \text{allpart}_{\geq 2}(B)$ and $(B,C) \in \text{bipart}([k])$. From definition of $\partial_b\kappa$,

$$\partial_{a_{k+1}}(\partial_b\kappa) = \partial_{a_{k+1}}\left(\prod_{\xi \in b} \partial_{a_\xi}^{|\xi|}\kappa\right) = \sum_{\xi \in b}\left(\partial_{a_\xi, a_{k+1}}^{|\xi|+1}\kappa\right) \prod_{\substack{\zeta \in b \\ \zeta \neq \xi}} \partial_{a_\zeta}^{|\zeta|}\kappa = \sum_{\xi \in b} \partial_{b_\xi}\kappa$$

where, for each $\xi \in b$, we have defined a new partition $b_\xi$ such that $k+1$ is added to the $\xi$ in the partition $b$. Formally, define $b_\xi := b \setminus \xi \sqcup \{\xi \sqcup \{k+1\}\}$, which is well defined because $\xi \notin (b \setminus \xi)$ and $k+1 \notin B$. Define $B_5 := B \sqcup \{k+1\}$ and $C_5 := C$, and note that $(B_5, C_5)$ is a valid partition of $[k+1]$. Since $|\zeta| \geq 2$, $\forall \zeta \in b$, we have $|\zeta'| \geq 2$, $\forall \zeta' \in b_\xi$. Since $b \in \text{allpart}_{\geq 2}(B)$, we have $b_\xi \in \text{allpart}_{\geq 2}(B_5)$ for all $\xi \in b$. Therefore, for any fixed $C(= C_5)$

$$\sum_{b \in \text{allpart}_{\geq 2}(B)} d_k(b,C)\left(\partial_{a_{k+1}}(\partial_b\kappa)\right) = \sum_{b \in \text{allpart}_{\geq 2}(B)} \sum_{\xi \in b} d_k(b,C)\partial_{b_\xi}\kappa$$

$$= \sum_{b_5 \in \text{allpart}_{\geq 2}(B_5)} d_5(b_5,C)\partial_{b_5}\kappa$$

for some constant $d_5(b_5, C)$, and thus

$$T_5 = \sum_{(B,C) \in \text{bipart}([k+1])} \sum_{b \in \text{allpart}_{\geq 2}(B)} d_5(b,C)(\partial_b\kappa) \prod_{c \in C} (x^{a_c} - \partial_{a_c}\kappa).$$

Finally, letting

$$d_{k+1}(b,C) := \sum_{j=2}^{5} d_j(b,C)$$

for each $b \in \text{allpart}_{\geq 2}(B)$ and $C$ such that $(B,C) \in \text{bipart}([k+1])$, we have shown that if $D_k(x, \tilde{x}_t)$ is in the form of (38), $D_{k+1}(x, \tilde{x}_t)$ is also in this form. Thus, claim (38) is valid for all $p \geq 2$.

**Part 2 of the proof of Lemma 10:** First, we remind readers of the definition of $\kappa(\tilde{x}_t)$ in (23). Also, the partial derivatives within the expectation over $X_t \sim Q_t$ do not affect the rate by Assumption 5. Note that $\nabla\kappa = \mu_t$ from direct differentiation. From (37) and (38), for fixed $x_t$, we have

$$\mathbb{E}_{X_{t-1} \sim Q_{t-1|t}}\left[\prod_{i=1}^{p}(X_{t-1}^{a_i} - \mu_t^{a_i})\right]$$

$$= \tilde{O}\left(\sup_{\substack{(B,C) \in \text{bipart}([p]) \\ b \in \text{allpart}_{\geq 2}(B)}} \partial_b\kappa(\tilde{x}_t)\mathbb{E}_{X_{t-1} \sim Q_{t-1|t}}\left[\prod_{c \in C}(X_{t-1}^{a_c} - \mu_t^{a_c})\right]\right)$$

$$= \tilde{O}\left(\sup_{(B,C) \in \text{bipart}([p])}\left(\sup_{b \in \text{allpart}_{\geq 2}(B)} \partial_b\kappa(\tilde{x}_t)\right)\mathbb{E}_{X_{t-1} \sim Q_{t-1|t}}\left[\prod_{c \in C}(X_{t-1}^{a_c} - \mu_t^{a_c})\right]\right). \quad (39)$$

We first consider the term $\sup_{b\in\text{allpart}_{\geq 2}(B)} \partial_b\kappa(\tilde{x}_t)$. Given a partition $b \in \text{allpart}_{\geq 2}(B)$, direct differentiation yields

$$\partial_{a_\xi}^{|\xi|}\kappa = \frac{1-\alpha_t}{\alpha_t} + \frac{(1-\alpha_t)^2}{\alpha_t}\partial_{a_\xi}^2 \log q_t(x_t) = \tilde{O}(1-\alpha_t), \quad \text{if } |\xi| = 2 \text{ and } \xi_1 = \xi_2$$

$$\partial_{a_\xi}^{|\xi|}\kappa = \frac{(1-\alpha_t)^{|\xi|}}{\alpha_t^{|\xi|/2}}\partial_\xi^{|\xi|} \log q_t(x_t) = \tilde{O}((1-\alpha_t)^{|\xi|}), \quad \text{for all other } \xi.$$

Since by definition $\partial_b\kappa = \prod_{\xi\in b}\partial_{a_\xi}^{|\xi|}\kappa$ and $\sqcup_{\xi\in b}\xi = B$, the slowest rate of $\partial_b\kappa$ (as a function of $B$) is determined by the partition $b$ containing the most number of equal pairs. The slowest rate is

$$\sup_{b\in\text{allpart}_{\geq 2}(B)} \partial_b\kappa(\tilde{x}_t) = \begin{cases} \tilde{O}\left((1-\alpha_t)^{(|B|-1)/2}(1-\alpha_t)^3\right) = \tilde{O}\left((1-\alpha_t)^{(|B|+5)/2}\right) & \text{if } |B| \text{ is odd} \\ \tilde{O}\left((1-\alpha_t)^{|B|/2}\right) & \text{if } |B| \text{ is even} \end{cases}$$

To proceed, we will again use induction to find the overall rate. From Lemma 8, base cases have been established that

$$\mathbb{E}_{X_{t-1},X_t\sim Q_{t-1,t}}\left[\prod_{i=1}^{2}(X_{t-1}^{a_i} - \mu_t^{a_i})\right] = \tilde{O}(1-\alpha_t), \quad \forall a \in [d]^2$$

$$\mathbb{E}_{X_{t-1},X_t\sim Q_{t-1,t}}\left[\prod_{i=1}^{3}(X_{t-1}^{a_i} - \mu_t^{a_i})\right] = \tilde{O}\left((1-\alpha_t)^3\right), \quad \forall a \in [d]^3$$

$$\mathbb{E}_{X_{t-1},X_t\sim Q_{t-1,t}}\left[\prod_{i=1}^{4}(X_{t-1}^{a_i} - \mu_t^{a_i})\right] = \tilde{O}\left((1-\alpha_t)^2\right), \quad \forall a \in [d]^4.$$

These rates satisfy (24) and (25) when $p = 2, 3, 4$. Now we turn to the inductive step. Suppose $k \geq 4$ is even. For purpose of induction, suppose (24) and (25) hold for all $p = 2, \ldots, k$. Then, following (39), for $p = k + 1$ (odd number), we have

$$\mathbb{E}_{X_{t-1},X_t\sim Q_{t-1,t}}\left[\prod_{i=1}^{k+1}(X_{t-1}^{a_i} - \mu_t^{a_i})\right]$$

$$= O\left(\sup_{\substack{(B,C)\in\text{bipart}([k+1]) \\ |B| \text{ odd}, |C| \text{ even}}} (1-\alpha_t)^{(|B|+5)/2}(1-\alpha_t)^{|C|/2}\right.$$

$$\left. + \sup_{\substack{(B,C)\in\text{bipart}([k+1]) \\ |B| \text{ even}, |C| \text{ odd}}} (1-\alpha_t)^{|B|/2}(1-\alpha_t)^{(|C|+3)/2}\right)$$

$$= O\left((1-\alpha_t)^{(k+1)/2+5/2} + (1-\alpha_t)^{(k+1)/2+3/2}\right)$$

$$= O\left((1-\alpha_t)^{(k+1)/2+3/2}\right).$$

Then, for $p = k + 2$ (even number), we have

$$\mathbb{E}_{X_{t-1},X_t\sim Q_{t-1,t}}\left[\prod_{i=1}^{k+2}(X_{t-1}^{a_i} - \mu_t^{a_i})\right]$$

$$= O\left(\sup_{\substack{(B,C)\in\text{bipart}([k+2]) \\ |B| \text{ odd}, |C| \text{ odd}}} (1-\alpha_t)^{(|B|+5)/2}(1-\alpha_t)^{(|C|+3)/2}\right.$$

$$\left. + \sup_{\substack{(B,C)\in\text{bipart}([k+1]) \\ |B| \text{ even}, |C| \text{even}}} (1-\alpha_t)^{|B|/2}(1-\alpha_t)^{|C|/2}\right)$$

$$= O\left((1-\alpha_t)^{(k+2)/2+4} + (1-\alpha_t)^{(k+2)/2}\right)$$

$$= O\left((1-\alpha_t)^{(k+2)/2}\right).$$

These show the validity of the claims (24) and (25). The proof is now complete.

### F.10 PROOF OF LEMMA 11

Before analyzing the rate of each moment, we need to guarantee the validity of exchanging the limit (in the Taylor expansion) and the expectation operator. Intuitively, this is achievable under Assumption 5, where the Taylor series is absolutely convergent in expectation due to its Gaussian-like moments. Specifically, since $\log q_{t-1}$ is analytic, all its partial derivatives exist. Following from the Taylor expansion of $\zeta'_{t,t-1}$ in (20),

$$
\lim_{k\to\infty} \left| \mathbb{E}_{\substack{X_t\sim Q_t \\ X_{t-1}\sim P'_{t-1|t}}} [\zeta'_{t,t-1}] - \mathbb{E}_{\substack{X_t\sim Q_t \\ X_{t-1}\sim P'_{t-1|t}}} \left[ T_1(\log q_{t-1}, X_{t-1}, \mu_t) + T'_2(\log q_{t-1}, X_{t-1}, \mu_t) \right. \right.
$$

$$
\left. \left. + \sum_{p=3}^{k} T_p(\log q_{t-1}, X_{t-1}, \mu_t) \right] \right|
$$

$$
\leq \lim_{k\to\infty} \mathbb{E}_{\substack{X_t\sim Q_t \\ X_{t-1}\sim P'_{t-1|t}}} \left| \zeta'_{t,t-1} - T_1(\log q_{t-1}, X_{t-1}, \mu_t) - T'_2(\log q_{t-1}, X_{t-1}, \mu_t) \right.
$$

$$
\left. - \sum_{p=3}^{k} T_p(\log q_{t-1}, X_{t-1}, \mu_t) \right|
$$

$$
\leq \lim_{k\to\infty} \mathbb{E}_{\substack{X_t\sim Q_t \\ X_{t-1}\sim P'_{t-1|t}}} \left[ \sum_{p=k+1}^{\infty} |T_p(\log q_{t-1}, X_{t-1}, \mu_t)| \right]
$$

$$
\overset{(i)}{\leq} \lim_{k\to\infty} \liminf_{\ell\to\infty} \sum_{p=k+1}^{\ell} \mathbb{E}_{\substack{X_t\sim Q_t \\ X_{t-1}\sim P'_{t-1|t}}} |T_p(\log q_{t-1}, X_{t-1}, \mu_t)|
$$

$$
\overset{(ii)}{=} 0.
$$

Here $(i)$ follows from Fatou's lemma, and $(ii)$ is because, under Assumption 5 and Lemma 7, we have $\mathbb{E}_{\substack{X_t\sim Q_t \\ X_{t-1}\sim P'_{t-1|t}}} |T_p(\log q_{t-1}, X_{t-1}, \mu_t)| = \tilde{O}\left(T^{-p/2}\right)$, and thus the infinite sum is convergent for all $(k, \ell)$ such that $1 \leq k < \ell < \infty$ since

$$
\sum_{p=1}^{\infty} \mathbb{E}_{\substack{X_t\sim Q_t \\ X_{t-1}\sim P'_{t-1|t}}} |T_p(\log q_{t-1}, x_{t-1}, \mu_t)| = \tilde{O}\left( \sum_{p=1}^{\infty} \frac{1}{p!} \cdot \frac{d^p}{T^{p/2}} \right) < \infty.
$$

The proof for $\mathbb{E}_{\substack{X_t\sim Q_t \\ X_{t-1}\sim Q_{t-1|t}}}$ is similar due to its Gaussian-like concentration of all centralized moments (see Lemma 10). Thus, we are able to exchange the infinite sum and the expectation under either $P'_{t-1|t} \times Q_t$ or $Q_{t-1,t}$.

Next, we put together the rates of the conditional moments. We use abbreviated notations as $T_p = T_p(\log q_{t-1}, X_{t-1}, \mu_t)$. To investigate the dominant term, we analyze the expected difference of the first 8 moments in the Taylor expansion (20) separately. First, for any fixed $x_t$,

$$
\mathbb{E}_{X_{t-1}\sim Q_{t-1|t}} [T_1] = 0 = \mathbb{E}_{X_{t-1}\sim P'_{t-1|t}} [T_1].
$$

Also, for $T'_2$, note that for any random variable $Z$ (regardless of its distribution) with $\mathbb{E}Z = 0$ and $\mathrm{Cov}(Z) = \Sigma$, the mean of the quadratic form (with fixed matrix $\Xi$) is

$$
\mathbb{E}[Z^\intercal \Xi Z] = \mathbb{E}[\mathrm{Tr}\left(Z^\intercal \Xi Z\right)] = \mathrm{Tr}\left(\Xi\Sigma\right).
$$

This implies that, for any fixed $x_t$,

$$
\mathbb{E}_{X_{t-1}\sim P'_{t-1|t}}[T'_2] = \frac{1}{2}\mathbb{E}_{X_{t-1}\sim P'_{t-1|t}} \left[ (X_{t-1} - \mu_t)^\intercal \left( \nabla^2 \log q_{t-1}(\mu_t) - \frac{\alpha_t}{1-\alpha_t} B_t \right) (X_{t-1} - \mu_t) \right]
$$

$$
= \frac{1}{2}\mathrm{Tr}\left( \left( \nabla^2 \log q_{t-1}(\mu_t) - \frac{\alpha_t}{1-\alpha_t} B_t \right) \Sigma_t \right)
$$

$$
= \frac{1}{2}\mathbb{E}_{X_{t-1}\sim Q_{t-1|t}} \left[ (X_{t-1} - \mu_t)^\intercal \left( \nabla^2 \log q_{t-1}(\mu_t) - \frac{\alpha_t}{1-\alpha_t} B_t \right) (X_{t-1} - \mu_t) \right]
$$

$$= \mathbb{E}_{X_{t-1} \sim Q_{t-1|t}}[T_2'].$$

Using Lemmas 7 and 8, the rate for $T_3$ is

$$\mathbb{E}_{X_t \sim Q_t} \left( \mathbb{E}_{X_{t-1} \sim Q_{t-1|t}} - \mathbb{E}_{X_{t-1} \sim P'_{t-1|t}} \right) [T_3(\log q_{t-1}, X_{t-1}, \mu_t)]$$

$$= \mathbb{E}_{X_{t-1}, X_t \sim Q_{t-1,t}} [T_3(\log q_{t-1}, X_{t-1}, \mu_t)]$$

$$= \frac{(1-\alpha_t)^3}{3! \alpha_t^{3/2}} \sum_{i,j,k=1}^d \mathbb{E}_{X_t \sim Q_t}[\partial_{ijk}^3 \log q_{t-1}(\mu_t(X_t)) \partial_{ijk}^3 \log q_t(X_t)].$$

Using Lemmas 7 and 10, and when the partial derivatives satisfy Assumption 5, the rate for $T_5$, $T_7$, and $T_p(p \geq 8)$ can also be determined:

$$\mathbb{E}_{X_t \sim Q_t} \left( \mathbb{E}_{X_{t-1} \sim Q_{t-1|t}} - \mathbb{E}_{X_{t-1} \sim P'_{t-1|t}} \right) [T_5(\log q_{t-1}, X_{t-1}, \mu_t)]$$

$$= \mathbb{E}_{X_{t-1}, X_t \sim Q_{t-1,t}} [T_5(\log q_{t-1}, X_{t-1}, \mu_t)]$$

$$= \tilde{O}((1-\alpha_t)^4),$$

$$\mathbb{E}_{X_t \sim Q_t} \left( \mathbb{E}_{X_{t-1} \sim Q_{t-1|t}} - \mathbb{E}_{X_{t-1} \sim P'_{t-1|t}} \right) [T_7(\log q_{t-1}, X_{t-1}, \mu_t)]$$

$$= \mathbb{E}_{X_{t-1}, X_t \sim Q_{t-1,t}} [T_7(\log q_{t-1}, X_{t-1}, \mu_t)]$$

$$= \tilde{O}((1-\alpha_t)^5),$$

$$\mathbb{E}_{X_t \sim Q_t} \left( \mathbb{E}_{X_{t-1} \sim Q_{t-1|t}} - \mathbb{E}_{X_{t-1} \sim P'_{t-1|t}} \right) [T_p(\log q_{t-1}, X_{t-1}, \mu_t)]$$

$$= \tilde{O}((1-\alpha_t)^4), \ \forall p \geq 8.$$

The remaining orders are $T_4$ and $T_6$. The following proof will draw from the results in Lemmas 7 to 9. Fix $p \geq 1$. Write $Z_i = X_{t-1}^i - \mu_t^i$ and $A^{ij} = [A_t]^{ij}$ for $i, j \in [d]$. For $T_4$, let $i, j, k, l \in [d]$ all differ, and the difference (in expectation) of each term of $T_4$ is

$$\mathbb{E}_{X_{t-1} \sim Q_{t-1|t}}[Z_i^4] - \mathbb{E}_{X_{t-1} \sim P'_{t-1|t}}[Z_i^4]$$

$$= 3 \left( \frac{1-\alpha_t}{\alpha_t} \right)^2 + 6 \frac{(1-\alpha_t)^3}{\alpha_t^2} \partial_{ii}^2 \log q_t(x_t) - 3 \left( \frac{1-\alpha_t}{\alpha_t} \right)^2 (1 + A^{ii})^2 + \tilde{O}_{\mathcal{L}^p(Q_t)}\left((1-\alpha_t)^4\right)$$

$$= -3 \left( \frac{1-\alpha_t}{\alpha_t} \right)^2 (A^{ii})^2 + \tilde{O}_{\mathcal{L}^p(Q_t)}\left((1-\alpha_t)^4\right),$$

$$\mathbb{E}_{X_{t-1} \sim Q_{t-1|t}}[Z_i^3 Z_j] - \mathbb{E}_{X_{t-1} \sim P'_{t-1|t}}[Z_i^3 Z_j]$$

$$= 3 \frac{(1-\alpha_t)^3}{\alpha_t^2} \partial_{ij}^2 \log q_t(x_t) - 3 \left( \frac{1-\alpha_t}{\alpha_t} \right)^2 A^{ij}(1 + A^{ii}) + \tilde{O}_{\mathcal{L}^p(Q_t)}\left((1-\alpha_t)^4\right)$$

$$= -3 \left( \frac{1-\alpha_t}{\alpha_t} \right)^2 A^{ij} A^{ii} + \tilde{O}_{\mathcal{L}^p(Q_t)}\left((1-\alpha_t)^4\right),$$

$$\mathbb{E}_{X_{t-1} \sim Q_{t-1|t}}[Z_i^2 Z_j^2] - \mathbb{E}_{X_{t-1} \sim P'_{t-1|t}}[Z_i^2 Z_j^2]$$

$$= \left( \frac{1-\alpha_t}{\alpha_t} \right)^2 + \frac{(1-\alpha_t)^3}{\alpha_t^2} \left( \partial_{ii}^2 \log q_t(x_t) + \partial_{jj}^2 \log q_t(x_t) \right) - \left( \frac{1-\alpha_t}{\alpha_t} \right)^2 (1 + A^{ii})(1 + A^{jj})$$

$$+ \tilde{O}_{\mathcal{L}^p(Q_t)}\left((1-\alpha_t)^4\right)$$

$$= -\left( \frac{1-\alpha_t}{\alpha_t} \right)^2 A^{ii} A^{jj} + \tilde{O}_{\mathcal{L}^p(Q_t)}\left((1-\alpha_t)^4\right),$$

$$\mathbb{E}_{X_{t-1} \sim Q_{t-1|t}}[Z_i^2 Z_j Z_k] - \mathbb{E}_{X_{t-1} \sim P'_{t-1|t}}[Z_i^2 Z_j Z_k]$$

$$= \frac{(1-\alpha_t)^3}{\alpha_t^2} \partial_{jk}^2 \log q_t(x_t) - \left( \frac{1-\alpha_t}{\alpha_t} \right)^2 (1 + A^{ii}) A^{jk} + \tilde{O}_{\mathcal{L}^p(Q_t)}\left((1-\alpha_t)^4\right)$$

$$= -\frac{(1-\alpha_t)^2}{\alpha_t^2} A^{ii} A^{jk} + \tilde{O}_{\mathcal{L}^p(Q_t)}\left((1-\alpha_t)^4\right),$$

$$\mathbb{E}_{X_{t-1}\sim Q_{t-1|t}}[Z_i Z_j Z_k Z_l] - \mathbb{E}_{X_{t-1}\sim P'_{t-1|t}}[Z_i Z_j Z_k Z_l] = \tilde{O}_{\mathcal{L}^p(Q_t)}\left((1-\alpha_t)^4\right).$$

Recall from (14) that $A_t = (1-\alpha_t)\nabla^2 \log q_t(x_t) = \tilde{O}_{\mathcal{L}^p(Q_t)}(1-\alpha_t)$ under Assumption 5. Hence, many low-order terms above are cancelled, and we get

$$\left(\mathbb{E}_{X_{t-1}\sim Q_{t-1|t}} - \mathbb{E}_{X_{t-1}\sim P'_{t-1|t}}\right)[T_4(\log q_{t-1}, X_{t-1}, \mu_t)] = \tilde{O}_{\mathcal{L}^p(Q_t)}\left((1-\alpha_t)^4\right).$$

Now we turn to $T_6$. Let $i, j, k \in [d]$ all differ, and the difference (in expectation) of each lowest-order term of $T_6$ is

$$\mathbb{E}_{X_{t-1}\sim Q_{t-1|t}}[Z_i^6] - \mathbb{E}_{X_{t-1}\sim P'_{t-1|t}}[Z_i^6]$$
$$= 15\left(\frac{1-\alpha_t}{\alpha_t}\right)^3 - 15\left(\frac{1-\alpha_t}{\alpha_t}\right)^3 (1+A^{ii})^3 + \tilde{O}_{\mathcal{L}^p(Q_t)}((1-\alpha_t)^4),$$

$$\mathbb{E}_{X_{t-1}\sim Q_{t-1|t}}[Z_i^4 Z_j^2] - \mathbb{E}_{X_{t-1}\sim P'_{t-1|t}}[Z_i^4 Z_j^2]$$
$$= 3\left(\frac{1-\alpha_t}{\alpha_t}\right)^3 - 3\left(\frac{1-\alpha_t}{\alpha_t}\right)^3 (1+A^{ii})^2(1+A^{jj}) + \tilde{O}_{\mathcal{L}^p(Q_t)}((1-\alpha_t)^4),$$

$$\mathbb{E}_{X_{t-1}\sim Q_{t-1|t}}[Z_i^2 Z_j^2 Z_k^2] - \mathbb{E}_{X_{t-1}\sim P'_{t-1|t}}[Z_i^2 Z_j^2 Z_k^2]$$
$$= \left(\frac{1-\alpha_t}{\alpha_t}\right)^3 - \left(\frac{1-\alpha_t}{\alpha_t}\right)^3 (1+A^{ii})(1+A^{jj})(1+A^{kk}) + \tilde{O}_{\mathcal{L}^p(Q_t)}((1-\alpha_t)^4).$$

Also, by Lemmas 7 and 9, the rest of the terms already satisfy $\tilde{O}_{\mathcal{L}^p(Q_t)}((1-\alpha_t)^4)$ under Assumption 5. The low-order terms cancel in the same way as for $T_4$, and thus,

$$\left(\mathbb{E}_{X_{t-1}\sim Q_{t-1|t}} - \mathbb{E}_{X_{t-1}\sim P'_{t-1|t}}\right)[T_6(\log q_{t-1}, X_{t-1}, \mu_t)] = \tilde{O}_{\mathcal{L}^p(Q_t)}((1-\alpha_t)^4).$$

Therefore, the lowest order term above is $T_3$, whose order is $\tilde{O}_{\mathcal{L}^p(Q_t)}((1-\alpha_t)^3)$. The proof is now complete.

### F.11 PROOF OF COROLLARY 3

The proof is very similar to Lemma 11 and (21), except with a perturbed covariance matrix. We employ the notations $\tilde{A}_t$ and $\tilde{B}_t$ from Remark 3. Here we have that $\tilde{A}_t(X_t) = A_t(X_t) + \Xi_t(X_t)$, and thus, $\forall r \geq 1$,

$$\tilde{B}_t(X_t) = B_t(X_t) + \tilde{O}_{\mathcal{L}^r(Q_t)}\left((1-\alpha_t)^2\right) = A_t(X_t) + \tilde{O}_{\mathcal{L}^r(Q_t)}\left((1-\alpha_t)^2\right)$$
$$= (1-\alpha_t)\nabla^2 \log q_t(X_t) + \tilde{O}_{\mathcal{L}^r(Q_t)}\left((1-\alpha_t)^2\right).$$

Compare with the proof of Lemma 11, the only difference is the expected difference of $T_2'$. Since $\tilde{A}_t(X_t) = A_t(X_t) + \tilde{O}_{\mathcal{L}^r(Q_t)}\left((1-\alpha_t)^2\right)$ and $\tilde{B}_t(X_t) = B_t(X_t) + \tilde{O}_{\mathcal{L}^r(Q_t)}\left((1-\alpha_t)^2\right)$, the expected differences of all higher order $T_p$'s have the same rate as the non-perturbed case.

Now, for any fixed $x_t$ and $r \geq 1$,

$$\mathbb{E}_{X_{t-1}\sim P'_{t-1|t}}[T_2']$$
$$= \frac{1}{2}\mathbb{E}_{X_{t-1}\sim P'_{t-1|t}}\left[(X_{t-1}-\mu_t)^\intercal \left(\nabla^2 \log q_{t-1}(\mu_t) - \frac{\alpha_t}{1-\alpha_t}\tilde{B}_t\right)(X_{t-1}-\mu_t)\right]$$
$$= \frac{1}{2}\text{Tr}\left(\left(\nabla^2 \log q_{t-1}(\mu_t) - \frac{\alpha_t}{1-\alpha_t}\tilde{B}_t\right)\tilde{\Sigma}_t\right),$$

and, from Lemma 8,

$$\mathbb{E}_{X_{t-1}\sim Q_{t-1|t}}[T_2']$$
$$= \frac{1}{2}\mathbb{E}_{X_{t-1}\sim Q_{t-1|t}}\left[(X_{t-1}-\mu_t)^\intercal \left(\nabla^2 \log q_{t-1}(\mu_t) - \frac{\alpha_t}{1-\alpha_t}\tilde{B}_t\right)(X_{t-1}-\mu_t)\right]$$
$$= \frac{1}{2}\text{Tr}\left(\left(\nabla^2 \log q_{t-1}(\mu_t) - \frac{\alpha_t}{1-\alpha_t}\tilde{B}_t\right)\Sigma_t\right).$$

Thus,

$$\left(\mathbb{E}_{X_{t-1}\sim Q_{t-1|t}} - \mathbb{E}_{X_{t-1}\sim P'_{t-1|t}}\right)[T'_2(\log q_{t-1}, X_{t-1}, \mu_t)]$$

$$= \frac{1}{2}\text{Tr}\left(\left(\nabla^2 \log q_{t-1}(\mu_t) - \frac{\alpha_t}{1-\alpha_t}\tilde{B}_t\right)\left(\Sigma_t - \tilde{\Sigma}_t\right)\right)$$

$$= -\frac{1-\alpha_t}{2\alpha_t}\text{Tr}\left(\left(\nabla^2 \log q_{t-1}(\mu_t) - \frac{\alpha_t}{1-\alpha_t}\tilde{B}_t\right)\Xi_t\right)$$

$$= -\frac{1-\alpha_t}{2\alpha_t}\text{Tr}\left(\left(\nabla^2 \log q_{t-1}(\mu_t) - \alpha_t\nabla^2 \log q_t(X_t)\right)\Xi_t\right) + \tilde{O}_{\mathcal{L}^r(Q_t)}\left((1-\alpha_t)^4\right).$$

Note that here the first term is in the order $\tilde{O}_{\mathcal{L}^r(Q_t)}\left((1-\alpha_t)^3\right)$ under Assumption 5 since $\Xi_t(X_t) = \tilde{O}_{\mathcal{L}^r(Q_t)}\left((1-\alpha_t)^2\right)$. Therefore, under the perturbed case,

$$\mathbb{E}_{X_t\sim Q_t}\left(\mathbb{E}_{X_{t-1}\sim Q_{t-1|t}} - \mathbb{E}_{X_{t-1}\sim P'_{t-1|t}}\right)[\zeta'_{t,t-1}]$$

$$= -\frac{1-\alpha_t}{2\alpha_t}\mathbb{E}_{X_t\sim Q_t}\text{Tr}\left(\left(\nabla^2 \log q_{t-1}(\mu_t(X_t)) - \alpha_t\nabla^2 \log q_t(X_t)\right)\Xi_t(X_t)\right)$$

$$+ \frac{(1-\alpha_t)^3}{3!\alpha_t^{3/2}}\sum_{i,j,k=1}^d \mathbb{E}_{X_t\sim Q_t}[\partial^3_{ijk}\log q_{t-1}(\mu_t(X_t))\partial^3_{ijk}\log q_t(X_t)]$$

$$+ \tilde{O}((1-\alpha_t)^4).$$

The final result can be achieved using (21). The proof is complete.

### F.12 PROOF OF LEMMA 12

From (1), the forward process at the first step is

$$x_1 = \sqrt{\alpha_1}x_0 + \sqrt{1-\alpha_1}w_1$$

where $w_1 \sim \mathcal{N}(0, I_d)$ is independent of $Q_0$. Thus,

$$\mathbb{E}_{X_1\sim Q_1, X_0\sim Q_0}\|X_1 - X_0\|^2 = \mathbb{E}_{W_1\sim \mathcal{N}(0,I_d), X_0\sim Q_0}\left\|\sqrt{1-\alpha_1}W_1 + (\sqrt{\alpha_t}-1)X_0\right\|^2$$

$$\overset{(i)}{=} \mathbb{E}_{W_1\sim \mathcal{N}(0,I_d)}\left\|\sqrt{1-\alpha_1}W_1\right\|^2 + \mathbb{E}_{X_0\sim Q_0}\left\|(\sqrt{\alpha_t}-1)X_0\right\|^2$$

$$\overset{(ii)}{\leq} (1-\alpha_1)d + (\sqrt{\alpha_1}-1)^2 M_2 d$$

$$\overset{(iii)}{\leq} (1-\alpha_1)(M_2+1)d$$

where $(i)$ follows from independence, $(ii)$ follows from Assumption 1, and $(iii)$ follows because $(\sqrt{z}-1)^2 \leq 1-z$ for all $z \in [0,1]$. The proof is complete since $\text{W}_2(Q_0, Q_1)^2 \leq \mathbb{E}_{X_1\sim Q_1, X_0\sim Q_0}\|X_1 - X_0\|^2$ by the definition of Wasserstein-2 distance.

## G PROOF OF THEOREMS 2 TO 4 AND 5

In this section, we instantiate Theorem 1 (along with Corollary 1) to provide upper bounds that have explicit parameter dependency for a number of interesting distribution classes. In order to obtain an upper bound that explicitly depends on system parameters, we need only to provide an explicit bound on the reverse-step error, which is the main topic that we address in the following subsections.

### G.1 PROOF OF THEOREM 2

We first introduce some relevant notations. Given that $Q_0$ is Gaussian mixture, the p.d.f. of $q_t$ at each time $t \geq 1$ can be calculated as

$$q_t(x) = \int_{x_0\in\mathbb{R}^d} q_{t|0}(x|x_0)\sum_{n=1}^N \pi_n q_{0,n}(x_0)\mathrm{d}x_0$$

$$= \sum_{n=1}^N \pi_n \int_{x_0\in\mathbb{R}^d} q_{t|0}(x|x_0)q_{0,n}(x_0)\mathrm{d}x_0 =: \sum_{n=1}^N \pi_n q_{t,n}(x).$$

Since the convolution of two Gaussian density is still Gaussian, we have that $q_{t,n}$ is the p.d.f. of $\mathcal{N}(\mu_{t,n}, \Sigma_{t,n})$, where $\mu_{t,n} := \sqrt{\bar{\alpha}_t}\mu_{0,n}$ and $\Sigma_{t,n} := \bar{\alpha}_t\Sigma_{0,n} + (1-\bar{\alpha}_t)I_d$. Note that $\Sigma_{t,n}$ has full rank.

### G.1.1 Checking Assumption 4

We first verify Assumption 4 for Gaussian mixture $Q_0$ for any $\alpha_t$ that satisfies Definition 1. The intuition is that its Gaussian-like tail (for all $t \geq 0$) is sufficient to control all higher-order derivatives of $\log q_t$.

In the following, Lemma 13 provides an upper bound on any order of partial derivative of a Gaussian mixture density for any fixed $x_t$, as long as each mixture component is well controlled. This directly implies that the partial derivatives are also well controlled in expectation, and thus we verify Assumption 4 for Gaussian mixture in Lemma 14.

**Lemma 13.** *Let $g(x|z)$ be the conditional Gaussian p.d.f. of $\mathcal{N}(\mu_z, \Sigma_z)$. Define $q(x) := \int g(x|z)\mathrm{d}\Pi(z)$, where $\Pi(z)$ is a mixing distribution (and denote $\mathcal{Z}$ its support). Suppose $b := \sup_{z \in \mathcal{Z}} \|\mu_z\| < \infty$, and suppose the following conditions on $\Sigma_z$ hold for all $z \in \mathcal{Z}$:*

*1. There exist $u, U \in \mathbb{R}$ such that $u \leq \det(\Sigma_z) \leq U$;*

*2. There exists $V \in \mathbb{R}$ such that $\left\|\Sigma_z^{-1}\right\| \leq V$;*

*3. There exists $w \in \mathbb{R}$ such that $\sup_{z \in \mathcal{Z}, i,j \in [d]^2} \left|[\Sigma_z^{-\frac{1}{2}}]^{ij}\right| \leq w$.*

*Then,*

$$\left|\partial_{\boldsymbol{a}}^k \log q(x)\right| \leq \min\left\{ C^k B_k \frac{d^{2k} \max\{w, 1\}^k}{u^{k/2}} U^k e^{k\frac{V}{2}(\|x\|^2 + b^2)}, B_k \frac{d^{2k} \max\{w, 1\}^k}{u^{k/2}} |\mathrm{poly}_k(x)| \right\},$$

*where $B_k$ is the Bell number, $C$ is some constant, and $\mathrm{poly}_k(x)$ is some $k$-th order polynomial in $x$.*

*Proof.* See Appendix H.1. □

**Lemma 14.** *When $Q_0$ is Gaussian mixture (see Theorem 2), Assumption 4 is satisfied.*

*Proof.* See Appendix H.2. □

### G.1.2 Expressing $\partial_{ijk}^3 \log q_t$

Now we continue from Theorem 1 to work for an explicit dependency on $d$. We first calculate the second partial derivative of its log-p.d.f. as

$$\nabla^2 \log q_t(x)$$
$$= \frac{1}{q_t^2(x)} \left( q_t(x) \left( \sum_n \pi_n q_{t,n}(x) \left( \Sigma_{t,n}^{-1}(x - \mu_{t,n})(x - \mu_{t,n})^\intercal \Sigma_{t,n}^{-1} - \Sigma_{t,n}^{-1} \right) \right) \right.$$
$$\left. - \left( \sum_n \pi_n q_{t,n}(x) \Sigma_{t,n}^{-1}(x - \mu_{t,n}) \right) \left( \sum_n \pi_n q_{t,n}(x) \Sigma_{t,n}^{-1}(x - \mu_{t,n}) \right)^\intercal \right). \quad (40)$$

Now write $z_{t,n}(x) := \Sigma_{t,n}^{-1}(x - \mu_{t,n})$. Note that $\partial_k z_{t,n}^i = [\Sigma_{t,n}^{-1}]^{ik}$, and that $\partial_k q_{t,n}(x) = q_{t,n}(x)(-z_{t,n}^k(x))$. We can rewrite (40) as

$$\partial_{ij}^2 \log q_t(x) = \frac{1}{q_t^2(x)} \left( q_t(x) \underbrace{\sum_{n=1}^N \pi_n q_{t,n}(x) \left( z_{t,n}^i(x) z_{t,n}^j(x) - [\Sigma_{t,n}^{-1}]^{ij} \right)}_{N1} \right.$$
$$\left. - \underbrace{\left( \sum_n \pi_n q_{t,n}(x) z_{t,n}^i(x) \right) \left( \sum_n \pi_n q_{t,n}(x) z_{t,n}^j(x) \right)}_{N2} \right).$$

To calculate the third partial derivative of its log-p.d.f., we need first to calculate the partial derivative of N1 and N2. The derivative for N1 is given by

$$\partial_k \sum_{n=1}^N \pi_n q_{t,n}(x) \left( z_{t,n}^i(x) z_{t,n}^j(x) - [\Sigma_{t,n}^{-1}]^{ij} \right)$$

$$= \sum_{n=1}^{N} \pi_n q_{t,n}(x)(-z_{t,n}^k(x)) \left( z_{t,n}^i(x) z_{t,n}^j(x) - [\Sigma_{t,n}^{-1}]^{ij} \right)$$

$$+ \sum_{n=1}^{N} \pi_n q_{t,n}(x)[\Sigma_{t,n}^{-1}]^{ik} z_{t,n}^j(x) + \pi_n q_{t,n}(x)[\Sigma_{t,n}^{-1}]^{jk} z_{t,n}^i(x),$$

and the derivative for term N2 is given by

$$\partial_k \left( \sum_{n=1}^{N} \pi_n q_{t,n}(x) z_{t,n}^i(x) \sum_{n=1}^{N} \pi_n q_{t,n}(x) z_{t,n}^j(x) \right)$$

$$= \sum_{n=1}^{N} \pi_n q_{t,n}(x) \left( (-z_{t,n}^k(x)) z_{t,n}^i(x) + [\Sigma_{t,n}^{-1}]^{ik} \right) \sum_{n=1}^{N} \pi_n q_{t,n}(x) z_{t,n}^j(x)$$

$$+ \sum_{n=1}^{N} \pi_n q_{t,n}(x) z_{t,n}^i(x) \sum_{n=1}^{N} \pi_n q_{t,n}(x) \left( (-z_{t,n}^k(x)) z_{t,n}^j(x) + [\Sigma_{t,n}^{-1}]^{jk} \right).$$

Combining these, the derivative for the numerator is

$$\partial_k(q_t(x)\text{N1} - \text{N2}) = \partial_k(q_t(x))\text{N1} + q_t(x)\partial_k(\text{N1}) - \partial_k(\text{N2})$$

$$= - \sum_{n=1}^{N} \pi_n q_{t,n}(x) z_{t,n}^k(x) \sum_{n=1}^{N} \pi_n q_{t,n}(x) \left( z_{t,n}^i(x) z_{t,n}^j(x) - [\Sigma_{t,n}^{-1}]^{ij} \right)$$

$$+ q_t(x) \Bigg( \sum_{n=1}^{N} \pi_n q_{t,n}(x)(-z_{t,n}^k(x)) \left( z_{t,n}^i(x) z_{t,n}^j(x) - [\Sigma_{t,n}^{-1}]^{ij} \right)$$

$$+ \pi_n q_{t,n}(x)[\Sigma_{t,n}^{-1}]^{ik} z_{t,n}^j(x) + \pi_n q_{t,n}(x)[\Sigma_{t,n}^{-1}]^{jk} z_{t,n}^i(x) \Bigg)$$

$$- \sum_{n=1}^{N} \pi_n q_{t,n}(x) \left( (-z_{t,n}^k(x)) z_{t,n}^i(x) + [\Sigma_{t,n}^{-1}]^{ik} \right) \sum_{n=1}^{N} \pi_n q_{t,n}(x) z_{t,n}^j(x)$$

$$- \sum_{n=1}^{N} \pi_n q_{t,n}(x) z_{t,n}^i(x) \sum_{n=1}^{N} \pi_n q_{t,n}(x) \left( (-z_{t,n}^k(x)) z_{t,n}^j(x) + [\Sigma_{t,n}^{-1}]^{jk} \right)$$

$$= -q_t(x) \sum_{n=1}^{N} \pi_n q_{t,n}(x) z_{t,n}^i(x) z_{t,n}^j(x) z_{t,n}^k(x) - \sum_{n=1}^{N} \pi_n q_{t,n}(x) z_{t,n}^k(x) \sum_{n=1}^{N} \pi_n q_{t,n}(x) z_{t,n}^i(x) z_{t,n}^j(x)$$

$$+ \sum_{n=1}^{N} \pi_n q_{t,n}(x) z_{t,n}^j(x) \sum_{n=1}^{N} \pi_n q_{t,n}(x) z_{t,n}^i(x) z_{t,n}^k(x) + \sum_{n=1}^{N} \pi_n q_{t,n}(x) z_{t,n}^i(x) \sum_{n=1}^{N} \pi_n q_{t,n}(x) z_{t,n}^j(x) z_{t,n}^k(x)$$

$$+ q_t(x) \sum_{n=1}^{N} \pi_n q_{t,n}(x)[\Sigma_{t,n}^{-1}]^{ij} z_{t,n}^k(x) + \sum_{n=1}^{N} \pi_n q_{t,n}(x)[\Sigma_{t,n}^{-1}]^{ij} \sum_{n=1}^{N} \pi_n q_{t,n}(x) z_{t,n}^k(x)$$

$$+ q_t(x) \sum_{n=1}^{N} \pi_n q_{t,n}(x)[\Sigma_{t,n}^{-1}]^{ik} z_{t,n}^j(x) - \sum_{n=1}^{N} \pi_n q_{t,n}(x)[\Sigma_{t,n}^{-1}]^{ik} \sum_{n=1}^{N} \pi_n q_{t,n}(x) z_{t,n}^j(x)$$

$$+ q_t(x) \sum_{n=1}^{N} \pi_n q_{t,n}(x)[\Sigma_{t,n}^{-1}]^{jk} z_{t,n}^i(x) - \sum_{n=1}^{N} \pi_n q_{t,n}(x) z_{t,n}^i(x) \sum_{n=1}^{N} \pi_n q_{t,n}(x)[\Sigma_{t,n}^{-1}]^{jk}.$$

Since

$$\partial_{ijk}^3 \log q_t(x) = \partial_k \left( \frac{q_t(x)\text{N1} - \text{N2}}{q_t^2(x)} \right)$$

$$= \frac{1}{q_t^3(x)} \left( \partial_k(q_t(x)\text{N1} - \text{N2})q_t(x) + 2(q_t(x)\text{N1} - \text{N2}) \sum_{n=1}^{N} \pi_n q_{t,n}(x) z_{t,n}^k(x) \right),$$

we get

$$q_t^3(x)\partial_{ijk}^3 \log q_t(x)$$

$$= -q_t^2(x)\sum_{n=1}^N \pi_n q_{t,n}(x)z_{t,n}^i(x)z_{t,n}^j(x)z_{t,n}^k(x) + q_t(x)\sum_{n=1}^N \pi_n q_{t,n}(x)z_{t,n}^k(x)\sum_{n=1}^N \pi_n q_{t,n}(x)z_{t,n}^i(x)z_{t,n}^j(x)$$

$$+ q_t(x)\sum_{n=1}^N \pi_n q_{t,n}(x)z_{t,n}^j(x)\sum_{n=1}^N \pi_n q_{t,n}(x)z_{t,n}^i(x)z_{t,n}^k(x)$$

$$+ q_t(x)\sum_{n=1}^N \pi_n q_{t,n}(x)z_{t,n}^i(x)\sum_{n=1}^N \pi_n q_{t,n}(x)z_{t,n}^j(x)z_{t,n}^k(x)$$

$$- 2\left(\sum_n \pi_n q_{t,n}(x)z_{t,n}^i(x)\right)\left(\sum_n \pi_n q_{t,n}(x)z_{t,n}^j(x)\right)\left(\sum_{n=1}^N \pi_n q_{t,n}(x)z_{t,n}^k(x)\right)$$

$$+ q_t^2(x)\sum_{n=1}^N \pi_n q_{t,n}(x)[\Sigma_{t,n}^{-1}]^{ij}z_{t,n}^k(x) - q_t(x)\sum_{n=1}^N \pi_n q_{t,n}(x)[\Sigma_{t,n}^{-1}]^{ij}\sum_{n=1}^N \pi_n q_{t,n}(x)z_{t,n}^k(x)$$

$$+ q_t^2(x)\sum_{n=1}^N \pi_n q_{t,n}(x)[\Sigma_{t,n}^{-1}]^{ik}z_{t,n}^j(x) - q_t(x)\sum_{n=1}^N \pi_n q_{t,n}(x)[\Sigma_{t,n}^{-1}]^{ik}\sum_{n=1}^N \pi_n q_{t,n}(x)z_{t,n}^j(x)$$

$$+ q_t^2(x)\sum_{n=1}^N \pi_n q_{t,n}(x)[\Sigma_{t,n}^{-1}]^{jk}z_{t,n}^i(x) - q_t(x)\sum_{n=1}^N \pi_n q_{t,n}(x)z_{t,n}^i(x)\sum_{n=1}^N \pi_n q_{t,n}(x)[\Sigma_{t,n}^{-1}]^{jk}.$$

Below, we write $\xi_t(x,i) := \max_n \left|z_{t,n}^i(x)\right|$ and $\bar\Sigma$ to be a matrix such that $\bar\Sigma^{ij} := \max_n \left|[\Sigma_{t,n}^{-1}]^{ij}\right|$. Also write $h_{t,n}(x) = \pi_n q_{t,n}(x)/q_t(x)$. Note that for any $x$, $\sum_{n=1}^N h_{t,n}(x) = 1$. Therefore, we take $\max_n$ within each summation above and get

$$\left|\partial_{ijk}^3 \log q_t(x)\right| \le 6\xi_t(x,i)\xi_t(x,j)\xi_t(x,k) + 2\bar\Sigma^{ij}\xi_t(x,k) + 2\bar\Sigma^{ik}\xi_t(x,j) + 2\bar\Sigma^{jk}\xi_t(x,i).$$

### G.1.3 ASYMPTOTIC EQUIVALENCE OF $\mu_t(x_t)$ AND $x_t$

Intuitively, $\mu_t(x_t)$ and $x_t$ are asymptotically close when $1 - \alpha_t$ is small, which will be useful for later analysis. In this subsubsection, we will show that $\xi_{t-1}(\mu_t, i) - \xi_t(x_t, i) = \tilde O(1 - \alpha_t)$.

Note that for each $n$ and fixed $x_t$ (writing $\mu_t(x_t) = \mu_t$),

$$z_{t-1,n}(\mu_t) - z_{t,n}(x_t)$$
$$= \Sigma_{t-1,n}^{-1}(\mu_t - \mu_{t-1,n}) - \Sigma_{t,n}^{-1}(x_t - \mu_{t,n})$$
$$= (\Sigma_{t-1,n}^{-1} - \Sigma_{t,n}^{-1})(\mu_t - \mu_{t-1,n}) - \Sigma_{t,n}^{-1}((x_t - \mu_{t,n}) - (\mu_t - \mu_{t-1,n})). \tag{41}$$

Here, since $\Sigma_{t-1,n}$ is real symmetric, we can write the eigen-decomposition as $\Sigma_{t-1,n} = UDU^\intercal$, where $U$ is an orthonormal matrix (having unit 2-norm) and $D$ is a diagonal matrix (with all diagonal elements positive). In the same notation, $\Sigma_{t-1,n}^{-1} = UD^{-1}U^\intercal$, and $\Sigma_{t,n}^{-1} = (\alpha_t \Sigma_{t-1,n} + (1 - \alpha_t)I_d)^{-1} = U(\alpha_t D + (1 - \alpha_t)I_d)^{-1}U^\intercal$. Since

$$\left|[D^{-1}]^{ii} - [(\alpha_t D + (1 - \alpha_t)I_d)^{-1}]^{ii}\right| = \left|\frac{1}{D^{ii}} - \frac{1}{\alpha_t D^{ii} + (1 - \alpha_t)}\right|$$

$$\le \frac{(1 - \alpha_t)(\left|D^{ii}\right| + 1)}{\alpha_t (D^{ii})^2 + (1 - \alpha_t)D^{ii}}$$

$$= \tilde O(1 - \alpha_t),$$

the following holds:

$$\left\|\Sigma_{t-1,n}^{-1} - \Sigma_{t,n}^{-1}\right\| = \tilde O(1 - \alpha_t).$$

Denote $[A]^{i*}$ as the $i$-th row of a matrix $A$. Thus, following from (41), for any $i \in [d]$,

$$\left\|[\Sigma_{t-1,n}^{-1}]^{i*} - [\Sigma_{t,n}^{-1}]^{i*}\right\| \overset{(i)}{\le} \left\|\Sigma_{t-1,n}^{-1} - \Sigma_{t,n}^{-1}\right\| = \tilde O(1 - \alpha_t), \tag{42}$$

$$\left|\mu_t^i - x_t^i\right| = \left|\frac{1-\sqrt{\alpha_t}}{\sqrt{\alpha_t}}x_t^i - \frac{1-\alpha_t}{\sqrt{\alpha_t}}\partial_i \log q_t(x_t)\right| = \tilde{O}(1-\alpha_t),$$

$$\left|\mu_{t,n}^i - \mu_{t-1,n}^i\right| = \left|(1-\sqrt{\alpha_t})\mu_{t-1,n}^i\right| = \tilde{O}(1-\alpha_t),$$

where $(i)$ follows from the definition of matrix 2-norm and from the fact that $[\Sigma_{t,n}^{-1}]^{i*} = \Sigma_{t,n}^{-1}\mathbf{1}_i$ ($\mathbf{1}_i$ is the unit vector where the $i$-th element is 1, and recall that $\Sigma_{t,n}^{-1}$ is symmetric). This implies that $\left|z_{t-1,n}^i(\mu_t) - z_{t,n}^i(x_t)\right| = \tilde{O}(1-\alpha_t), \forall i$. Thus,

$$\xi_{t-1}(\mu_t, i) - \xi_t(x_t, i) = \max_n \left|z_{t-1,n}^i(\mu_t)\right| - \max_n \left|z_{t,n}^i(x_t)\right|$$

$$\leq \max_n \left|z_{t-1,n}^i(\mu_t) - z_{t,n}^i(x_t)\right| = \tilde{O}(1-\alpha_t), \tag{43}$$

where the last inequality follows because $\max_n |a_n| + \max_n |b_n| \geq \max_n(|a_n| + |b_n|) \geq \max_n |a_n + b_n|$.

Following from Theorem 1, we have

$$\mathbb{E}_{X_t \sim Q_t}\left[\sum_{i,j,k=1}^d \partial_{ijk}^3 \log q_{t-1}(\mu_t(X_t))\partial_{ijk}^3 \log q_t(X_t)\right]$$

$$\leq \mathbb{E}_{X_t \sim Q_t}\left[\sum_{i,j,k=1}^d \left(6\xi(\mu_t(X_t),i)\xi(\mu_t(X_t),j)\xi(\mu_t(X_t),k) + 2\bar{\Sigma}^{ij}\xi(\mu_t(X_t),k) + 2\bar{\Sigma}^{ik}\xi(\mu_t(X_t),j)\right.$$

$$\left. + 2\bar{\Sigma}^{jk}\xi(\mu_t(X_t),i)\right)\left(6\xi(X_t,i)\xi(X_t,j)\xi(X_t,k) + 2\bar{\Sigma}^{ij}\xi(X_t,k) + 2\bar{\Sigma}^{ik}\xi(X_t,j) + 2\bar{\Sigma}^{jk}\xi(X_t,i)\right)\right]$$

$$\overset{(ii)}{\lesssim} \mathbb{E}_{X_t \sim Q_t}\left[\sum_{i,j,k=1}^d \left(\xi(X_t,i)\xi(X_t,j)\xi(X_t,k) + \bar{\Sigma}^{ij}\xi(X_t,k) + \bar{\Sigma}^{ik}\xi(X_t,j) + \bar{\Sigma}^{jk}\xi(X_t,i)\right)^2\right]$$

$$\leq 2\mathbb{E}_{X_t \sim Q_t}\left[\sum_{i,j,k=1}^d \xi(X_t,i)^2\xi(X_t,j)^2\xi(X_t,k)^2 + (\bar{\Sigma}^{ij})^2\xi(X_t,k)^2 + (\bar{\Sigma}^{ik})^2\xi(X_t,j)^2 + (\bar{\Sigma}^{jk})^2\xi(X_t,i)^2\right]$$

$$\tag{44}$$

where $(ii)$ follows from (43).

### G.1.4 EXPLICIT PARAMETER DEPENDENCY

We are now ready for the explicit parameter dependency for Gaussian mixture $Q_0$. In the following, we provide two different ways to upper-bound the terms in (44) depending on how $N$ is compared to $d$. The first approach can be applied when $N < d$. For the $\xi(x, \cdot)$ ($\forall x \in \mathbb{R}^d$) terms,

$$\sum_{i=1}^d \xi(x,i)^2 = \sum_{i=1}^d \max_n([\Sigma_{t,n}^{-1}]^{i*}(x-\mu_{t,n}))^2 \leq \sum_{i=1}^d \sum_{n=1}^N ([\Sigma_{t,n}^{-1}]^{i*}(x-\mu_{t,n}))^2$$

$$= \sum_{n=1}^N \left\|\Sigma_{t,n}^{-1}(x-\mu_{t,n})\right\|^2 \leq N \max_n \left\|\Sigma_{t,n}^{-1}\right\|^2 \max_n \left\|x-\mu_{t,n}\right\|^2$$

$$\overset{(i)}{\lesssim} N \max_n \left\|x-\mu_{t,n}\right\|^2,$$

where $(i)$ follows because of the following. Since $\Sigma_{t,n}$ is a (full-rank) covariance matrix, all its eigenvalues are positive. Let $\lambda_{n,\min} > 0$ be the smallest eigenvalue of $\Sigma_{0,n}$, and thus

$$\max_n \left\|\Sigma_{t,n}^{-1}\right\|_2 \leq \frac{1}{\bar{\alpha}_t \min_n \lambda_{n,\min} + (1-\bar{\alpha}_t)} \leq \frac{1}{\min\{1, \min_n \lambda_{n,\min}\}} < \infty. \tag{45}$$

In particular, this bound does not depend on $d$ or $T$. Also, for the $\bar{\Sigma}$ terms,

$$\sum_{i,j=1}^d (\bar{\Sigma}^{ij})^2 = \sum_{i,j=1}^d \max_n([\Sigma_{t,n}^{-1}]^{ij})^2 \leq \sum_{i,j=1}^d \sum_{n=1}^N ([\Sigma_{t,n}^{-1}]^{ij})^2 = \sum_{n=1}^N \left\|\Sigma_{t,n}^{-1}\right\|_F^2 \lesssim Nd,$$

where the last inequality follows from (45) and the fact that for any matrix full-rank $A$, $\|A\|_F \leq \sqrt{d}\|A\|_2$. The second approach can be applied when $N \geq d$, where we can bound the $\xi(x, \cdot)$ ($\forall x \in \mathbb{R}^d$) terms instead as

$$
\sum_{i=1}^d \xi(x,i)^2 = \sum_{i=1}^d \max_n ([\Sigma_{t,n}^{-1}]^{i*}(x-\mu_{t,n}))^2
$$
$$
\overset{(ii)}{\leq} \sum_{i=1}^d \max_n \left( \left\|[\Sigma_{t,n}^{-1}]^{i*}\right\|^2 \|x-\mu_{t,n}\|^2 \right) \leq \sum_{i=1}^d \max_n \left\|[\Sigma_{t,n}^{-1}]^{i*}\right\|^2 \max_n \|x-\mu_{t,n}\|^2
$$
$$
\overset{(iii)}{\leq} \sum_{i=1}^d \max_n \left\|\Sigma_{t,n}^{-1}\right\|^2 \max_n \|x-\mu_{t,n}\|^2 \overset{(iv)}{\lesssim} d \max_n \|x-\mu_{t,n}\|^2 .
$$

Here $(ii)$ follows from Cauchy-Schwartz inequality, $(iii)$ follows from definition of matrix 2-norm and the fact that $[\Sigma_{t,n}^{-1}]^{i*} = \Sigma_{t,n}^{-1}\mathbf{1}_i$ ($\mathbf{1}_i$ is the unit vector where the $i$-th element is 1), and $(iv)$ follows from (45). Also, for the second term, we can obtain an alternative upper bound as follows. Write the eigen-decomposition as $\Sigma_{0,n} = Q_n \mathrm{diag}(\lambda_{n,1}, \ldots, \lambda_{n,d})Q_n^\mathsf{T}$, where $Q_n$ here is an orthonormal matrix (that does not depend on $T$). Then,

$$
\Sigma_{t,n}^{-1} = Q_n(\bar{\alpha}_t \mathrm{diag}(\lambda_{n,1}, \ldots, \lambda_{n,d}) + (1-\bar{\alpha}_t)I_d)^{-1}Q_n^\mathsf{T}
$$
$$
= Q_n \mathrm{diag}((\bar{\alpha}_t \lambda_{n,1} + (1-\bar{\alpha}_t))^{-1}, \ldots, (\bar{\alpha}_t \lambda_{n,d} + (1-\bar{\alpha}_t))^{-1})Q_n^\mathsf{T},
$$

and thus

$$
\max_{n \in [N]} \left|[\Sigma_{t,n}^{-1}]^{ij}\right| = \max_{n \in [N]} \left| \sum_{k=1}^d (\bar{\alpha}_t \lambda_{n,k} + (1-\bar{\alpha}_t))^{-1} Q_n^{ik} Q_n^{kj} \right|
$$
$$
\leq (\min\{1, \min_n \lambda_{n,\min}\})^{-1} \max_{n \in [N], i,j \in [d]} \left|(Q_n^{i*})^\mathsf{T}(Q_n^{j*})\right|
$$
$$
\leq (\min\{1, \min_n \lambda_{n,\min}\})^{-1} \max_{n \in [N], i \in [d]} \left\|Q_n^{i*}\right\|^2
$$
$$
= (\min\{1, \min_n \lambda_{n,\min}\})^{-1},
$$

where the last line follows because $Q_n$ is orthonormal for all $n \in [N]$. Note that this is a uniform bound that does not depend on $N$, $T$ or $d$, which further implies that

$$
\sum_{i,j=1}^d (\bar{\Sigma}^{ij})^2 \lesssim d^2.
$$

Combining the two cases, we get

$$
\sum_{i=1}^d \xi(x,i)^2 \lesssim \min\{d,N\} \max_n \|x-\mu_{t,n}\|^2 , \tag{46}
$$

$$
\sum_{i,j=1}^d (\bar{\Sigma}^{ij})^2 \lesssim d \min\{d,N\}. \tag{47}
$$

Therefore, using (46) and (47), we can continue from (44) and get

$$
\mathbb{E}_{X_t \sim Q_t} \left[ \sum_{i,j,k=1}^d \partial_{ijk}^3 \log q_{t-1}(\mu_t(X_t))\partial_{ijk}^3 \log q_t(X_t) \right]
$$
$$
\lesssim \min\{d,N\}^3 \mathbb{E}_{X_t \sim Q_t} \left[ \|X_t\|^6 + \max_n \|\mu_{t,n}\|^6 \right] + (d\min\{d,N\})(d\min\{d,N\}).
$$

Now, note that

$$
\max_n \|\mu_{t,n}\|^6 \leq \max_n \|\mu_{0,n}\|^6 \lesssim d^3
$$

since $\mu_{0,n} < \infty$ is a fixed vector. Also, the expected sixth power of the norm can be bounded as

$$\mathbb{E}\|X_t\|^6 = \mathbb{E}\left[\left(\|\sqrt{\bar{\alpha}_t}X_0 + \sqrt{1-\bar{\alpha}_t}\bar{W}_t\|^2\right)^3\right] \lesssim \mathbb{E}\|X_0\|^6 + \mathbb{E}\|\bar{W}_t\|^6 \lesssim \mathbb{E}\|X_0\|^6 + d^3,$$

and, when $Q_0$ is a Gaussian mixture,

$$\int \|x_0\|^6 q_0(x_0)\mathrm{d}x_0 = \sum_{n=1}^{N} \pi_n \int \|x_0\|^6 q_{0,n}(x_0)\mathrm{d}x_0 \asymp d^3.$$

Therefore, we finally obtain a bound on the reverse-step error with explicit system parameters:

$$\sum_{t=1}^{T} \mathbb{E}_{X_{t-1}, X_t \sim Q_{t-1,t}} \left[\log \frac{q_{t-1|t}(X_{t-1}|X_t)}{p'_{t-1|t}(X_{t-1}|X_t)}\right] \lesssim \frac{d^3 \min\{d, N\}^3 \log^3 T}{T^2}.$$

## G.2 PROOF OF THEOREM 3

Throughout the proof of Theorem 3 we adopt the noise schedule $\alpha_t$ defined in (10). We first investigate some nice properties of the noise schedule in (10). Since $c \asymp \log(1/\delta)$, we have $1 - \alpha_t \lesssim \log(1/\delta)\log T/T$. Using a similar argument from (Li et al., 2024c, Equation (39)),

$$\frac{1-\alpha_t}{\alpha_t - \bar{\alpha}_t}, \frac{1-\alpha_t}{1-\bar{\alpha}_t}, \frac{1-\alpha_t}{1-\bar{\alpha}_{t-1}} \lesssim \frac{\log(1/\delta)\log T}{T}, \quad \forall 2 \le t \le T, \tag{48}$$

$$\frac{1-\bar{\alpha}_t}{1-\bar{\alpha}_{t-1}} - 1 = \frac{\bar{\alpha}_{t-1}(1-\alpha_t)}{1-\bar{\alpha}_{t-1}} \le \frac{1-\alpha_t}{1-\bar{\alpha}_{t-1}} = \tilde{O}\left(\frac{\log T}{T}\right), \quad \forall 2 \le t \le T.$$

We note that Li et al. (2024c) does not highlight $\delta$ dependency in their results. Also, note that if $T$ is large,

$$\delta\left(1 + \frac{c\log T}{T}\right)^{\frac{T}{\log T}} \asymp \delta e^c \ge 1.$$

Thus, with any fixed $r \in (0, 1)$ such that $t \ge rT \ (\ge \frac{T}{\log T})$, we have

$$1 - \alpha_t = \frac{c\log T}{T} \min\left\{\delta\left(1 + \frac{c\log T}{T}\right)^t, 1\right\} = \frac{c\log T}{T}.$$

As a result,

$$\bar{\alpha}_T \le \prod_{t=\lfloor rT \rfloor}^{T} \alpha_t = \left(1 - \frac{c\log T}{T}\right)^{\lceil(1-r)T\rceil} \asymp \exp\left(\lceil(1-r)T\rceil\left(-\frac{c\log T}{T}\right)\right) = \tilde{O}(T^{-(1-r)c}). \tag{49}$$

Given any $c > 2$, we can always find some $r$ such that $(1-r)c > 2$. For example, this is satisfied when $r = (c-2)/4$ if $c \in (2, 4)$ and $r = 1/4$ otherwise. This shows that the $\alpha_t$ in (10) satisfies $\bar{\alpha}_T = o\left(T^{-2}\right)$ if $c > 2$. Therefore, the $\alpha_t$ in (10) satisfies Definition 1.

Since the parameter dependency is clear in the bound for the initialization and estimation errors (Lemmas 3 and 4), it remains to provide a bound on the reverse-step error that depends explicitly on the system parameters, which is the main topic below.

### G.2.1 CHECKING ASSUMPTION 5

Instead of Assumption 4, we check the more general Assumption 5 below. In particular, we verify Assumption 5 with the $\alpha_t$ in (10). In the following, Lemma 15 is used to establish the first half of Assumption 5. Next, the following Lemma 16 is used to establish the behavior of the expected moments under the perturbed posterior $Q_{0|t-1}(\cdot|\mu_t(X_t))$ when $X_t \sim Q_t$. Both Lemmas 15 and 17 will be useful for establishing the second half of Assumption 5 with the $\alpha_t$ in (10).

**Lemma 15.** *For all $t \ge 1$, $\ell \ge 1$, and $\boldsymbol{a} \in [d]^p$ such that $|\boldsymbol{a}| = p \ge 1$,*

$$\mathbb{E}_{X_t \sim Q_t} |\partial_{\boldsymbol{a}}^p \log q_t(X_t)|^\ell \lesssim \frac{d^{p\ell/2}}{(1-\bar{\alpha}_t)^{p\ell/2}}.$$

*Proof.* See Appendix H.3. □

**Lemma 16.** *For all $t \geq 2$ and $p \geq 1$, with the $\alpha_t$ in (10),*

$$\int_{x_0, x_t} \|\mu_t(x_t) - \sqrt{\bar{\alpha}_{t-1}} x_0\|^p \, \mathrm{d}Q_{0|t-1}(x_0|\mu_t(x_t)) \mathrm{d}Q_t(x_t) \lesssim d^{p/2}(1 - \bar{\alpha}_{t-1})^{p/2}.$$

*Proof.* See Appendix H.4. □

Finally, the following Lemma 17 verifies the second half of Assumption 5 with the $\alpha_t$ defined in (10).

**Lemma 17.** *For all $t \geq 2$, $\ell \geq 1$, and $\boldsymbol{a} \in [d]$ such that $|\boldsymbol{a}| = p \geq 1$, with the $\alpha_t$ in (10),*

$$\mathbb{E}_{X_t \sim Q_t} |\partial_{\boldsymbol{a}}^p \log q_{t-1}(\mu_t(X_t))|^\ell \lesssim \frac{d^{p\ell/2}}{(1 - \bar{\alpha}_{t-1})^{p\ell/2}}.$$

*Combining this with Lemma 15, Assumption 5 holds.*

*Proof.* See Appendix H.5. □

Now, Assumption 5 is satisfied since $\frac{1}{1-\bar{\alpha}_t} \leq \frac{1}{1-\alpha_1} = \delta^{-1}$ for all $t \geq 1$ if $\delta$ is constant. Thus, if $\delta$ is a constant, Assumption 4 is already satisfied (as is Assumption 5). This is not necessary, however, when $\delta = 1/\mathrm{poly}(T)$ is vanishing with $T$. Fortunately, in this case, from (48), we still get $\frac{1-\alpha_t}{1-\bar{\alpha}_{t-1}} = \tilde{O}(1-\alpha_t)$. Thus, Assumption 5 is still satisfied.

### G.2.2 EXPRESSING $\partial_{ijk}^3 \log q_t$

We begin by investigating $\nabla^2 \log q_t$ $(t \geq 2)$, for which we can derive the Hessian of $\log q_t(x)$ as

$$\nabla^2 \log q_t(x) = \frac{\partial}{\partial x} \left( \frac{\int_{x_0 \in \mathbb{R}^d} \nabla q_{t|0}(x|x_0) \mathrm{d}Q_0(x_0)}{\int_{x_0 \in \mathbb{R}^d} q_{t|0}(x|x_0) \mathrm{d}Q_0(x_0)} \right)$$

$$= \frac{q_t(x) \int_{x_0 \in \mathbb{R}^d} \nabla^2 q_{t|0}(x|x_0) \mathrm{d}Q_0(x_0) - \left( \int_{x_0 \in \mathbb{R}^d} \nabla q_{t|0}(x|x_0) \mathrm{d}Q_0(x_0) \right) \left( \int_{x_0 \in \mathbb{R}^d} \nabla q_{t|0}(x|x_0) \mathrm{d}Q_0(x_0) \right)^{\mathsf{T}}}{q_t^2(x)}$$

$$= \frac{1}{(1-\bar{\alpha}_t)^2 q_t^2(x)} \left( q_t(x) \int_{x_0 \in \mathbb{R}^d} q_{t|0}(x|x_0) \left( (x - \sqrt{\bar{\alpha}_t} x_0)(x - \sqrt{\bar{\alpha}_t} x_0)^{\mathsf{T}} - (1 - \bar{\alpha}_t) I_d \right) \mathrm{d}Q_0(x_0) \right.$$

$$\left. - \left( \int_{x_0 \in \mathbb{R}^d} q_{t|0}(x|x_0)(x - \sqrt{\bar{\alpha}_t} x_0) \mathrm{d}Q_0(x_0) \right) \left( \int_{x_0 \in \mathbb{R}^d} q_{t|0}(x|x_0)(x - \sqrt{\bar{\alpha}_t} x_0) \mathrm{d}Q_0(x_0) \right)^{\mathsf{T}} \right)$$

$$= -\frac{1}{1-\bar{\alpha}_t} I_d + \frac{1}{(1-\bar{\alpha}_t)^2} \left( \mathbb{E}_{X_0 \sim Q_{0|t}(\cdot|x)} \left[ (x - \sqrt{\bar{\alpha}_t} X_0)(x - \sqrt{\bar{\alpha}_t} X_0)^{\mathsf{T}} \right] \right.$$

$$\left. - \left( \mathbb{E}_{X_0 \sim Q_{0|t}(\cdot|x)} \left[ x - \sqrt{\bar{\alpha}_t} X_0 \right] \right) \left( \mathbb{E}_{X_0 \sim Q_{0|t}(\cdot|x)} \left[ x - \sqrt{\bar{\alpha}_t} X_0 \right] \right)^{\mathsf{T}} \right). \quad (50)$$

For the third-order partial derivatives, we employ the notation

$$z := \frac{x - \sqrt{\bar{\alpha}_t} x_0}{1 - \bar{\alpha}_t}.$$

Note that $\partial_k q_{t|0}(x|x_0) = q_{t|0}(x|x_0)(-z^k)$. Then, we can write (50) as

$$\partial_{ij}^2 \log q_t(x) = \frac{1}{q_t^2(x)} \left( q_t(x) \underbrace{\int q_{t|0}(x|x_0) z^i z^j \mathrm{d}Q_0(x_0)}_{\mathrm{N1}} \right.$$

$$\left. - \underbrace{\int q_{t|0}(x|x_0) z^i \mathrm{d}Q_0(x_0) \int q_{t|0}(x|x_0) z^j \mathrm{d}Q_0(x_0)}_{\mathrm{N2}} \right) - \frac{1}{1-\bar{\alpha}_t} I_d.$$

Note that the last term is a constant. The derivative for term N1 is given by

$$\partial_k \int q_{t|0}(x|x_0) z^i z^j \mathrm{d}Q_0(x_0)$$

$$= \int q_{t|0}(x|x_0)(-z^k) z^i z^j + \mathbb{1}(k=i) q_{t|0}(x|x_0)(1-\bar{\alpha}_t)^{-1} z^j$$

$$+ \mathbb{1}(k=j) q_{t|0}(x|x_0)(1-\bar{\alpha}_t)^{-1} z^i \mathrm{d}Q_0(x_0),$$

and the derivative for term N2 is given by

$$\partial_k \left( \int q_{t|0}(x|x_0) z^i \mathrm{d}Q_0(x_0) \int q_{t|0}(x|x_0) z^j \mathrm{d}Q_0(x_0) \right)$$

$$= \int q_{t|0}(x|x_0) \left( (-z^k) z^i + \mathbb{1}(k=i)(1-\bar{\alpha}_t)^{-1} \right) \mathrm{d}Q_0(x_0) \int q_{t|0}(x|x_0) z^j \mathrm{d}Q_0(x_0)$$

$$+ \int q_{t|0}(x|x_0) z^i \mathrm{d}Q_0(x_0) \int q_{t|0}(x|x_0) \left( (-z^k) z^j + \mathbb{1}(k=j)(1-\bar{\alpha}_t)^{-1} \right) \mathrm{d}Q_0(x_0)$$

$$= \left( \int q_{t|0}(x|x_0)(-z^k) z^i \mathrm{d}Q_0(x_0) + \mathbb{1}(k=i)(1-\bar{\alpha}_t)^{-1} q_t(x) \right) \int q_{t|0}(x|x_0) z^j \mathrm{d}Q_0(x_0)$$

$$+ \int q_{t|0}(x|x_0) z^i \mathrm{d}Q_0(x_0) \left( \int q_{t|0}(x|x_0)(-z^k) z^j \mathrm{d}Q_0(x_0) + \mathbb{1}(k=j)(1-\bar{\alpha}_t)^{-1} q_t(x) \right).$$

Combining these, the derivative for the numerator is given by

$$\partial_k(q_t(x)\mathrm{N1} - \mathrm{N2}) = \partial_k(q_t(x))\mathrm{N1} + q_t(x)\partial_k(\mathrm{N1}) - \partial_k(\mathrm{N2})$$

$$= -q_t(x) \int q_{t|0}(x|x_0) z^i z^j z^k \mathrm{d}Q_0(x_0)$$

$$- \int q_{t|0}(x|x_0) z^k \mathrm{d}Q_0(x_0) \int q_{t|0}(x|x_0) z^i z^j \mathrm{d}Q_0(x_0)$$

$$+ \int q_{t|0}(x|x_0) z^j \mathrm{d}Q_0(x_0) \int q_{t|0}(x|x_0) z^i z^k \mathrm{d}Q_0(x_0)$$

$$+ \int q_{t|0}(x|x_0) z^i \mathrm{d}Q_0(x_0) \int q_{t|0}(x|x_0) z^j z^k \mathrm{d}Q_0(x_0).$$

Thus,

$$\partial^3_{ijk} \log q_t(x) = \partial_k \frac{q_t(x)\mathrm{N1} - \mathrm{N2}}{q_t^2(x)}$$

$$= \frac{1}{q_t^3(x)} \left( \partial_k(q_t(x)\mathrm{N1} - \mathrm{N2}) q_t(x) + 2(q_t(x)\mathrm{N1} - \mathrm{N2}) \int q_{t|0}(x|x_0) z^k \mathrm{d}Q_0(x_0) \right)$$

$$= \frac{1}{q_t^3(x)} \Bigg( -q_t^2(x) \int q_{t|0}(x|x_0) z^i z^j z^k \mathrm{d}Q_0(x_0)$$

$$+ q_t(x) \sum_{\substack{a_1 = i,j,k \\ a_2 < a_3,\ a_2,a_3 \neq a_1}} \int q_{t|0}(x|x_0) z^{a_1} \mathrm{d}Q_0(x_0) \int q_{t|0}(x|x_0) z^{a_2} z^{a_3} \mathrm{d}Q_0(x_0)$$

$$- 2 \int q_{t|0}(x|x_0) z^i \mathrm{d}Q_0(x_0) \int q_{t|0}(x|x_0) z^j \mathrm{d}Q_0(x_0) \int q_{t|0}(x|x_0) z^k \mathrm{d}Q_0(x_0) \Bigg)$$

$$= - \int z^i z^j z^k \mathrm{d}Q_{0|t}(x_0|x)$$

$$+ \sum_{\substack{a_1 = i,j,k \\ a_2 < a_3,\ a_2,a_3 \neq a_1}} \int z^{a_1} \mathrm{d}Q_{0|t}(x_0|x) \int z^{a_2} z^{a_3} \mathrm{d}Q_{0|t}(x_0|x)$$

$$- 2 \int z^i \mathrm{d}Q_{0|t}(x_0|x) \int z^j \mathrm{d}Q_{0|t}(x_0|x) \int z^k \mathrm{d}Q_{0|t}(x_0|x) \qquad (51)$$

### G.2.3 EXPLICIT PARAMETER DEPENDENCY

By Cauchy-Schwartz inequality, we have

$$
\mathbb{E}_{X_t \sim Q_t} \left[ \sum_{i,j,k=1}^{d} \partial^3_{ijk} \log q_{t-1}(\mu_t(X_t)) \partial^3_{ijk} \log q_t(X_t) \right]
$$

$$
\leq \sqrt{\mathbb{E}_{X_t \sim Q_t} \left[ \sum_{i,j,k=1}^{d} \left( \partial^3_{ijk} \log q_{t-1}(\mu_t(X_t)) \right)^2 \right]} \times \sqrt{\mathbb{E}_{X_t \sim Q_t} \left[ \sum_{i,j,k=1}^{d} \left( \partial^3_{ijk} \log q_t(X_t) \right)^2 \right]}.
$$

$$(52)$$

We now analyze the two terms in (52) separately.

We begin with the second term in (52). Recall that $Z = \frac{X_t - \sqrt{\bar{\alpha}_t} X_0}{1 - \bar{\alpha}_t}$ is standard Gaussian under $Q_{0,t}$. Also note that for a standard Gaussian random variable $Z$, $\mathbb{E} \|Z\|^6 = d(d+2)(d+4) \lesssim d^3$. Now, substituting (51) into the second term of (52), we get

$$
\sum_{i,j,k=1}^{d} \mathbb{E}_{X_t \sim Q_t} \left( \int z^i z^j z^k \mathrm{d}Q_{0|t}(x_0|X_t) \right)^2
$$

$$
\leq \frac{1}{(1-\bar{\alpha}_t)^3} \mathbb{E}_{X_0, X_t \sim Q_{0,t}} \left[ \sum_{i,j,k=1}^{d} \left( \frac{X_t^i - \sqrt{\bar{\alpha}_t} X_0^i}{\sqrt{1 - \bar{\alpha}_t}} \right)^2 \left( \frac{X_t^j - \sqrt{\bar{\alpha}_t} X_0^j}{\sqrt{1 - \bar{\alpha}_t}} \right)^2 \left( \frac{X_t^k - \sqrt{\bar{\alpha}_t} X_0^k}{\sqrt{1 - \bar{\alpha}_t}} \right)^2 \right]
$$

$$
= \frac{1}{(1-\bar{\alpha}_t)^3} \mathbb{E}_{X_0, X_t \sim Q_{0,t}} \left\| \frac{X_t - \sqrt{\bar{\alpha}_t} X_0}{\sqrt{1 - \bar{\alpha}_t}} \right\|^6
$$

$$
= \frac{1}{(1-\bar{\alpha}_t)^3} \mathbb{E} \|Z\|^6
$$

$$
\lesssim \frac{d^3}{(1-\bar{\alpha}_t)^3},
$$

and

$$
\sum_{i,j,k=1}^{d} \mathbb{E}_{X_t \sim Q_t} \left( \int z^i \mathrm{d}Q_{0|t}(x_0|x) \int z^j z^k \mathrm{d}Q_{0|t}(x_0|x) \right)^2
$$

$$
= \mathbb{E}_{X_t \sim Q_t} \left[ \left\| \int z \mathrm{d}Q_{0|t}(x_0|x) \right\|^2 \sum_{j,k=1}^{d} \left( \int z^j z^k \mathrm{d}Q_{0|t}(x_0|x) \right)^2 \right]
$$

$$
\leq \left( \mathbb{E}_{X_t \sim Q_t} \left\| \int z \mathrm{d}Q_{0|t}(x_0|x) \right\|^6 \right)^{1/3} \left( \mathbb{E}_{X_t \sim Q_t} \left( \sum_{j,k=1}^{d} \left( \int z^j z^k \mathrm{d}Q_{0|t}(x_0|x) \right)^2 \right)^{3/2} \right)^{2/3}
$$

$$
\leq \mathbb{E}_{X_0, X_t \sim Q_{0,t}} \left\| \frac{X_t - \sqrt{\bar{\alpha}_t} X_0}{1 - \bar{\alpha}_t} \right\|^6
$$

$$
= \frac{1}{(1-\bar{\alpha}_t)^3} \mathbb{E} \|Z\|^6
$$

$$
\lesssim \frac{d^3}{(1-\bar{\alpha}_t)^3},
$$

and

$$
\sum_{i,j,k=1}^{d} \mathbb{E}_{X_t \sim Q_t} \left( \int z^i \mathrm{d}Q_{0|t}(x_0|X_t) \int z^j \mathrm{d}Q_{0|t}(x_0|X_t) \int z^k \mathrm{d}Q_{0|t}(x_0|X_t) \right)^2
$$

$$
= \mathbb{E}_{X_t \sim Q_t} \left( \sum_{i=1}^{d} \left( \int \frac{X_t^i - \sqrt{\bar{\alpha}_t} x_0^i}{1 - \bar{\alpha}_t} \mathrm{d}Q_{0|t}(x_0|X_t) \right)^2 \right)^3
$$

$$= \frac{1}{(1-\bar{\alpha}_t)^3} \mathbb{E}_{X_t \sim Q_t} \left\| \int \frac{X_t - \sqrt{\bar{\alpha}_t}x_0}{\sqrt{1-\bar{\alpha}_t}} dQ_{0|t}(x_0|X_t) \right\|^6$$

$$\leq \frac{1}{(1-\bar{\alpha}_t)^3} \mathbb{E} \|Z\|^6$$

$$\lesssim \frac{d^3}{(1-\bar{\alpha}_t)^3}.$$

Thus, the second term of (52) satisfies that

$$\mathbb{E}_{X_t \sim Q_t} \left[ \sum_{i,j,k=1}^d \left( \partial_{ijk}^3 \log q_t(X_t) \right)^2 \right] \lesssim \frac{d^3}{(1-\bar{\alpha}_t)^3}.$$

Now we turn to the first term in (52). Note that $Z = \frac{\mu_t(X_t) - \sqrt{\bar{\alpha}_{t-1}}X_0}{1-\bar{\alpha}_{t-1}}$. While $Z$ is no longer standard Gaussian under $Q_{0,t}$, we can still achieve moment bounds using Lemma 16. Now, substituting (51) into the first term of (52), we apply Lemma 16 and get

$$\sum_{i,j,k=1}^d \mathbb{E}_{X_t \sim Q_t} \left( \int z^i z^j z^k dQ_{0|t-1}(x_0|\mu_t(X_t)) \right)^2$$

$$\leq \frac{1}{(1-\bar{\alpha}_{t-1})^3} \mathbb{E}_{\substack{X_0 \sim Q_{0|t-1}(\cdot|\mu_t(X_t)) \\ X_t \sim Q_t}} \left\| \frac{\mu_t(X_t) - \sqrt{\bar{\alpha}_{t-1}}X_0}{\sqrt{1-\bar{\alpha}_{t-1}}} \right\|^6 \lesssim \frac{d^3}{(1-\bar{\alpha}_{t-1})^3},$$

and similarly,

$$\sum_{i,j,k=1}^d \mathbb{E}_{X_t \sim Q_t} \left( \int z^i dQ_{0|t-1}(x_0|\mu_t(X_t)) \int z^j z^k dQ_{0|t-1}(x_0|\mu_t(X_t)) \right)^2$$

$$\lesssim \frac{d^3}{(1-\bar{\alpha}_{t-1})^3},$$

$$\sum_{i,j,k=1}^d \mathbb{E}_{X_t \sim Q_t} \left( \int z^i dQ_{0|t-1}(x_0|\mu_t(X_t)) \int z^j dQ_{0|t-1}(x_0|\mu_t(X_t)) \int z^k dQ_{0|t-1}(x_0|\mu_t(X_t)) \right)^2$$

$$\lesssim \frac{d^3}{(1-\bar{\alpha}_{t-1})^3}.$$

Thus, the first term of (52) satisfies that

$$\mathbb{E}_{X_t \sim Q_t} \left[ \sum_{i,j,k=1}^d \left( \partial_{ijk}^3 \log q_{t-1}(\mu_t(X_t)) \right)^2 \right] \lesssim \frac{d^3}{(1-\bar{\alpha}_{t-1})^3}.$$

Finally, since $\frac{1-\alpha_t}{1-\bar{\alpha}_t}, \frac{1-\alpha_t}{1-\bar{\alpha}_{t-1}} \lesssim \frac{\log(1/\delta)\log T}{T}$, we arrive at

$$(1-\alpha_t)^3 \mathbb{E}_{X_t \sim Q_t} \left[ \sum_{i,j,k=1}^d \partial_{ijk}^3 \log q_{t-1}(\mu_t(X_t)) \partial_{ijk}^3 \log q_t(X_t) \right] \lesssim \frac{d^3 \log^3(1/\delta) \log^3 T}{T^3}.$$

Summation over $t \geq 2$ gives us the desirable result.

### G.3 THEOREM 5 AND ITS PROOF

Before we enter the proof of Theorem 4, we introduce an intermediate result which might have independent interest. Previously, for regular samplers, linear dimensional dependency can be shown when all $Q_t$'s ($\forall t \geq 0$) have Lipschitz score (Chen et al., 2023a;d). The following Theorem 5 provides an accelerated convergence guarantee when *all* $Q_t$'s ($\forall t \geq 0$) have Lipschitz Hessians.

**Theorem 5** (Accelerated Sampler for All-Path Lipschitz Hessians). *Suppose that* $\nabla^2 \log q_t(x)$, $\forall t \geq 0$ *is 2-norm $M$-Lipschitz, i.e., $\exists M > 0$ such that*

$$\left\| \nabla^2 \log q_t(x) - \nabla^2 \log q_t(y) \right\| \leq M \|x - y\| \tag{53}$$

*for all $x, y \in \mathbb{R}^d$ and $t \geq 0$. Then, under Assumptions 1, 3 and 5, if the $\alpha_t$ satisfies Definition 1, the distribution $\widehat{P}_0'$ from the accelerated sampler satisfies*

$$\mathrm{KL}(Q_0 || \widehat{P}_0') \lesssim \frac{d^2 M^2 \log^3 T}{T^2} + (\log T)\varepsilon^2 + \frac{\log^2 T}{T}\varepsilon_H^2.$$

### G.3.1 PROOF OF THEOREM 5

In order to continue from Theorem 1 (in particular, the reverse-step error in (26)), we need to introduce some useful notations for the distribution class in (53). For a matrix $A$, define its vectorization as $\mathrm{vec}(A) := [A^{11}, \ldots, A^{1d}, \ldots, A^{d1}, \ldots, A^{dd}]^\mathsf{T} \in \mathbb{R}^{d^2}$. Define $K_t \in \mathbb{R}^{d^2 \times d}$ to be the matrix that reorganizes the third-order partial derivative tensor, i.e.,

$$[K_t(x)]^{mk} := \partial^3_{ijk} \log q_t(x), \text{ s.t. } m = (i-1)d + j, \ \forall i, j, k \in [d].$$

With these notations, consider $y = x + \xi u$ where $u \in \mathbb{R}^d$ satisfies $\|u\|^2 = 1$ and $\xi \in \mathbb{R}$ is some small constant. Then,

$$\mathrm{vec}(\nabla^2 \log q_t(y)) - \mathrm{vec}(\nabla^2 \log q_t(x)) = K_t(x^*)(y - x) = \xi K_t(x^*)u.$$

Here $x^* = \gamma x + (1 - \gamma)y$ for some $\gamma \in (0, 1)$. Also, we have

$$\begin{aligned}
&\left\| \mathrm{vec}(\nabla^2 \log q_t(y)) - \mathrm{vec}(\nabla^2 \log q_t(x)) \right\| \\
&= \left\| \nabla^2 \log q_t(y) - \nabla^2 \log q_t(x) \right\|_F \\
&\leq \sqrt{d} \left\| \nabla^2 \log q_t(y) - \nabla^2 \log q_t(x) \right\| \leq \sqrt{d} M \|y - x\|
\end{aligned}$$

where the last inequality comes from (53). Thus, noting that $y = x + \xi u$ and that $\|u\|^2 = 1$, we take the limit of $\xi$ to 0 and get

$$\|K_t(x)\| \leq \sqrt{d}M, \quad \forall x \in \mathbb{R}^d, \ \forall t \geq 0. \tag{54}$$

We now derive an explicit upper bound on the reverse-step error. Using Cauchy-Schwartz inequality, for any $t \geq 1$ and $x_t \in \mathbb{R}^d$, we have

$$\begin{aligned}
&\sum_{i,j,k=1}^d \partial^3_{ijk} \log q_{t-1}(\mu_t) \partial^3_{ijk} \log q_t(x_t) \\
&\leq \sqrt{\sum_{i,j,k=1}^d (\partial^3_{ijk} \log q_{t-1}(\mu_t))^2} \sqrt{\sum_{i,j,k=1}^d (\partial^3_{ijk} \log q_t(x_t))^2} \\
&= \|K_{t-1}(\mu_t)\|_F \times \|K_t(x_t)\|_F \\
&\leq (\sqrt{d} \|K_{t-1}(\mu_t)\|) \times (\sqrt{d} \|K_t(x_t)\|) \\
&\leq d^2 M^2.
\end{aligned} \tag{55}$$

Therefore, following from Theorem 1, we obtain

$$\sum_{t=1}^T \mathbb{E}_{X_{t-1}, X_t \sim Q_{t-1,t}} \left[ \log \frac{q_{t-1|t}(X_{t-1}|X_t)}{p_{t-1|t}(X_{t-1}|X_t)} \right] \lesssim \frac{d^2 M^2 \log^3 T}{T^2}.$$

### G.4 PROOF OF THEOREM 4

Throughout the proof of Theorem 4 we adopt the noise schedule $\alpha_t$ defined in (10) with $\delta = 1/(M^{\frac{2}{3}} T^{\frac{3}{2}})$ and $c \geq \log(M^{\frac{2}{3}} T^{\frac{3}{2}})$. Note that such $\alpha_t$ satisfies Definition 1 for all $t \geq 1$, and thus the bound on the estimation error still applies. Also, Assumption 5 is satisfied for $t \geq 2$, as shown in Appendix G.2.1. Thus, Theorem 3 can be applied and the reverse-step error at $t \geq 2$ satisfies, $\forall t = T, \ldots, 2$,

$$(1 - \alpha_t)^3 \mathbb{E}_{X_t \sim Q_t} \left[ \sum_{i,j,k=1}^d \partial^3_{ijk} \log q_{t-1}(\mu_t(X_t)) \partial^3_{ijk} \log q_t(X_t) \right]$$

$$\lesssim \frac{d^3 (\log^3 M + \log^3 T) \log^3 T}{T^3}. \quad (56)$$

In order to determine the dimensional dependency of the reverse-step error, the key is thus to establish a similar upper bound at $t = 1$.

Now, we provide a modified version of Theorem 1 which does not require $q_0$ to be analytic (as in Assumption 2) or to have regular partial derivatives (as in Assumption 5). We recall from (21) that the reverse-step error at time $t = 1$ can be upper-bounded as

$$\mathbb{E}_{X_0 \sim Q_{0|1}} \left[ \log \frac{q_{0|1}(X_0|x_1)}{p'_{0|1}(X_0|x_1)} \right] \leq \mathbb{E}_{X_0 \sim Q_{0|1}}[\zeta'_{1,0}] - \mathbb{E}_{X_0 \sim P'_{0|1}}[\zeta'_{1,0}].$$

Instead of the Taylor expansion in (20), we employ the following different expansion from Taylor's theorem. The only difference is that the expansion stops at the third-order term.

$$\zeta'_{1,0} = (\nabla \log q_0(\mu_1) - \sqrt{\alpha_0} \nabla \log q_1(x_1))^\intercal (x_0 - \mu_1)$$
$$+ \frac{1}{2}(x_0 - \mu_1)^\intercal \left( \nabla^2 \log q_0(\mu_1) - \frac{\alpha_0}{1 - \alpha_0} B_t \right) (x_0 - \mu_1)$$
$$+ \frac{1}{3!} \sum_{i,j,k=1}^d \partial^3_{ijk} \log q_0(\mu_1^*)(x_0^i - \mu_1^i)(x_0^j - \mu_1^j)(x_0^k - \mu_1^k). \quad (57)$$

Here $\mu_1^*(x_1, x_0) := \varsigma \mu_1(x_1) + (1 - \varsigma)x_0$ for some $\varsigma \in [0, 1]$. Note that $\mu_1^*$ is a function of both $x_1$ and $x_0$.

A remarkable difference from the proof of Theorem 1 is that we do not require $q_0$ to be analytic for this expansion. Indeed, it only requires that the third-order partial derivative exists. With this new expansion, we have the following lemma, which serves as a counterpart of Lemma 11.

**Lemma 18.** *Suppose that $q_0$ exists and $\nabla^2 \log q_0$ is 2-norm $M$-Lipschitz. Then, with the $\alpha_t$ in (10), we have*

$$\mathbb{E}_{X_0 \sim Q_0} \left( \mathbb{E}_{X_0 \sim Q_{0|1}} - \mathbb{E}_{X_0 \sim P'_{0|1}} \right) [\zeta'_{1,0}] \lesssim \frac{(1 - \alpha_1)^{3/2}}{3! \alpha_1^{3/2}} d^4 M.$$

*Proof.* See Appendix H.6. □

Finally, with the chosen $\delta = 1 - \alpha_1 = 1/(M^{\frac{2}{3}} T^{\frac{3}{2}})$, the rate at the first step satisfies

$$\frac{(1 - \alpha_1)^{3/2}}{3! \alpha_1^{3/2}} d^4 M \lesssim \frac{d^4}{T^{9/4}} = o(T^{-2}).$$

As $T$ becomes large, the rate of the total reverse-step error, which decays as $\tilde{O}(T^{-2})$, is not affected. The proof is now complete.

## H AUXILIARY PROOFS OF THEOREMS 2 TO 4

In this section, we provide the proofs for the lemmas in the proofs for Theorems 2 to 4.

### H.1 PROOF OF LEMMA 13

Fix $k \geq 1$ and $\boldsymbol{a} \in [d]^k$. Recall that $u \leq \det(\Sigma_z) \leq U$, $\left\| \Sigma_z^{-1} \right\| \leq V$, and $\sup_{z \in \mathcal{Z}, i,j \in [d]^2} \left| [\Sigma_z^{-\frac{1}{2}}]^{ij} \right| \leq w$ for all $z \in \mathcal{Z}$. Also write $\phi(y)$ as the p.d.f. of the unit Gaussian. We are interested in upper-bounding the absolute partial derivatives of $\log q(x)$ with a function of $x$ where

$$q(x) = \int g(x|z) \mathrm{d}\Pi(z),$$

where, using the change-of-variable formula,

$$g(x|z) = \frac{1}{\det(\Sigma_z)^{\frac{1}{2}}} \phi \left( \Sigma_z^{-\frac{1}{2}}(x - \mu_z) \right). \quad (58)$$

We first identify an upper bound on the absolute partial derivatives of $q(x)$. Now,

$$\partial_{\boldsymbol{a}}^k q(x) \stackrel{(i)}{=} \int \partial_{\boldsymbol{a}}^k g(x|z) \mathrm{d}\Pi(z)$$

$$\stackrel{(ii)}{\leq} \frac{1}{\inf_{z \in \mathcal{Z}} \det(\Sigma_z)^{\frac{1}{2}}} \int \partial_{\boldsymbol{a}}^k \phi\left(\Sigma_z^{-\frac{1}{2}}(x - \mu_z)\right) \mathrm{d}\Pi(z)$$

where $(i)$ follows from the dominated convergence theorem (see (31)), and $(ii)$ follows from (58). To obtain an upper bound on the $k$-th derivative of Gaussian density, we invoke the multivariate version of the Faá di Bruno's formula (Constantine & Savits, 1996, Theorem 2.1). Since $y = \Sigma_z^{-\frac{1}{2}}(x - \mu_z)$ is linear in $x$, only the first-order partial derivative is non-zero and is equal to an entry in $\Sigma_z^{-\frac{1}{2}}$. Thus, we have

$$\left| \partial_{\boldsymbol{a}}^k \phi\left(\Sigma_z^{-\frac{1}{2}}(x - \mu_z)\right) \right| = \left| \sum_{\boldsymbol{a}' \in [d]^k} \phi_{\boldsymbol{a}'}^{(k)}(y) \prod_{s=1}^k \frac{\partial}{\partial x_{a_s}} [\Sigma_z^{-\frac{1}{2}}(x - \mu_z)]^{a'_s} \right|$$

$$\leq \left| \sum_{\boldsymbol{a}' \in [d]^k} \phi_{\boldsymbol{a}'}^{(k)}\left(\Sigma_z^{-\frac{1}{2}}(x - \mu_z)\right) \right| \max\{w, 1\}^k, \ \forall \boldsymbol{a} : |\boldsymbol{a}| = k.$$

Here we define $\phi_{\boldsymbol{a}}^{(k)}(y) := \partial_{\boldsymbol{a}}^k \phi(y)$. Since $\phi(y)$ is a Gaussian density which is infinitely differentiable and decays exponentially at the tail, its $k$-th order derivative satisfies $\phi_{\boldsymbol{a}}^{(k)}(y) = \mathrm{poly}_k(y)\phi(y)$ where $\mathrm{poly}_k(y)$ is a $k$-th order polynomial function in $y_1, \ldots, y_d$ (and thus in $x_1, \ldots, x_d$ by linearity). Also note that, for any $\boldsymbol{a} \in [d]^k$,

$$\lim_{\|y\| \to \infty} \left| \phi_{\boldsymbol{a}}^{(k)}(y) \right| = \lim_{\|y\| \to \infty} |\mathrm{poly}_k(y)\phi(y)| = 0.$$

By the continuity of $\phi_{\boldsymbol{a}}^{(k)}(y)$, there exists $\bar{y}_{\boldsymbol{a}}$ such that $\left| \phi_{\boldsymbol{a}}^{(k)}(y) \right| \leq \left| \phi_{\boldsymbol{a}}^{(k)}(\bar{y}_{\boldsymbol{a}}) \right| \leq \mathrm{poly}_k(\bar{y}_{\boldsymbol{a}})$ for all $y \in \mathbb{R}^d$. Now, for all $x \in \mathbb{R}^d$,

$$\left| \partial_{\boldsymbol{a}}^k q(x) \right| \leq \int \det(\Sigma_z)^{-\frac{1}{2}} \left| \partial_{\boldsymbol{a}}^k \phi\left(\Sigma_z^{-\frac{1}{2}}(x - \mu_z)\right) \right| \mathrm{d}\Pi(z)$$

$$\leq \max\{w, 1\}^k \int \det(\Sigma_z)^{-\frac{1}{2}} \left( \sum_{\boldsymbol{a} \in [d]^k} \left| \mathrm{poly}_k\left(\Sigma_z^{-\frac{1}{2}}(x - \mu_z)\right) \right| \right) \phi\left(\Sigma_z^{-\frac{1}{2}}(x - \mu_z)\right) \mathrm{d}\Pi(z) \tag{59}$$

$$\leq \frac{d^k \max\{w, 1\}^k}{\sqrt{u}} \left| \mathrm{poly}_k(\bar{y}_{\boldsymbol{a}}) \right| \phi(\bar{y}_{\boldsymbol{a}}). \tag{60}$$

We have thus obtained a constant upper bound on all partial derivatives of $q(x)$ of order $k$.

Next, we convert the partial derivative bound into that for $\log q(x)$. We again invoke Faá di Bruno's formula Constantine & Savits (1996). Note that

$$\partial_{\boldsymbol{a}}^k \log q(x) = q(x)^{-k} \sum_{\boldsymbol{b}_1, \ldots, \boldsymbol{b}_k} \prod_{j=1}^k \partial_{\boldsymbol{b}_j}^{|\boldsymbol{b}_j|} q(x) =: \sum_{\boldsymbol{b}_1, \ldots, \boldsymbol{b}_k} r_{\boldsymbol{b}_1, \ldots, \boldsymbol{b}_k}(x) \tag{61}$$

in which we define each summation term as $r$. Here $\{\boldsymbol{b}_1, \ldots, \boldsymbol{b}_k\}$ is some (possibly empty) partition of $\boldsymbol{a}$, i.e., $\sum_j \boldsymbol{b}_j = \boldsymbol{a}$ and $\sum_j |\boldsymbol{b}_j| = k$ (thus, at most $k$ partitions). We order this partition such that $k \geq |\boldsymbol{b}_1| \geq \cdots \geq |\boldsymbol{b}_k| \geq 0$. Note that the total number of partition can be upper-bounded by $d^k \sum_{l=1}^k B_{k,l}(1, \ldots, 1) = d^k B_k$, where $B_{k,l}(\cdot)$ and $B_k$ are the Bell polynomials and the Bell number, respectively.

We first showcase a simple yet useful upper bound. From (60), we get,

$$\left| \prod_{j=1}^k \partial_{\boldsymbol{b}_j}^{|\boldsymbol{b}_j|} q(x) \right| \leq \prod_{j=1}^k \left| \partial_{\boldsymbol{b}_j}^{|\boldsymbol{b}_j|} q(x) \right|$$

$$\leq \frac{(d\max\{w,1\})^{\sum_j |\boldsymbol{b}_j|}}{\min\{u,1\}^{k/2}} \max\{\max_y \phi(y),1\}^k \prod_{j=1}^k \left|\mathrm{poly}_{|\boldsymbol{b}_j|}(\bar{y}_{\boldsymbol{b}_j})\right|$$

$$\leq \frac{(d\max\{w,1\})^{\sum_j |\boldsymbol{b}_j|}}{\min\{u,1\}^{k/2}} \max\{\max_y \phi(y),1\}^k \max_j \left|\mathrm{poly}_{|\boldsymbol{b}_j|}(\bar{y}_{\boldsymbol{b}_j})\right|^k$$

$$\leq C_{\boldsymbol{b_1},\dots,\boldsymbol{b_j}}^k \frac{d^k \max\{w,1\}^k}{\min\{u,1\}^{k/2}}$$

where, as noted above, $\bar{y}_{\boldsymbol{b}_j}$ does not depend on $x$. Here $C_{\boldsymbol{b_1},\dots,\boldsymbol{b_j}}$ is some constant which depends only on the partition $\{\boldsymbol{b_1},\dots,\boldsymbol{b_j}\}$ and is independent of $x$. On the other hand, we can also obtain a simple lower bound on $q(x)$. Observe that $q(x)$ is continuous and always positive. Recall that $b = \sup_{z\in\mathcal{Z}} \|\mu_z\|$. Thus,

$$q(x) = \int_{\mathcal{Z}} g(x|z)\mathrm{d}\Pi(z)$$

$$\geq \frac{1}{(2\pi)^{d/2} \sup_{z\in\mathcal{Z}} \det(\Sigma_z)^{\frac{1}{2}}} \int_{\mathcal{Z}} \exp\left(-\frac{1}{2}\sup_{z\in\mathcal{Z}}(x-\mu_z)^{\mathsf{T}}\Sigma_z^{-1}(x-\mu_z)\right) \mathrm{d}\Pi(z)$$

$$\geq \frac{1}{(2\pi)^{d/2} \sup_{z\in\mathcal{Z}} \det(\Sigma_z)^{\frac{1}{2}}} \int_{\mathcal{Z}} \exp\left(-\frac{1}{2}\sup_{z\in\mathcal{Z}}\left\|\Sigma_z^{-1}\right\|\left(\|x\|^2 + \|\mu_z\|^2\right)\right) \mathrm{d}\Pi(z)$$

$$\geq \frac{1}{(2\pi)^{d/2} \sup_{z\in\mathcal{Z}} \det(\Sigma_z)^{\frac{1}{2}}} \int_{\mathcal{Z}} \exp\left(-\frac{1}{2}\sup_{z\in\mathcal{Z}}\left\|\Sigma_z^{-1}\right\|\left(\|x\|^2 + b^2\right)\right) \mathrm{d}\Pi(z)$$

$$\geq \frac{1}{(2\pi)^{d/2}U} \exp\left(-\frac{V}{2}(\|x\|^2 + b^2)\right).$$

Therefore, if we set $C := \max_{\boldsymbol{b_1},\dots,\boldsymbol{b_j}} C_{\boldsymbol{b_1},\dots,\boldsymbol{b_j}}$, we obtain

$$\left|\partial_{\boldsymbol{a}}^k \log q(x)\right| \leq C^k B_k \frac{d^{2k}\max\{w,1\}^k}{\min\{u,1\}^{k/2}} U^k e^{k\frac{V}{2}(\|x\|^2+b^2)}. \tag{62}$$

The upper bound above, though it depends only on parameters $u, U, V, w$, has an exponential dependency on $x$, which is not desirable. We next derive a more refined bound in $x$. For brevity of analysis, we re-express $r$ (defined in (61)) to avoid empty partitions:

$$r_{\boldsymbol{b_1},\dots,\boldsymbol{b_k}}(x) = q(x)^{-p} \prod_{j=1}^p \partial_{\boldsymbol{b_j}}^{|\boldsymbol{b_j}|} q(x), \text{ s.t. } |\boldsymbol{b_{p+1}}| = \dots = |\boldsymbol{b_k}| = 0.$$

Now, by the boundedness of $\|\mu_z\|$ and $\left\|\Sigma_z^{-1/2}\right\|$ on $\mathcal{Z}$, for each $x$, there exist (bounded) $\bar{\Sigma}_{\boldsymbol{b_j}}$ and $\bar{\mu}_{\boldsymbol{b_j}}$ such that, $\forall z \in \mathcal{Z}$,

$$\sum_{\boldsymbol{b}\in[d]^{|\boldsymbol{b_j}|}} \left|\mathrm{poly}_{|\boldsymbol{b_j}|}\left(\Sigma_z^{-\frac{1}{2}}(x-\mu_z)\right)\right| \leq \sum_{\boldsymbol{b}\in[d]^{|\boldsymbol{b_j}|}} \left|\mathrm{poly}_{|\boldsymbol{b_j}|}\left(\bar{\Sigma}_{\boldsymbol{b_j}}^{-\frac{1}{2}}(x-\bar{\mu}_{\boldsymbol{b_j}})\right)\right| < \infty.$$

Then, following from (59), we obtain

$$|r_{\boldsymbol{b_1},\dots,\boldsymbol{b_k}}(x)| = q(x)^{-p} \left|\prod_{j=1}^p \partial_{\boldsymbol{b_j}}^{|\boldsymbol{b_j}|} q(x)\right| \leq q(x)^{-p} \prod_{j=1}^p \left|\partial_{\boldsymbol{b_j}}^{|\boldsymbol{b_j}|} q(x)\right|$$

$$\leq \frac{(d\max\{w,1\})^{\sum_{j=1}^p |\boldsymbol{b_j}|}}{u^{p/2}} \times$$

$$\prod_{j=1}^p \frac{\int \det(\Sigma_z)^{-\frac{1}{2}} \sum_{\boldsymbol{c_j}\in[d]^{|\boldsymbol{b_j}|}} \left|\mathrm{poly}_{|\boldsymbol{b_j}|}\left(\Sigma_z^{-\frac{1}{2}}(x-\mu_z)\right)\right| \phi\left(\Sigma_z^{-\frac{1}{2}}(x-\mu_z)\right) \mathrm{d}\Pi(z)}{\int \det(\Sigma_z)^{-\frac{1}{2}} \phi\left(\Sigma_z^{-\frac{1}{2}}(x-\mu_z)\right) \mathrm{d}\Pi(z)}$$

$$\leq \frac{(d\max\{w,1\})^k}{\min\{1,u\}^{k/2}} \times$$

$$\prod_{j=1}^{p} \frac{\sum_{\boldsymbol{c_j} \in [d]^{|\boldsymbol{b_j}|}} \left| \text{poly}_{|\boldsymbol{b_j}|} \left( \bar{\Sigma}_{\boldsymbol{b_j}}^{-\frac{1}{2}} (x - \bar{\mu}_{\boldsymbol{b_j}}) \right) \right| \left( \int \det(\Sigma_z)^{-\frac{1}{2}} \phi \left( \Sigma_z^{-\frac{1}{2}} (x - \mu_z) \right) d\Pi(z) \right)}{\int \det(\Sigma_z)^{-\frac{1}{2}} \phi \left( \Sigma_z^{-\frac{1}{2}} (x - \mu_z) \right) d\Pi(z)}$$

$$= \frac{(d \max\{w, 1\})^k}{\min\{1, u\}^{k/2}} \prod_{j=1}^{p} \sum_{\boldsymbol{c_j} \in [d]^{|\boldsymbol{b_j}|}} \left| \text{poly}_{|\boldsymbol{b_j}|} \left( \bar{\Sigma}_{\boldsymbol{b_j}}^{-\frac{1}{2}} (x - \bar{\mu}_{\boldsymbol{b_j}}) \right) \right|$$

Note that for each $j$, the number of terms in the summation above is upper-bounded by $d^{|\boldsymbol{b_j}|}$. Thus, expanding the product of summations would result in no more than $\prod_{j=1}^{p} d^{|\boldsymbol{b_j}|} = d^k$ terms. Also, since $\left| \text{poly}_{k_1}(y) \right| \cdot \left| \text{poly}_{k_2}(y) \right| = \left| \text{poly}_{k_1 + k_2}(y) \right|$, and since any $\bar{\Sigma}_{\boldsymbol{b_j}}^{-\frac{1}{2}} (x - \bar{\mu}_{\boldsymbol{b_j}})$ is linear in $x$ and independent in $z$, each product term is a $k$-th order polynomial in $x$. Therefore, we obtain

$$|r_{\boldsymbol{b_1}, \ldots, \boldsymbol{b_k}}(x)| \leq \frac{d^{2k} \max\{w, 1\}^k}{\min\{1, u\}^{k/2}} \max_{\boldsymbol{c_j} \in [d]^{|\boldsymbol{b_j}|}, \forall j=1,\ldots,p} |\text{poly}_k(x)|$$

and thus

$$\left| \partial_{\boldsymbol{a}}^k \log q(x) \right| \leq B_k \frac{d^{2k} \max\{w, 1\}^k}{\min\{1, u\}^{k/2}} \max_{\boldsymbol{b_1}, \ldots, \boldsymbol{b_k}} \max_{\boldsymbol{c_j} \in [d]^{|\boldsymbol{b_j}|}, \forall j=1,\ldots,p} |\text{poly}_k(x)|. \tag{63}$$

We have thus identified an upper bound on $\left| \partial_{\boldsymbol{a}}^k \log q(x) \right|$ which is polynomial in $x$. The proof is now complete by combining (62) and (63).

## H.2 PROOF OF LEMMA 14

We first identify $u, U, V, w$ for $\Sigma_{t,n}$ such that they are independent of $T$ and $k$ for all $t \geq 1$. Fix $t \geq 1$. We use the fact that $\Sigma_{t,n} = \bar{\alpha}_t \Sigma_{0,n} + (1 - \bar{\alpha}_t) I_d$. If we let $\lambda_{n,1} \geq \cdots \geq \lambda_{n,d} > 0$ as the eigenvalues of $\Sigma_{0,n}$ (which do not depend on $T$), the eigenvalues of $\Sigma_{t,n}$ are $\{\bar{\alpha}_t \lambda_{n,i} + (1 - \bar{\alpha}_t)\}_{i=1}^d$. Therefore, for any $n = 1, \ldots, N$ and $t \geq 1$,

$$(u :=) \prod_{i=1}^{d} \min\{\min_n \lambda_{n,i}, 1\} \leq \det(\Sigma_{t,n}) \leq \prod_{i=1}^{d} \max\{\max_n \lambda_{n,i}, 1\} (=: U).$$

Also, following from (45), we have $V := \frac{1}{\min\{1, \min_n \lambda_{n,d}\}}$. Next, write the eigen-decomposition as $\Sigma_{0,n} = Q_n \text{diag}(\lambda_{n,1}, \ldots, \lambda_{n,d}) Q_n^{\mathsf{T}}$, where $Q_n$ here is an orthonormal matrix (that does not depend on $T$). Then, for any $t \geq 1$,

$$\Sigma_{t,n}^{-\frac{1}{2}} = Q_n (\bar{\alpha}_t \text{diag}(\lambda_{n,1}, \ldots, \lambda_{n,d}) + (1 - \bar{\alpha}_t) I_d)^{-\frac{1}{2}} Q_n^{\mathsf{T}}$$
$$= Q_n \text{diag}((\bar{\alpha}_t \lambda_{n,1} + (1 - \bar{\alpha}_t))^{-\frac{1}{2}}, \ldots, (\bar{\alpha}_t \lambda_{n,d} + (1 - \bar{\alpha}_t))^{-\frac{1}{2}}) Q_n^{\mathsf{T}}$$

and thus, for all $t \geq 1$,

$$[\Sigma_{t,n}^{-\frac{1}{2}}]^{ij} = \sum_{k=1}^{d} (\bar{\alpha}_t \lambda_{n,k} + (1 - \bar{\alpha}_t))^{-\frac{1}{2}} Q_n^{ik} Q_n^{kj}$$

$$\leq (\min\{1, \min_n \lambda_{n,d}\})^{-\frac{1}{2}} \max_{n \in [N], i,j \in [d]} \left| \sum_{k=1}^{d} Q_n^{ik} Q_n^{kj} \right| =: w.$$

Since the identified $u, U, V, w$ are all independent of $T$ and $k$, by Lemma 13 we have obtained an upper bound on $\left| \partial_{\boldsymbol{a}}^k \log q(x) \right|$ for any fixed $x$ which is independent of $T$. Thus,

$$(1 - \alpha_t)^{k/2} \mathbb{E}_{X_t \sim Q_t} \left| \partial_{\boldsymbol{a}}^k \log q_t(X_t) \right|, \quad (1 - \alpha_t)^{k/2} \mathbb{E}_{X_t \sim Q_t} \left| \partial_{\boldsymbol{a}}^k \log q_{t-1}(\mu_t(X_t)) \right|$$
$$= \tilde{O} \left( (1 - \alpha_t)^{k/2} \right) = \tilde{O} \left( \frac{1}{T^{k/2}} \right).$$

Hence, we have shown Assumption 5.

### H.3 PROOF OF LEMMA 15

Fix $t \geq 1$. We will draw some notations introduced in Lemma 13. Specifically, we recall from (61) that

$$
\begin{aligned}
\partial_{\boldsymbol{a}}^p \log q_t(x_t) &= q_t(x_t)^{-p} \sum_{\boldsymbol{b_1},\ldots,\boldsymbol{b_p}} \prod_{j=1}^p \partial_{\boldsymbol{b_j}}^{|\boldsymbol{b_j}|} q_t(x_t) \\
&= q_t(x_t)^{-p} \sum_{\boldsymbol{b_1},\ldots,\boldsymbol{b_p}} \prod_{j=1}^p \int_{x_0} q_{t|0}(x_t|x_0) \mathrm{poly}_{|\boldsymbol{b_j}|}\left(\frac{x_t - \sqrt{\bar{\alpha}_t}x_0}{1 - \bar{\alpha}_t}\right) \mathrm{d}Q_0(x_0) \\
&= \sum_{\boldsymbol{b_1},\ldots,\boldsymbol{b_p}} \frac{1}{(1 - \bar{\alpha}_t)^{\frac{p}{2}}} \prod_{j=1}^p \int_{x_0} \mathrm{poly}_{|\boldsymbol{b_j}|}\left(\frac{x_t - \sqrt{\bar{\alpha}_t}x_0}{\sqrt{1 - \bar{\alpha}_t}}\right) \mathrm{d}Q_{0|t}(x_0|x_t) \quad (64)
\end{aligned}
$$

in which we have defined $\mathrm{poly}_k(y)$ as a $k$-th order polynomial function in $y_1,\ldots,y_d$. Recall that here $\{\boldsymbol{b_1},\ldots,\boldsymbol{b_p}\}$ is some (possibly empty) partition of $\boldsymbol{a}$, i.e., $\sum_j \boldsymbol{b_j} = \boldsymbol{a}$ and $\sum_j |\boldsymbol{b_j}| = p$.

Thus,

$$
\begin{aligned}
&\mathbb{E}_{X_t \sim Q_t} |\partial_{\boldsymbol{a}}^p \log q_t(X_t)|^\ell \\
&\leq \frac{1}{(1 - \bar{\alpha}_t)^{\frac{p\ell}{2}}} p^\ell \sum_{\boldsymbol{b_1},\ldots,\boldsymbol{b_p}} \mathbb{E}_{X_t \sim Q_t}\left[\prod_{j=1}^p \left|\int_{x_0} \mathrm{poly}_{|\boldsymbol{b_j}|}\left(\frac{X_t - \sqrt{\bar{\alpha}_t}x_0}{\sqrt{1 - \bar{\alpha}_t}}\right) \mathrm{d}Q_{0|t}(x_0|X_t)\right|^\ell\right] \\
&\overset{(i)}{\leq} \frac{1}{(1 - \bar{\alpha}_t)^{\frac{p\ell}{2}}} p^\ell \sum_{\boldsymbol{b_1},\ldots,\boldsymbol{b_p}} \prod_{j=1}^p \left(\mathbb{E}_{X_t \sim Q_t}\left|\int_{x_0} \mathrm{poly}_{|\boldsymbol{b_j}|}\left(\frac{X_t - \sqrt{\bar{\alpha}_t}x_0}{\sqrt{1 - \bar{\alpha}_t}}\right) \mathrm{d}Q_{0|t}(x_0|X_t)\right|^{\frac{p\ell}{|\boldsymbol{b_j}|}}\right)^{\frac{|\boldsymbol{b_j}|}{p}} \\
&\overset{(ii)}{\leq} \frac{1}{(1 - \bar{\alpha}_t)^{\frac{p\ell}{2}}} p^\ell \sum_{\boldsymbol{b_1},\ldots,\boldsymbol{b_p}} \prod_{j=1}^p \left(\mathbb{E}_{X_0,X_t \sim Q_{0,t}}\left|\mathrm{poly}_{|\boldsymbol{b_j}|}\left(\frac{X_t - \sqrt{\bar{\alpha}_t}X_0}{\sqrt{1 - \bar{\alpha}_t}}\right)\right|^{\frac{p\ell}{|\boldsymbol{b_j}|}}\right)^{\frac{|\boldsymbol{b_j}|}{p}} \\
&= \frac{1}{(1 - \bar{\alpha}_t)^{\frac{p\ell}{2}}} p^\ell \sum_{\boldsymbol{b_1},\ldots,\boldsymbol{b_p}} \prod_{j=1}^p \left(\mathbb{E}\left|\mathrm{poly}_{|\boldsymbol{b_j}|}(Z)\right|^{\frac{p\ell}{|\boldsymbol{b_j}|}}\right)^{\frac{|\boldsymbol{b_j}|}{p}} \\
&\lesssim \frac{d^{\frac{p\ell}{2}}}{(1 - \bar{\alpha}_t)^{\frac{p\ell}{2}}}
\end{aligned}
$$

where $Z \sim \mathcal{N}(0, I_d)$ is a standard Gaussian random variable (that does not depend on $T$ here) and any $r$-th order of polynomial of $Z_1,\ldots,Z_d$ has finite expectation (that does not depend on $T$ and with at most $d^{r/2}$ dimensional dependency). Here $(i)$ holds by Hölder's inequality, and $(ii)$ holds by Jensen's inequality since $p\ell/|\boldsymbol{b_j}| \geq 1$ for all $\boldsymbol{b_j}$ and $\ell \geq 1$. The proof is now complete.

### H.4 PROOF OF LEMMA 16

Fix $t \geq 2$. We first introduce the following notations. Write $\mu_t = \mu_t(x_t)$. Let $Q_{\mu_t}$ be the distribution of $\mu_t(X_t)$ where $X_t \sim Q_t$, and let $q_{\mu_t}$ be the corresponding p.d.f. (w.r.t. the Lebesgue measure). Let $Q_{\mu_t,x_0}$ be the joint distribution of $\mu_t$ and $x_0$.

Now, we can re-write the integral as

$$
\begin{aligned}
&\int_{x_0,x_t} \|\mu_t(x_t) - \sqrt{\bar{\alpha}_{t-1}}x_0\|^p \, \mathrm{d}Q_{0|t-1}(x_0|\mu_t(x_t)) \mathrm{d}Q_t(x_t) \\
&= \int_{x_0,\mu_t} \|\mu_t - \sqrt{\bar{\alpha}_{t-1}}x_0\|^p \, \mathrm{d}Q_{0|t-1}(x_0|\mu_t) \mathrm{d}Q_{\mu_t}(\mu_t) \\
&= \int_{x_0,\mu_t} \|\mu_t - \sqrt{\bar{\alpha}_{t-1}}x_0\|^p \frac{q_{\mu_t}(\mu_t)}{q_{t-1}(\mu_t)} \, \mathrm{d}Q_{0|t-1}(x_0|\mu_t) \mathrm{d}Q_{t-1}(\mu_t)
\end{aligned}
$$

$$\leq \sqrt{\int_{x_0,\mu_t} \left\| \mu_t - \sqrt{\bar{\alpha}_{t-1}} x_0 \right\|^{2p} \mathrm{d}Q_{0|t-1}(x_0|\mu_t)\mathrm{d}Q_{t-1}(\mu_t)}$$

$$\times \sqrt{\int_{x_0,\mu_t} \left( \frac{q_{\mu_t}(\mu_t)}{q_{t-1}(\mu_t)} \right)^2 \mathrm{d}Q_{0|t-1}(x_0|\mu_t)\mathrm{d}Q_{t-1}(\mu_t)} \tag{65}$$

where the last line follows from Cauchy-Schwartz inequality.

Now, for the first term of (65) we recovered the matched moment, and we have

$$\sqrt{\int_{x_0,\mu_t} \left\| \mu_t - \sqrt{\bar{\alpha}_{t-1}} x_0 \right\|^{2p} \mathrm{d}Q_{0|t-1}(x_0|\mu_t)\mathrm{d}Q_{t-1}(\mu_t)}$$

$$= \sqrt{\int_{x_0,x_{t-1}} \left\| x_{t-1} - \sqrt{\bar{\alpha}_{t-1}} x_0 \right\|^{2p} \mathrm{d}Q_{0,t-1}(x_0, x_{t-1})}$$

$$= (1 - \bar{\alpha}_{t-1})^{\frac{p}{2}} \sqrt{\int_{x_0,x_{t-1}} \left\| \frac{x_{t-1} - \sqrt{\bar{\alpha}_{t-1}} x_0}{\sqrt{1 - \bar{\alpha}_{t-1}}} \right\|^{2p} \mathrm{d}Q_{0,t-1}(x_0, x_{t-1})}$$

$$= (1 - \bar{\alpha}_{t-1})^{\frac{p}{2}} \sqrt{\mathbb{E} \left\| Z \right\|^{2p}} \lesssim d^{\frac{p}{2}} (1 - \bar{\alpha}_{t-1})^{\frac{p}{2}}$$

where $Z \sim \mathcal{N}(0, I_d)$ is a Gaussian random variable.

Now we upper bound the second term in (65), whose square is equal to

$$\int_{x_0,\mu_t} \left( \frac{q_{\mu_t}(\mu_t)}{q_{t-1}(\mu_t)} \right)^2 \mathrm{d}Q_{0|t-1}(x_0|\mu_t)\mathrm{d}Q_{t-1}(\mu_t)$$

$$= \int_{x_{t-1}} \left( \frac{q_{\mu_t}(x_{t-1})}{q_{t-1}(x_{t-1})} \right)^2 q_{t-1}(x_{t-1})\mathrm{d}x_{t-1}$$

$$= 1 + \chi^2(Q_{\mu_t}\|Q_{t-1})$$

$$\overset{(i)}{\leq} 1 + \chi^2(Q_{\mu_t,x_0}\|Q_{t-1,0})$$

$$= \int_{x_0} \left( \int_{\mu_t} \left( \frac{q_{\mu_t|x_0}(\mu_t|x_0)}{q_{t-1|0}(\mu_t|x_0)} \right)^2 q_{t-1|0}(\mu_t|x_0)\mathrm{d}\mu_t \right) \mathrm{d}Q_0(x_0)$$

$$= \int_{x_0} \left( \int_{x_t} \frac{(q_{t|0}(x_t|x_0))^2}{q_{t-1|0}(\mu_t(x_t)|x_0)} \det \left( \frac{\mathrm{d}\mu_t(x_t)}{\mathrm{d}x_t} \right)^{-1} \mathrm{d}x_t \right) \mathrm{d}Q_0(x_0)$$

$$\overset{(ii)}{\leq} \sqrt{\int_{x_0,x_t} \left( \frac{q_{t|0}(x_t|x_0)}{q_{t-1|0}(\mu_t(x_t)|x_0)} \right)^2 \mathrm{d}Q_{t,0}(x_t, x_0)} \times$$

$$\sqrt{\int_{x_0,x_t} \det \left( \frac{\mathrm{d}\mu_t(x_t)}{\mathrm{d}x_t} \right)^{-2} \mathrm{d}Q_{t,0}(x_t, x_0)}$$

where $\chi^2(P\|Q)$ is the chi-squared divergence between $P$ and $Q$. Here $(i)$ follows from the data processing inequality for f-divergence, and $(ii)$ again follows from Cauchy-Schwartz inequality. We can calculate the determinant term above as

$$\det \left( \frac{\mathrm{d}\mu_t}{\mathrm{d}x_t} \right)^{-2} = \det \left( \frac{1}{\sqrt{\alpha_t}} I_d + \frac{1 - \alpha_t}{\sqrt{\alpha_t}} \nabla^2 \log q_t(x_t) \right)^{-2}$$

$$= \left( \frac{1}{\alpha_t^{\frac{d}{2}}} \left( 1 + (1 - \alpha_t)\mathrm{Tr}(\nabla^2 \log q_t(x_t)) + \epsilon_T(x_t) \right) \right)^{-2}$$

$$\leq \alpha_t^{\frac{d}{2}} \left( 1 - 2(1 - \alpha_t)\mathrm{Tr}(\nabla^2 \log q_t(x_t)) + \epsilon_T(x_t) \right)$$

where we denote the residual terms as $\epsilon_T(x_t) := \sum_{p=2}^{\infty}(1-\alpha_t)^p \sum_{\boldsymbol{I}:|\boldsymbol{I}|=p} c_{\boldsymbol{I}} \prod_{(i,j)\in\boldsymbol{I}} \partial_{ij}^2 \log q_t(x_t)$, where $c_{\boldsymbol{I}}$ is some coefficient that does not depend on $T$. Since from Lemma 15,

$$\mathbb{E}_{X_t\sim Q_t} \left|\partial_{ij}^2 \log q_t(X_t)\right|^{\ell} = \tilde{O}\left(\frac{1}{(1-\bar{\alpha}_t)^{\ell}}\right), \quad \forall i,j \in [d], \forall \ell \geq 1,$$

and note that $\frac{1-\alpha_t}{1-\bar{\alpha}_t} = \tilde{O}\left(\frac{\log T}{T}\right)$ with the $\alpha_t$ in (10), we have that

$$
\begin{aligned}
\mathbb{E}_{X_t\sim Q_t} |\epsilon_T(X_t)| &\leq \sum_{p=2}^{\infty}(1-\alpha_t)^p \sum_{\boldsymbol{I}:|\boldsymbol{I}|=p} c_{\boldsymbol{I}} \mathbb{E}_{X_t\sim Q_t} \prod_{(i,j)\in\boldsymbol{I}} \left|\partial_{ij}^2 \log q_t(X_t)\right| \\
&\leq \sum_{p=2}^{\infty}(1-\alpha_t)^p \sum_{\boldsymbol{I}:|\boldsymbol{I}|=p} c_{\boldsymbol{I}} \prod_{(i,j)\in\boldsymbol{I}} \left(\mathbb{E}_{X_t\sim Q_t} \left|\partial_{ij}^2 \log q_t(X_t)\right|^p\right)^{\frac{1}{p}} \\
&= \sum_{p=2}^{\infty} \tilde{O}\left(\frac{(1-\alpha_t)^p}{(1-\bar{\alpha}_t)^p}\right) \\
&= \tilde{O}\left(\frac{(\log T)^2}{T^2}\right),
\end{aligned}
$$

and thus

$$\mathbb{E}_{X_t\sim Q_t} \det\left(\frac{\mathrm{d}\mu_t}{\mathrm{d}x_t}\right)^{-2} = \alpha_t^{\frac{d}{2}} + \tilde{O}\left(\frac{\log T}{T}\right) \leq 1 + \tilde{O}\left(\frac{\log T}{T}\right).$$

Also, since

$$
\begin{aligned}
\left(\frac{q_{t|0}(x_t|x_0)}{q_{t-1|0}(\mu_t|x_0)}\right)^2 &= \frac{\frac{1}{(1-\bar{\alpha}_t)^d} \exp\left(-\frac{\left\|x_t - \sqrt{\bar{\alpha}_t}x_0\right\|^2}{1-\bar{\alpha}_t}\right)}{\frac{1}{(1-\bar{\alpha}_{t-1})^d} \exp\left(-\frac{\left\|x_t + (1-\alpha_t)\nabla\log q_t(x_t) - \sqrt{\bar{\alpha}_t}x_0\right\|^2}{\alpha_t - \bar{\alpha}_t}\right)} \\
&= \left(\frac{1-\bar{\alpha}_{t-1}}{1-\bar{\alpha}_t}\right)^d \exp\left(\left\|x_t - \sqrt{\bar{\alpha}_t}x_0\right\|^2 \left(\frac{1}{\alpha_t - \bar{\alpha}_t} - \frac{1}{1-\bar{\alpha}_t}\right)\right) \times \\
&\quad \exp\left(\frac{2(1-\alpha_t)\nabla\log q_t(x_t)^{\mathsf{T}}(x_t - \sqrt{\bar{\alpha}_t}x_0) + (1-\alpha_t)^2\left\|\nabla\log q_t(x_t)\right\|^2}{\alpha_t - \bar{\alpha}_t}\right) \\
&\overset{(iii)}{\leq} \exp\left(\left\|\frac{x_t - \sqrt{\bar{\alpha}_t}x_0}{\sqrt{1-\bar{\alpha}_t}}\right\|^2 \frac{1-\alpha_t}{\alpha_t - \bar{\alpha}_t}\right) \times \\
&\quad \exp\left(\frac{2(1-\alpha_t)\nabla\log q_t(x_t)^{\mathsf{T}}(x_t - \sqrt{\bar{\alpha}_t}x_0) + (1-\alpha_t)^2\left\|\nabla\log q_t(x_t)\right\|^2}{\alpha_t - \bar{\alpha}_t}\right) \\
&\overset{(iv)}{=} \exp\left(\left\|\frac{x_t - \sqrt{\bar{\alpha}_t}x_0}{\sqrt{1-\bar{\alpha}_t}}\right\|^2 \frac{1-\alpha_t}{\alpha_t - \bar{\alpha}_t}\right) \times \\
&\quad \left(1 + \tilde{O}\left(\frac{(1-\alpha_t)\nabla\log q_t(x_t)^{\mathsf{T}}(x_t - \sqrt{\bar{\alpha}_t}x_0) + (1-\alpha_t)^2\left\|\nabla\log q_t(x_t)\right\|^2}{\alpha_t - \bar{\alpha}_t}\right)\right)
\end{aligned}
$$

where $(iii)$ follows because $\frac{1-\bar{\alpha}_{t-1}}{1-\bar{\alpha}_t} < 1$, and $(iv)$ follows because $e^z = 1 + \tilde{O}(z)$ when $z \to 0$ and because $\frac{1-\alpha_t}{\alpha_t - \bar{\alpha}_t}, \frac{1-\alpha_t}{1-\bar{\alpha}_t} = \tilde{O}\left(\frac{\log T}{T}\right)$ with the $\alpha_t$ in (10). Thus,

$$\mathbb{E}_{X_t,X_0\sim Q_{t,0}} \left(\frac{q_{t|0}(X_t|X_0)}{q_{t-1|0}(\mu_t(X_t)|X_0)}\right)^2$$

$$\leq \sqrt{\mathbb{E}_{X_t,X_0\sim Q_{t,0}} \exp\left(2\left\|\frac{X_t - \sqrt{\bar{\alpha}_t}X_0}{\sqrt{1-\bar{\alpha}_t}}\right\|^2 \frac{1-\alpha_t}{\alpha_t - \bar{\alpha}_t}\right)} \times$$

$$\sqrt{1 + \tilde{O}\left(\mathbb{E}_{X_t, X_0 \sim Q_{t,0}}\left[\frac{(1-\alpha_t)\|\nabla \log q_t(x_t)\| \|x_t - \sqrt{\bar{\alpha}_t}x_0\| + (1-\alpha_t)^2 \|\nabla \log q_t(x_t)\|^2}{\alpha_t - \bar{\alpha}_t}\right]\right)}$$

$$\stackrel{(v)}{=} \sqrt{\mathbb{E}_{X_t, X_0 \sim Q_{t,0}} \exp\left(2\left\|\frac{X_t - \sqrt{\bar{\alpha}_t}X_0}{\sqrt{1-\bar{\alpha}_t}}\right\|^2 \frac{1-\alpha_t}{\alpha_t - \bar{\alpha}_t}\right) \times \left(1 + \tilde{O}\left(\frac{\log T}{T}\right)\right)}$$

where $(v)$ follows from Lemma 15 and Cauchy-Schwartz inequality, and

$$\mathbb{E}_{X_t, X_0 \sim Q_{t,0}} \exp\left(2\left\|\frac{X_t - \sqrt{\bar{\alpha}_t}X_0}{\sqrt{1-\bar{\alpha}_t}}\right\|^2 \frac{1-\alpha_t}{\alpha_t - \bar{\alpha}_t}\right)$$

$$= \frac{1}{(2\pi)^{\frac{d}{2}}} \int_z e^{2\frac{1-\alpha_t}{\alpha_t - \bar{\alpha}_t}\|z\|^2 - \frac{1}{2}\|z\|^2} dz$$

$$= \frac{1}{(2\pi)^{\frac{d}{2}}} \int_z e^{-\frac{1}{2}\|z\|^2(1+\tilde{O}(\log T/T))} dz$$

$$= 1 + \tilde{O}\left(\frac{\log T}{T}\right).$$

Therefore, we arrive at a bound for the second term in (65):

$$\sqrt{\int_{x_0, \mu_t} \left(\frac{q_{\mu_t}(\mu_t)}{q_{t-1}(\mu_t)}\right)^2 dQ_{0|t-1}(x_0|\mu_t)dQ_{t-1}(\mu_t)} \leq 1 + \tilde{O}\left(\frac{\log T}{T}\right).$$

and the lemma follows immediately.

## H.5 PROOF OF LEMMA 17

Fix $t \geq 2$. From (64), we also have

$$\mathbb{E}_{X_t \sim Q_t} |\partial_a^p \log q_{t-1}(\mu_t(X_t))|^\ell$$

$$\leq \frac{1}{(1-\bar{\alpha}_{t-1})^{\frac{p\ell}{2}}} p^\ell \sum_{\boldsymbol{b_1}, \ldots, \boldsymbol{b_p}} \mathbb{E}_{X_t \sim Q_t}\left[\prod_{j=1}^p \left|\int_{x_0} \text{poly}_{|\boldsymbol{b_j}|}\left(\frac{\mu_t(X_t) - \sqrt{\bar{\alpha}_{t-1}}x_0}{\sqrt{1-\bar{\alpha}_{t-1}}}\right) dQ_{0|t-1}(x_0|\mu_t(X_t))\right|^\ell\right]$$

$$\leq \frac{1}{(1-\bar{\alpha}_{t-1})^{\frac{p\ell}{2}}} p^\ell \sum_{\boldsymbol{b_1}, \ldots, \boldsymbol{b_p}} \prod_{j=1}^p \left(\mathbb{E}_{X_t \sim Q_t}\left|\int_{x_0} \text{poly}_{|\boldsymbol{b_j}|}\left(\frac{\mu_t(X_t) - \sqrt{\bar{\alpha}_{t-1}}x_0}{\sqrt{1-\bar{\alpha}_{t-1}}}\right) dQ_{0|t-1}(x_0|\mu_t(X_t))\right|^{\frac{p\ell}{|\boldsymbol{b_j}|}}\right)^{\frac{|\boldsymbol{b_j}|}{p}}$$

$$\leq \frac{1}{(1-\bar{\alpha}_{t-1})^{\frac{p\ell}{2}}} p^\ell \sum_{\boldsymbol{b_1}, \ldots, \boldsymbol{b_p}} \prod_{j=1}^p \left(\mathbb{E}_{X_t \sim Q_t} \int_{x_0} \left|\text{poly}_{|\boldsymbol{b_j}|}\left(\frac{\mu_t(X_t) - \sqrt{\bar{\alpha}_{t-1}}x_0}{\sqrt{1-\bar{\alpha}_{t-1}}}\right)\right|^{\frac{p\ell}{|\boldsymbol{b_j}|}} dQ_{0|t-1}(x_0|\mu_t(X_t))\right)^{\frac{|\boldsymbol{b_j}|}{p}}$$

$$\leq \frac{1}{(1-\bar{\alpha}_{t-1})^{\frac{p\ell}{2}}} p^\ell \sum_{\boldsymbol{b_1}, \ldots, \boldsymbol{b_p}} \max_{j \in [p]} \mathbb{E}_{X_t \sim Q_t} \int_{x_0} \left|\text{poly}_{p\ell}\left(\frac{\mu_t(X_t) - \sqrt{\bar{\alpha}_{t-1}}x_0}{\sqrt{1-\bar{\alpha}_{t-1}}}\right)\right| dQ_{0|t-1}(x_0|\mu_t(X_t))$$

$$\lesssim \frac{1}{(1-\bar{\alpha}_{t-1})^{\frac{p\ell}{2}}} \cdot \mathbb{E}_{X_t \sim Q_t} \int_{x_0} \left\|\frac{\mu_t(X_t) - \sqrt{\bar{\alpha}_{t-1}}x_0}{\sqrt{1-\bar{\alpha}_{t-1}}}\right\|^{p\ell} dQ_{0|t-1}(x_0|\mu_t(X_t))$$

$$\lesssim \frac{d^{\frac{p\ell}{2}}}{(1-\bar{\alpha}_{t-1})^{\frac{p\ell}{2}}}$$

where the last line follows from Lemma 16. Now, together with Lemma 15, Assumption 5 is established noting that $\frac{1-\alpha_t}{1-\bar{\alpha}_{t-1}} = \tilde{O}\left(\frac{\log T}{T}\right) = \tilde{O}(1-\alpha_t)$ for all $t \geq 2$.

## H.6 PROOF OF LEMMA 18

Recall the expansion of $\zeta'_{1,0}$ in (57). As in the proof of Lemma 11, with the choice of $\mu_1$ and $\Sigma_1$, we still have

$$\mathbb{E}_{X_0 \sim P'_{0|1}}[T_1] = \mathbb{E}_{X_0 \sim Q_{0|1}}[T_1],$$

$$\mathbb{E}_{X_0 \sim P'_{0|1}}[T'_2] = \mathbb{E}_{X_0 \sim Q_{0|1}}[T'_2].$$

Define $T'_3 := \frac{1}{3!} \sum_{i,j,k=1}^d \partial^3_{ijk} \log q_0(\mu^*_1)(x^i_0 - \mu^i_1)(x^j_0 - \mu^j_1)(x^k_0 - \mu^k_1)$. Here $\mu^*_1 = \mu^*_1(x_1, x_0)$ is a function of both $x_1$ and $x_0$. A useful result from Lemma 15 is that, with the $\alpha_t$ in (10), we have, $\forall i, j, k \in [d]$ and $\ell \geq 1$,

$$(1-\alpha_1)^\ell \mathbb{E}_{X_1 \sim Q_1} \left| \partial^2_{ij} \log q_1(X_1) \right|^\ell \lesssim \frac{(1-\alpha_1)^\ell d^\ell}{(1-\bar{\alpha}_1)^\ell} = d^\ell, \tag{66}$$

$$(1-\alpha_1)^3 \mathbb{E}_{X_1 \sim Q_1} \left| \partial^3_{ijk} \log q_1(X_1) \right|^2 \lesssim \frac{(1-\alpha_1)^3 d^3}{(1-\bar{\alpha}_1)^3} = d^3. \tag{67}$$

First, using Lemma 8, we have that

$\mathbb{E}_{X_0, X_1 \sim Q_{0,1}}[T'_3]$

$= \frac{(1-\alpha_1)^3}{3! \alpha_1^{3/2}} \sum_{i,j,k=1}^d \mathbb{E}_{X_0, X_1 \sim Q_{0,1}}[\partial^3_{ijk} \log q_0(\mu^*_1(X_1, X_0)) \partial^3_{ijk} \log q_1(X_1)]$

$\leq \frac{(1-\alpha_1)^3}{3! \alpha_1^{3/2}} \sqrt{\mathbb{E}_{X_0, X_1 \sim Q_{0,1}} \sum_{i,j,k=1}^d (\partial^3_{ijk} \log q_0(\mu^*_1(X_1, X_0)))^2} \sqrt{\mathbb{E}_{X_1 \sim Q_1} \sum_{i,j,k=1}^d (\partial^3_{ijk} \log q_1(X_1))^2}$

$\leq \frac{(1-\alpha_1)^3}{3! \alpha_1^{3/2}} dM \sqrt{\mathbb{E}_{X_1 \sim Q_1} \sum_{i,j,k=1}^d (\partial^3_{ijk} \log q_1(X_1))^2}.$

Here in the last line we have used a similar technique in (55), which assumes that $\nabla^2 \log q_0$ is 2-norm $M$-Lipschitz. Now, from (67) we have

$$\mathbb{E}_{X_0, X_1 \sim Q_{0,1}}[T'_3] \lesssim \frac{(1-\alpha_1)^{3/2}}{3! \alpha_1^{3/2}} d^4 M.$$

Also,

$\mathbb{E}_{\substack{X_0 \sim P'_{0|1} \\ X_1 \sim Q_1}}[T'_3]$

$= \frac{1}{3!} \sum_{i,j,k=1}^d \mathbb{E}_{\substack{X_0 \sim P'_{0|1} \\ X_1 \sim Q_1}} \left[ \partial^3_{ijk} \log q_0(\mu^*_1(X_1, X_0)) \prod_{c=i,j,k} (X^c_0 - \mu^c_1(X_1)) \right]$

$\overset{(i)}{\leq} \frac{1}{3!} dM \sqrt{\mathbb{E}_{\substack{X_0 \sim P'_{0|1} \\ X_1 \sim Q_1}} \|X_0 - \mu_1(X_1)\|^6}$

$\leq \frac{1}{3!} d^2 M \sqrt{\sum_{i=1}^d \mathbb{E}_{\substack{X_0 \sim P'_{0|1} \\ X_1 \sim Q_1}} \left( X^i_0 - \mu_1(X_1)^i \right)^6}$

$\overset{(ii)}{=} \frac{1}{3!} d^2 M \sqrt{\sum_{i=1}^d 15 \left( \frac{1-\alpha_1}{\alpha_1} \right)^3 \mathbb{E}_{X_1 \sim Q_1} (1 + (1-\alpha_1) \partial^2_{ii} \log q_1(X_1))^3}$

$\overset{(iii)}{\lesssim} \frac{(1-\alpha_1)^{3/2}}{3! \alpha_1^{3/2}} d^4 M$

where $(i)$ holds with a similar technique in (55) assuming $\nabla^2 \log q_0$ is $M$-Lipschitz, $(ii)$ holds by Lemma 7, and $(iii)$ holds by (66). The proof is now complete.

