# OpenReview forum: "Broadening Target Distributions for Accelerated Diffusion Models via a Novel Analysis Approach"
_ICLR.cc/2025/Conference — ICLR 2025 Poster_

### Official Review · Reviewer_ZY6t · 2024-10-30

**Soundness:** 4
**Presentation:** 3
**Contribution:** 2
**Rating:** 6
**Confidence:** 4

**Summary:**

This paper continues the study of proving convergence bounds for diffusion models under the assumption of accurate score estimation, with a focus on improved dependence on the target error. The "standard" rate achieved by DDPMs is an iteration complexity of $O(d/\epsilon^2)$ for sampling any distribution with finite variance to $\epsilon$ error in TV (after it has been perturbed with a small amount of noise), due to Benton et al. '23 and Conforti et al. '23. A number of recent works have tried to improve the $\epsilon$ dependence to $1/\epsilon$, but only in more specialized settings (e.g. assuming smoothness or bounded support). One of the main results of this work is to show such a result under only the assumption of finite variance. They achieve an iteration complexity of $O(d^{3/2}/\epsilon)$ using a diffusion-based sampler that has approximate access to both the score function *and its Jacobian*. In the smooth setting, they obtain samplers with similar scaling under one of two assumptions: 1) second-order of the smoothness, and 2) Gaussian mixture.

**Strengths:**

The result in the finite variance setting (Theorem 3) improves upon prior work in a few directions. Firstly, among works that achieve scaling $O(1/\epsilon)$ or better, theirs achieves the best dimension dependence. Secondly, unlike the most relevant preceding work of Li et al. '24c, the present work can tolerate score error. Finally, compared to those prior works, the present work applies in a more general setting where the distribution only has finite variance rather than bounded support.

The Gaussian mixture result is interesting as it requires proving new bounds on the higher-order derivatives of the score function for Gaussian mixtures, which are of independent interest.

**Weaknesses:**

One weakness is that the sampler requires approximate access to *higher-order* score functions (namely, the Jacobian of the score). While there are examples of diffusion generative models in practice that use higher-order scores essentially by applying automatic differentiation to the first-order score (e.g. GENIE), this is not standard and introduces significant computational overhead. Additionally, these models in practice are not actually trained according to some kind of explicit Hessian matching loss, so it is unclear why one should expect the Jacobian of the score estimates to be close to the Jacobian of the scores. That said, the prior work of Li et al. on acceleration, which appeared in a previous ICLR, also works with second-order access, so this weakness is somewhat justifiable.

Also, not a weakness per se, but note that the result for second-order smoothness (Theorem 4) is technically incomparable to the other previous works getting $1/\epsilon$ scaling because while it does not assume smoothness along the trajectory, it needs to assume that the *Hessian* of the log-density is Lipschitz. And as mentioned above, this result requires second-order scores whereas the probability flow ODE paper of Chen et al. only uses first-order scores.

Another note: while the authors are careful to clarify the chronology of the various related works, it is worth pointing out that in the last month or so (and before the ICLR deadline), there have been important developments in this literature that supersede some of the main results of this submission. For instance, [Li & Yan](https://arxiv.org/pdf/2409.18959) showed that even for the standard SDE-based sampler (without second-order score), one can achieve an accelerated rate of $O(d/\epsilon)$, in fact under only a *first* moment bound.

**Questions:**

- A recent work of [Gatmiry-Kelner-Lee](https://arxiv.org/abs/2404.18869) also established bounds on higher-order scores of Gaussian mixtures, though specifically in the context of identity-covariance components. How do their bounds compare to the ones proved in this submission?
- To get intuition for the assumption for Theorem 4: Is there an example of a second-order smooth data distribution for which the scores along the trajectory are not all first-order smooth?
- Perhaps it is worth including in the related work some discussion on the use of higher-order scores for diffusion models in practice like GENIE?

---

> ### Author Response · Authors · 2024-11-21
>
> We thank the reviewer for providing the highly inspiring feedback. In the revised paper, we have made changes based on your review comments and have highlighted all revisions in blue fonts.
>
> **Q1:** One weakness is that the sampler requires approximate access to higher-order score functions (namely, the Jacobian of the score). While there are examples of diffusion generative models in practice that use higher-order scores essentially by applying automatic differentiation to the first-order score (e.g. GENIE), this is not standard and introduces significant computational overhead. Additionally, these models in practice are not actually trained according to some kind of explicit Hessian matching loss, so it is unclear why one should expect the Jacobian of the score estimates to be close to the Jacobian of the scores. That said, the prior work of Li et al. on acceleration, which appeared in a previous ICLR, also works with second-order access, so this weakness is somewhat justifiable.
>
> **A1:** We appreciate that the reviewer justifies such a requirement in the context of the existing work. Compared to previous works, our accelerated sampler can be applied to a much larger class of distributions. In particular, we do not require the Lipschitz-smooth condition along the entire sampling path nor require the target to have bounded support. Further, within the class of non-smooth targets, our sampler achieves a better dimensional dependency than previous works.
>
> Numerically, our Fig. 1 demonstrates that our proposed sampling method substantially outperforms the previous design in [1]. Also, following the reviewer’s suggestion in Q6, we have added the work of GENIE [5] in the Related Works in the Appendix and have provided the paragraph to discuss that work. Our paper theoretically justifies the accelerated empirical performance of [6] in the regime when the Hessian of $\log q_t$ is well-estimated.
>
> **Q2:** Also, not a weakness per se, but note that the result for second-order smoothness (Theorem 4) is technically incomparable to the other previous works getting scaling because while it does not assume smoothness along the trajectory, it needs to assume that the Hessian of the log-density is Lipschitz. And as mentioned above, this result requires second-order scores whereas the probability flow ODE paper of Chen et al. only uses first-order scores.
>
> **A2:** We agree with the reviewer’s comment. Our results provide new contributions that complement existing studies by exploring different assumptions of distributions, which enriches the existing set of distributions studied in the literature. We have added this remark after Theorem 4.
>
> **Q3:** Another note: while the authors are careful to clarify the chronology of the various related works, it is worth pointing out that in the last month or so (and before the ICLR deadline), there have been important developments in this literature that supersede some of the main results of this submission. For instance, Li & Yan showed that even for the standard SDE-based sampler (without second-order score), one can achieve an accelerated rate of $O(d/\epsilon)$, in fact under only a first moment bound.
>
> **A3:** We appreciate the reviewer’s note, and we are also aware of these recent studies. However, we are following the ICLR guidelines that the paper does not need to discuss the research studies that appeared within the two months of the submission deadline, which clearly extends to the work posted after the ICLR deadline. Hence, we kindly ask the review to evaluate this paper based on the prior research, not the con-current and later research.
>
> Further, even given those recent new developments, we believe that our paper still holds significant new contributions in (i) the proposed Hessian-based accelerated sampler is still novel, which is a representative line of accelerating DDPM following [2]; and (ii) our analysis technique is distinct in DDPM analysis and holds special advantage for certain target distributions.
>
> **Q4:** A recent work of Gatmiry-Kelner-Lee also established bounds on higher-order scores of Gaussian mixtures, though specifically in the context of identity-covariance components. How do their bounds compare to the ones proved in this submission?
>
> **A4:** The work by Gatmiry, Kelner, and Lee [4] addresses the problem for efficiently learning the score function. In comparison, our work addresses the problem “Given an accurately learned score and Hessian function, what guarantees can we obtain for sampling from the data distribution?”. Therefore, our work lies in an orthogonal direction to [4], which is not comparable in a straightforward way. This is also the reason why we have included [4] in the related work on Theory on Score Estimation in the related works.

---

> > ### Author Response · Authors · 2024-11-21
> >
> > **Q5:** To get intuition for the assumption for Theorem 4: Is there an example of a second-order smooth data distribution for which the scores along the trajectory are not all first-order smooth?
> >
> > **A5:** Consider the 1-d case for simplicity. Let $q_0(x) = \exp(-|x^3|-C)$, where $C < \infty$ is some normalizing constant independent of $x$. It can be verified that the second derivative of $\log q_0(x)$ is $-6|x|$, which satisfies the smoothness condition with $M = 6$. Now, given the expression of the forward process, the $Q_t$’s are Gaussian mixtures, with some scaled version $q_0$ being the continuous mixing prior. However, there is no known smoothness constant for the score of a Gaussian mixture, even in the case of discrete mixing priors. Therefore, it is hard to guarantee the first-order smoothness condition along the entire sampling path under this setting.
> >
> > **Q6:** Perhaps it is worth including in the related work some discussion on the use of higher-order scores for diffusion models in practice like GENIE?
> >
> > **A6:** Thank you for pointing this out. We have added the work of GENIE in the Related Works in the Appendix and have provided the paragraph to discuss that work.
> >
> > To obtain higher-order scores, another method is to use automatic differentiation, as in GENIE [6].
> > There, higher-order score functions are used to accelerate the diffusion sampling process empirically. In particular, [6] shows that GENIE achieves better empirical performance than deterministic samplers such as DDIM. Our paper theoretically justifies the accelerated empirical performance of [6] in the regime when the Hessian of $\log q_t$ is well-estimated.
> >
> > We deeply appreciate the reviewer’s thoughtful and inspiring feedback. We hope our responses have satisfactorily addressed your concerns. If so, we wonder if you could kindly consider raising your score. Please don’t hesitate to reach out with any additional questions or comments—we would be delighted to address them.
> >
> > [1] Gen Li, Yu Huang, Timofey Efimov, Yuting Wei, Yuejie Chi, and Yuxin Chen. Accelerating Convergence of Score-Based Diffusion Models, Provably. In the Forty-first International Conference on Machine Learning, 2024.
> >
> > [2] Gen Li, Yuting Wei, Yuxin Chen, and Yuejie Chi. Towards Non-Asymptotic Convergence for Diffusion-Based Generative Models. The Twelfth International Conference on Learning Representations. 2024.
> >
> > [3] Gen Li and Yuling Yan. $O(d/T)$ Convergence Theory for Diffusion Probabilistic Models under Minimal Assumptions. arXiv:2409.18959.
> >
> > [4] Khashayar Gatmiry, Jonathan Kelner, and Holden Lee. Learning Mixtures of Gaussians Using Diffusion Models. arXiv preprint arXiv:2404.18869.
> >
> > [5] Joe Benton, Valentin De Bortoli, Arnaud Doucet, and George Deligiannidis. Nearly $d$-linear convergence bounds for diffusion models via stochastic localization. In The Twelfth International Conference on Learning Representations, 2024.
> >
> > [6] Tim Dockhorn, Arash Vahdat, and Karsten Kreis. GENIE: Higher-Order Denoising Diffusion Solvers. In Advances in Neural Information Processing Systems, 2022.

---

> ### Author Response · Authors · 2024-11-27
>
> Dear Reviewer ZY6t,
>
> We've taken your initial feedback into careful consideration in our response. Could you please check whether our responses have properly addressed your concerns? If so, could you please kindly consider increasing your initial score accordingly? Certainly, we are more than happy to answer your further questions.
>
> Thank you for your time and effort in reviewing our work!

---

> ### Author Response · Authors · 2024-12-01
>
> Dear Reviewer ZY6t,
>
> Since the author-reviewer discussion period will end soon, could you please kindly check whether our responses have appropriately addressed your concerns? If so, could you please kindly consider increasing your initial score accordingly? Please also feel free to let us know if you have further comments.
>
> Thank you for your time and effort in reviewing our work!
>
> Many thanks,
> Authors

---

> > ### Comment · Reviewer_ZY6t · 2024-12-03
> > **Thanks for the response!**
> >
> > A1, A2, and A6 make sense. Regarding A3, this did not affect my score, apologies for the confusion! For A4, I understand that their work addresses a different question, though my comment was specifically about their estimates for the higher-order smoothness of Gaussian mixture scores, which feels like a relevant point of comparison. That said, like A3, this did not affect my impression of the paper, just thought it might be worth mentioning in case it would help the authors refine the manuscript. For A5, I'm not convinced that this example is not first-order smooth along the trajectory, but I agree that this would be hard to establish.
> >
> > While I still have reservations about whether it is reasonable to assume accuracy in estimating the Jacobian, I am happy to raise my score to a 6 given the thoughtful rebuttal.

---

> > > ### Author Response · Authors · 2024-12-03
> > >
> > > Many thanks for your further comments and clarifying your questions, and thank you very much for raising the score! We highly appreciate it.
> > >
> > > Regarding A4, now we see your point, which does make sense. Here are some of our thoughts. In [4], the expression for all higher-orders of the *generator function* of the posterior mean is proved, and an upper bound of its norm in $L^p$ space is also provided (see [4, Lemmas 4.2 and 4.4]). There could be some relationship between the power of generator functions (see definition in [4, Equation (10)]) and that of partial derivatives of log-p.d.f.s. It would be interesting to understand the relationship between the two quantities, and how higher-orders of the generator function could be related to the smoothness properties of the mixture score. These are certainly intriguing questions for us to explore in the future.
> > >
> > > Regarding A5, recently, in [7, Lemma 3.3], the Lipschitz constant for the Gaussian mixture score is calculated exactly. In particular, [7, Lemma 3.3] shows that the constant is a function of the support size of $x_t$. In the example provided, since the support size of $Q_0$ is infinite, there is no finite uniform smoothness constant for the entire support.
> > >
> > > Regarding whether it is reasonable to assume accuracy in estimating the Jacobian, our justification comes from the practical Hessian matching algorithm that we propose in the paper, which could explicitly guarantee accurate Hessian estimation under regularity conditions (see Lemmas 1 and 2). Additionally, as demonstrated in Fig. 1, our proposed accelerated method shows substantial numerical benefits when leveraging Hessian information compared to previous accelerated samplers explored in theoretical studies.
> > >
> > > We thank the reviewer again for your highly insightful feedback and engaging in discussion with us.
> > >
> > > [4] Khashayar Gatmiry, Jonathan Kelner, and Holden Lee. Learning Mixtures of Gaussians Using Diffusion Models. arXiv preprint arXiv:2404.18869.
> > >
> > > [7] Yingyu Liang, Zhenmei Shi, Zhao Song, and Yufa Zhou. Unraveling the Smoothness Properties of Diffusion Models: A Gaussian Mixture Perspective. arXiv preprint arXiv:2405.16418.

---

### Official Review · Reviewer_cL12 · 2024-10-30

**Soundness:** 2
**Presentation:** 2
**Contribution:** 2
**Rating:** 6
**Confidence:** 4

**Summary:**

The paper proposes a new approach to diffusion-based generative modeling that leverages second-order derivatives of the log-density function to achieve accelerated convergence. The framework assumes that the underlying distribution has a finite second-order moment and an infinitely smooth density with respect to the Lebesgue measure. For cases where the distribution lacks infinite smoothness, the algorithm is shown to converge to a convolution of the original distribution with a Gaussian distribution.

The main results provide an upper bound on the KL-divergence between the learned and target distributions. This bound depends on the number of iterations, the error in estimating the gradient and Hessian of the log-density, and a remainder term involving higher-order derivatives of the smoothed density function. The paper also examines Gaussian mixtures and distributions with Lipschitz-continuous log-densities in detail, explicitly characterizing the dependence of the remainder term on the dimension in these cases.

**Strengths:**

1. The paper deals with an important and timely problem.
2. The idea of considering the second-order derivatives to accelerate the convergence is interesting and the results presented in the paper show that it leads to some improvements.
3. The authors did a great job in discussing the prior work and in comparing their results to previous ones.

**Weaknesses:**

### **Main concern**
My primary concern with this paper is the quality of the writing, which requires significant polishing to reach a publishable standard.

The main contributions of this work are clearly mathematical and reside in the theorems presented. The proofs are highly technical, involving lengthy and intricate computations. However, the current presentation makes the proofs challenging to follow and verify. This poses a problem, as reviewers often lack the time to check such detailed proofs thoroughly. If an error were to be found in a proof, it would substantially diminish the paper’s value.

I strongly recommend that the authors dedicate additional time and effort to revising the manuscript for clarity, making the technical sections easier to parse and verify.

### **Specific comments/typos**
- Lines 176-177: Avoid writing expressions such as "for all $1 \leqslant t$" or "$\forall 1 \leqslant t$", as these read ambiguously as "for all 1 that are less than $t$," which is clearly not the intended meaning. Instead, use "for all integers $t \in [0, T]$" or "for all $t \in \\{1, \ldots, T\\}$" to convey the intended range of values.
- line 215: I find it worrisome that the total variation is defined as the supremum over **all** subsets $A$ of $\mathbb R^d$. This should be replaced by all the measurable subsets of $\mathbb R^d$.
- Lines 213–218: I do not see any reason for using the notation $\hat P'_0$ in this paragraph. It would be preferable to maintain consistency with the next paragraph by defining the KL-divergence between two measures $Q$ and $P$ instead.
- Line 220: Instead of 'when $q_0$ exists', it would be more accurate to write "when $Q_0$ is absolutely continuous with respect to $\hat P_0'$." The existence of $q_0$ alone does not ensure that the KL-divergence is finite.
- Line 243: For the Ozaki discretization, it would be more suitable to cite the papers
  1.  T. Ozaki. A bridge between nonlinear time series models and nonlinear stochastic dynamical systems: a local linearization approach. Statistica Sinica, 2(1):113–135, 1992.
  2. O. Stramer and R. L. Tweedie. Langevin-type models. II. Self-targeting candidates for MCMC algorithms. Methodol. Comput. Appl. Probab., 1(3):307–328, 1999b
- Line 244: "estimate" -> "estimates"
- Eq. (6): There is no guarantee that this matrix is positive semi-definite and, therefore, can be used as a covariance matrix. I agree that for most $x_t$ this will be the case for large $T$, but it is necessary here to have a definition that works for **every** $x_t$.
- Line 251: "denote as Hessian matching" -> "refer to as Hessian matching".
- Line 262: It is better to remove the second equality here and to keep it in the claim of Lemma 1 only.
- Line 269 (and in several places in the proof): instead of writing the marginal distributions of $X_0$ and $\bar W_t$, it is necessary to indicate the joint distribution of these random variables. It suffices to write $(X_0,\bar W_t)\sim Q_0\otimes \mathcal N(0,I_d)$.
- line 280: "satisfy" -> "satisfies"
- lines 277-289: It is generally not recommended to refer to material, such as an assumption, before it has been introduced. In this case, Assumption 3 is mentioned before it is properly stated. Similarly, within the text of Assumption 3, Assumption 4 is referenced prior to its statement. I strongly recommend that the authors adjust the flow of the text to avoid such inversions.
- Lemma 2: In the statement of this lemma and in other places, the meaning of the Landau notation $O(\cdot)$ is unclear. It should be specified that this notation is understood in the asymptotic limit $T \to \infty$. Additionally, in Eq. (10), there should be no term depending on $t$ on the right-hand side. Either $\alpha_t$ should be replaced with $\alpha_T$, or, preferably, the left-hand side should be divided by $(1 - \alpha_t)^2$ within the maximum. **My recommendation would be to avoid these notation and to prefer more conventional writing** ``there exist constants $C_1$ and $C_2$ such that ... for all $T\in\mathbb N$''.
- Line 281: $\alpha_t$ is not defined in (1). I guess you wanted to write Definition 1 instead of (1).
- Lines 287–288: Since $H_t$ first appears on line 287, it would be clearer to move the indication "defined by (8)" forward to accompany its first mention.
- Line 287: It is unclear why the factor $(1 - \alpha_t)$ is retained on the left-hand side while $O(1 - \alpha_t)$ is used on the right-hand side. Rather than writing $(1 - \alpha_t) g_t = O(1 - \alpha_t)$, wouldn’t it be clearer to simply write $g_t = O(1)$? Additionally, should this relation hold uniformly for $\ell \geqslant 1$?
- Line 300: What is the precise meaning of *analytic* in this context? I was unable to find a definition in the text, either in the main body or in the appendix.
- Assumption 4: Same remark as above, I guess it is possible to suppress the factors $(1-\alpha_t)^{p\ell}/2$ on both sides of the equalities.
- Assumption 4: What is $p$ in this assumption? Should the boundedness of $p$-th order derivatives hold for every $p$? Should the upper bound be independent of $p$?
- Definition 1: This should be stated as a condition rather than a definition.
- Line 342: I did not find the definition of $\widehat P_0'$.
- Line 376: "titling" -> "tilting"
- Line 389: "rest two terms" -> "remaining two terms".

- I guess in the claim of Theorem 1 it is also necessary to assume that $q_0(x)>0$ for all $x$. Indeed, if this condition is not satisfied, the term $\log q_0(\mu_1(X_1))$ appearing in the right-hand side of the equation on line 344 is meaningless.

**Questions:**

I have several questions regarding Theorem 1 and Corollary 1, which form the central results of the paper.

In Theorem 1, constants are encapsulated within the notation $\lesssim$. I assume these constants depend on the distribution $Q_0$. It would be valuable to clarify which specific characteristics of $Q_0$ influence these constants. In particular, given that the theorem assumes the density $q_0$ of $Q_0$ is analytic, it would be helpful to know whether the constants depend on the derivatives of $q_0$.

If these constants are independent of $q_0$ and its derivatives, this would imply that the claim of Theorem 1 might not require the assumption that $q_0$ is infinitely smooth. Conversely, if the constants do depend on $q_0$ or its derivatives, then it raises the possibility that Theorem 1's claim may not be applicable in proving Corollary 1.

---

> ### Author Response · Authors · 2024-11-21
>
> We thank the reviewer for providing the highly inspiring feedback. In the revised paper, we have made changes based on your review comments and have highlighted all revisions in blue fonts.
>
> **Q1:** Eq. (6): There is no guarantee that this matrix is positive semi-definite and, therefore, can be used as a covariance matrix. I agree that for most $x_t$ this will be the case for large $T$, but it is necessary here to have a definition that works for every $x_t$.
>
> **A1:** Thanks for the great question/suggestion! To proceed more rigorously, we can project the matrices $\Sigma_t$ and $\widehat{\Sigma}_t$ onto the space of positive-semi definite (PSD) matrices for those $x_t$’s where either of these two matrices is not PSD (through eigen-decomposition and zeroing-out all negative eigenvalues). Since the measure of the events containing such bad $x_t$’s decreases to zero, all theoretical results in this paper, which are derived in expectation, will not be affected. We have made a footnote there in our revision.
>
> **Q2:** lines 277-289: It is generally not recommended to refer to material, such as an assumption, before it has been introduced. In this case, Assumption 3 is mentioned before it is properly stated. Similarly, within the text of Assumption 3, Assumption 4 is referenced prior to its statement. I strongly recommend that the authors adjust the flow of the text to avoid such inversions.
>
> **A2:** Thank you for your suggestion! To prevent such inversions, we have moved the statement of Lemma 2 into the appendix. Note that Lemma 2 is not the main contribution of this paper. We have also moved all discussions of the assumptions to be after the assumptions are stated.
>
> **Q3:** Line 287: It is unclear why the factor $(1-\alpha_t)$ is retained on the left-hand side while $O(1-\alpha_t)$ is used on the right-hand side. Rather than writing $(1-\alpha_t)g_t = O(1-\alpha_t)$, wouldn’t it be clearer to simply write $g_t = O(1)$? Additionally, should this relation hold uniformly for $\ell \geq 1$?
>
> **A3:** Many thanks for the question! This common term was initially kept there for our own understanding of the original terms on both sides of the equation. Certainly, it can go away for simplicity without losing mathematical rigor. We have changed the notation for the requirement of $H_t$ and make it simpler (in Lines 298 and 935). Also, as suggested, this relation needs to hold uniformly for $\ell \geq 1$.
>
> **Q4:** Line 300: What is the precise meaning of analytic in this context? I was unable to find a definition in the text, either in the main body or in the appendix.
>
> **A4:** This “analytic” is in the context of function analysis, where a function is analytic means that its Taylor series converges to the functional value at each point in the domain. We have provided this piece of clarification in the footnote of Assumption 2.
>
> **Q5:** Assumption 4: Same remark as above, I guess it is possible to suppress the factors $(1-\alpha_t)^{p \ell}$ on both sides of the equalities.
>
> **A5:** Thank you for the suggestion! In our revised draft, we have made our Assumption 4 clearer by suppressing the factors and left the more general version (which is Assumption 5) in the Appendix. In particular, Assumption 4 is still satisfied under two cases: (1) when $Q_0$ has finite variance, and (2) when $Q_0$ is Gaussian mixture.
>
> **Q6:** Assumption 4: What is $p$ in this assumption? Should the boundedness of $p$-th order derivatives hold for every $p$? Should the upper bound be independent of $p$?
>
> **A6:** In Assumption 4, $p$ is the order of partial derivatives. From Lemmas 15 and 17, the $p$-th order derivatives are a function of $p$, which might not have an upper bound of a constant in $T$. Nevertheless, Assumption 4 is not as strong as it appears to be. We can verify Assumption 4 on the following two common cases: (1) when $Q_0$ has finite variance, and (2) when $Q_0$ is Gaussian mixture. Case 1 clearly covers a broad set of target distributions of practical interest, such as images, and many theoretical studies of diffusion models have been specially focused on such a distribution [1-2]. Case 2 has also been well studied for diffusion models [3-4].
>
> Note that Assumption 4 can be verified for Gaussian mixture $Q_0$’s (Lemma 14) and for a proper choice of $\alpha_t$ (Lemma 17).
>
> **Q7:** Line 342: I did not find the definition of $\widehat{P}’_0$.
>
> **A7:** Thank you.  $\widehat{P}’_t$ is the distribution at intermediate time $t$ using the estimated scores and Hessians. Thus, $\widehat{P}’_0$ is the distribution of the final output from the accelerated diffusion process. We added the definition of $\widehat{P}’_t$ in Section 3.1.

---

> > ### Author Response · Authors · 2024-11-21
> >
> > **Q8:** I guess in the claim of Theorem 1 it is also necessary to assume that $q_0(x) > 0$ for all $x$.
> >
> > **A8:** Thanks for pointing this out. We added the condition $q_0(x) > 0$ for all $x$ in our Assumption 2, which is primarily used in Theorem 1.
> >
> > **Q9:** In Theorem 1, constants are encapsulated within the notation $\lesssim$. I assume these constants depend on the distribution $Q_0$. It would be valuable to clarify which specific characteristics of $Q_0$ influence these constants. In particular, given that the theorem assumes the density $q_0$ of $Q_0$ is analytic, it would be helpful to know whether the constants depend on the derivatives of $q_0$.
> >
> > **A9:** Under the assumptions of Theorem 1, the $\lesssim$ only omits terms that decay faster than the rate-determining terms being displayed; there are **no** $Q_0$-related coefficients/constants hidden in these rate-determining terms themselves (cf. Equations (21), (26), Lemmas 4 and 11).
> > Higher-order derivatives of $\log q_t$ are contained in those higher-order (Taylor) terms that decay faster than those rate-determining terms.
> >
> > **Q10:** If these constants are independent of $q_0$ and its derivatives, this would imply that the claim of Theorem 1 might not require the assumption that $q_0$ is infinitely smooth. Conversely, if the constants do depend on $q_0$ or its derivatives, then it raises the possibility that Theorem 1's claim may not be applicable in proving Corollary 1.
> >
> > **A10:** Many thanks for the question! Even though higher-order derivatives do not contribute to the rate-determining terms, we still need the derivatives of $\log q_0$ to satisfy Assumption 4, which guarantees that the Taylor expansion is valid and all higher-order Taylor terms indeed decay faster. A necessary condition of this, then, is that $q_0$ is infinitely smooth. However, in Corollary 1, we can show the convergence for any $Q_0$ with finite variance, without any smoothness constraint (as in Assumptions 2 or 4). This is because of the technique of early-stopping (before reaching $Q_0$). Although $q_0$ does not exist or is not smooth, all Gaussian perturbations of $Q_0$ still have a density $q_t$ and are still smooth. Note that Assumption 4 can be satisfied for all $Q_1,\dots,Q_T$ upon choosing a proper sequence of $\alpha_t$ (Theorem 3).
> >
> > We sincerely thank the reviewer once again for your insightful and inspiring comments. We hope our responses have adequately addressed your concerns. If so, we wonder if you could kindly consider raising your score. Of course, we would be more than happy to address any further questions or concerns you may have.
> >
> > [1] Gen Li, Yu Huang, Timofey Efimov, Yuting Wei, Yuejie Chi, and Yuxin Chen. Accelerating Convergence of Score-Based Diffusion Models, Provably. In the Forty-first International Conference on Machine Learning, 2024.
> >
> > [2] Gen Li, Yuting Wei, Yuxin Chen, and Yuejie Chi. Towards Non-Asymptotic Convergence for Diffusion-Based Generative Models. In The Twelfth International Conference on Learning Representations. 2024.
> >
> > [3] Khashayar Gatmiry, Jonathan Kelner, and Holden Lee. Learning Mixtures of Gaussians Using Diffusion Models. arXiv preprint arXiv:2404.18869.
> >
> > [4] Sitan Chen, Vasilis Kontonis, and Kulin Shah. Learning General Gaussian Mixtures with Efficient Score Matching. arXiv preprint arXiv:2404.18893.

---

> ### Author Response · Authors · 2024-11-27
>
> Dear Reviewer cL12,
>
> We've taken your initial feedback into careful consideration in our response. Could you please check whether our responses have properly addressed your concerns? If so, could you please kindly consider increasing your initial score accordingly? Certainly, we are more than happy to answer your further questions.
>
> Thank you for your time and effort in reviewing our work!

---

> ### Author Response · Authors · 2024-12-01
>
> Dear Reviewer cL12,
>
> Since the author-reviewer discussion period will end soon, could you please kindly check whether our responses have appropriately addressed your concerns? If so, could you please kindly consider increasing your initial score accordingly? Please also feel free to let us know if you have further comments.
>
> Thank you for your time and effort in reviewing our work!
>
> Many thanks,
> Authors

---

> > ### Comment · Reviewer_cL12 · 2024-12-02
> > **Response to the rebuttal**
> >
> > I would like to thank the authors for their very detailed answers to my review and for updating their manuscript. I still believe that the manuscript would benefit from a substantial révision, but I nevertheless increase my score since I find that even in its current form the paper could be of interest for the ML community.

---

### Official Review · Reviewer_gLE4 · 2024-11-03

**Soundness:** 3
**Presentation:** 3
**Contribution:** 3
**Rating:** 6
**Confidence:** 3

**Summary:**

This paper introduces a novel accelerated stochastic DDPM sampler that leverages the tilting factor representation and Tweedie's formula. This approach is designed to accommodate a broader class of target distributions, requiring weaker conditions than previous methods.

**Strengths:**

The paper introduces a novel accelerated stochastic DDPM sampler and demonstrates its applicability to various distribution classes while relying on weaker smoothness conditions and the bounded second moment condition. Additionally, it establishes the first acceleration guarantees for Gaussian mixture models.

**Weaknesses:**

Assumption 4 plays a crucial role in the construction of the tilting factor, requiring that the score function has derivatives of all orders and that the $p$-th moment is bounded $p \ge 1$. In my view, this condition is not as "rather soft" as suggested in the submission. The requirement for all derivatives to exist means that the score function is not merely smooth but infinitely differentiable, a level of regularity that is often absent in practical score functions, particularly for complex or non-smooth distributions.
Furthermore, the requirement that the $p$-th moment of the derivatives is bounded indicates that the derivatives exist and do not grow excessively. This can be quite a restrictive requirement, especially for distributions with heavy tails or irregularities. While this assumption is essential for maintaining theoretical rigor, it may limit the applicability of the models to a narrower range of problems.

**Questions:**

The proof is quite lengthy, and I did not have the opportunity to examine all the details. It would be beneficial if the author could elaborate on how the accelerated convergence is achieved using the proposed novel proof technique (specifically, in Step 2 regarding the tilting factor). Additionally, restating Tweedie's formula and clarifying its extension in this context would help enhance the clarity of the proof for readers.

---

> ### Author Response · Authors · 2024-11-21
>
> We thank the reviewer for providing the highly inspiring feedback. In the revised paper, we have made changes based on your review comments and have highlighted all revisions in blue fonts.
>
> **Q1:** Assumption 4 plays a crucial role in the construction of the tilting factor, requiring that the score function has derivatives of all orders and that the $p$-th moment is bounded $p \geq 1$. In my view, this condition is not as "rather soft" as suggested in the submission. The requirement for all derivatives to exist means that the score function is not merely smooth but infinitely differentiable, a level of regularity that is often absent in practical score functions, particularly for complex or non-smooth distributions. Furthermore, the requirement that the $p$-th moment of the derivatives is bounded indicates that the derivatives exist and do not grow excessively. This can be quite a restrictive requirement, especially for distributions with heavy tails or irregularities. While this assumption is essential for maintaining theoretical rigor, it may limit the applicability of the models to a narrower range of problems.
>
> **A1:** Thanks for the question. We can verify Assumption 4 on the following two common cases: (1) when $Q_0$ has finite variance, and (2) when $Q_0$ is Gaussian mixture. Case 1 clearly covers a broad set of target distributions of practical interest, such as images, and many theoretical studies of diffusion models have been specially focused on such a distribution [1-2]. Case 2 has also been well studied for diffusion models [3-4].
>
> A special note is that Case 1 in fact includes substantial complex or non-smooth distributions. To see this, even though $\log q_0$ might not be smooth, all Gaussian perturbed $\log q_t$’s are still infinitely smooth, which allow us to use early-stopping (Corollary 1 and Theorem 3). Furthermore, we do not require Assumption 4 in case that the Hessian of $\log q_0$ is smooth (Theorem 4).
>
> We have added these explanations below Assumption 4 in the revised paper.
>
> **Q2:** The proof is quite lengthy, and I did not have the opportunity to examine all the details. It would be beneficial if the author could elaborate on how the accelerated convergence is achieved using the proposed novel proof technique (specifically, in Step 2 regarding the tilting factor). Additionally, restating Tweedie's formula and clarifying its extension in this context would help enhance the clarity of the proof for readers.
>
> **A2:** Many thanks for the question. We added the following paragraph to explain Step 2 in the revision, especially on how the acceleration is achieved:
>
> For regular DDPMs, there is no control for the variance of the reverse sampling process, and thus $B_t(x_t) \equiv 0$. In this case, the dominating rate is determined by the expected values of $T_2$. With the variance correction in our accelerated sampler, the corresponding $B_t(x_t)$ enables us to cancel out the second-order Taylor term (see Lemma 11). As a result, the rate-determining term becomes the expected values of $T_3$, which decays faster. Thus, the acceleration is achieved.
>
> We also provide the following clarification for how we extend Tweedie’s formula in Step 4:
>
> To calculate posterior moments, we extend Tweedie's formula in a non-trivial way. Whereas the original Tweedie's formula provides an explicit expression for the posterior mean for Gaussian perturbed observations, we explicitly calculate the first six centralized posterior moments and provide the asymptotic order of all higher-order moments, drawing techniques from combinatorics.
>
> We thank the reviewer again for your highly inspiring comments. We hope that our responses resolved your concerns. If so, we wonder if the reviewer could kindly consider to increase your score. Certainly, we are more than happy to answer your further questions.
>
> [1] Gen Li, Yu Huang, Timofey Efimov, Yuting Wei, Yuejie Chi, and Yuxin Chen. Accelerating Convergence of Score-Based Diffusion Models, Provably. In the Forty-first International Conference on Machine Learning, 2024.
>
> [2] Gen Li, Yuting Wei, Yuxin Chen, and Yuejie Chi. Towards Non-Asymptotic Convergence for Diffusion-Based Generative Models. In The Twelfth International Conference on Learning Representations. 2024.
>
> [3] Khashayar Gatmiry, Jonathan Kelner, and Holden Lee. Learning Mixtures of Gaussians Using Diffusion Models. arXiv preprint arXiv:2404.18869.
>
> [4] Sitan Chen, Vasilis Kontonis, and Kulin Shah. Learning General Gaussian Mixtures with Efficient Score Matching. arXiv preprint arXiv:2404.18893.

---

> ### Author Response · Authors · 2024-11-27
>
> Dear Reviewer gLE4,
>
> We've taken your initial feedback into careful consideration in our response. Could you please check whether our responses have properly addressed your concerns? If so, could you please kindly consider increasing your initial score accordingly? Certainly, we are more than happy to answer your further questions.
>
> Thank you for your time and effort in reviewing our work!

---

> ### Author Response · Authors · 2024-12-01
>
> Dear Reviewer gLE4,
>
> Since the author-reviewer discussion period will end soon, could you please kindly check whether our responses have appropriately addressed your concerns? If so, could you please kindly consider increasing your initial score accordingly? Please also feel free to let us know if you have further comments.
>
> Thank you for your time and effort in reviewing our work!
>
> Many thanks,
> Authors

---

### Meta-Review · Area_Chair_w8tS · 2024-12-22

**Metareview:**

This paper considers the problem of establishing convergence bounds for diffusion models.
Authors assume that score can be estimated accurately, and provide guarantees in terms of error tolerance. Authors improve the existing rates of diffusion models under finite variance condition, with better  dimension and tolerance dependence. This is achieved by assuming access to Jacobian.


This paper was reviewed by three expert reviewers the following Scores/Confidence: 6/3, 6/3, 6/4. I think the paper is studying an interesting topic and the results are relevant to ICLR community. The following concerns were brought up by the reviewers:

- Several typos and ambiguous statements were pointed by the reviewers. These should be carefully addressed. I strongly encourage the authors to revise their manuscript, and improve the text based on the feedback.

- Assumed access to Jacobian may not be readily avaialble in practice. This should be clarified/discussed in detail in the paper.


Authors should carefully go over reviewers' suggestions and address any remaining concerns in their final revision. Based on the reviewers' suggestion, as well as my own assessment of the paper, I recommend including this paper to the ICLR 2025 program.

**Additional Comments On Reviewer Discussion:**

Authors successfully answered reviewers' question during the discussion period.

---

### Decision · Program_Chairs · 2025-01-22

Accept (Poster)